# Inhibition of the minor spliceosome restricts the growth of a broad spectrum of cancers

Karen Doggett [1,2,✉], Kimberly J Morgan [1,2], Anouk M Olthof[3,4], Stephen Mieruszynski[1,2], Benjamin B Williams[1,2], Alexandra L Garnham[1,2], Michael J G Milevskiy[1,2], Lachlan Whitehead[1,2], Janine Coates[1], Michael Buchert [5,6], Robert J J O'Donoghue[5,6], Thomas E Hall [7], Tracy L Putoczki[1,2], Matthias Ernst [5,6], Kate D Sutherland[1,2], Rahul N Kanadia [3,8] & Joan K Heath [1,2]

## Abstract

**Minor splicing is an under-appreciated splicing system required for the correct expression of ~700 genes in the human genome. This small subset of genes (0.35%) harbours introns containing non-canonical splicing sequences that are recognised uniquely by the minor spliceosome and cannot be processed by the major spliceosome. Using in vivo zebrafish and mouse cancer models, we show that heterozygous expression of *Rnpc3*, encoding a unique protein component of the minor spliceosome, restricts the growth and survival of liver, lung and gastric tumours without impacting healthy cells. *RNPC3* knockdown in human lung cancer-derived A549 cells also impairs cell proliferation and RNA-seq analysis reveals a robust and selective disruption to minor intron splicing and transcription-wide effects on gene expression. We further demonstrate that these perturbations are accompanied by DNA replication stress, DNA damage, accumulation of TP53 protein and activation of a Tp53-dependent transcriptional program that induces cell cycle arrest and apoptosis. Together our data reveal a vulnerability of cancer cells to minor splicing inhibition that restricts tumour growth.**

**Keywords** Cancer; RNA Processing; Minor Splicing; RNPC3; DNA Damage
**Subject Categories** Cancer; RNA Biology

## Introduction

Splicing is required to generate the correct mRNA template for the synthesis of proteins and is an essential step in gene expression (Berget et al, 1977). For the most part, pre-mRNA splicing is carried out by the major or U2-dependent spliceosome, which removes approximately 99.65% of all introns. However, a second splicing system, known as the minor or U12-dependent spliceosome (Hall and Padgett, 1996; Tarn and Steitz, 1996), is required to excise the remaining 0.35% of introns, which equates to 755 and 706 minor introns distributed across 699 human and 650 mouse genes, respectively (Olthof et al, 2019). Minor introns are readily distinguished from major introns by the presence of two highly conserved sequence motifs: a 7 bp sequence immediately downstream of the 5' splice site (ss) and a branch point sequence (BPS) upstream of the 3'ss. They also lack the highly conserved polypyrimidine tract that is found upstream of the 3'ss of major introns (Burge et al, 1998). These conserved features of minor introns are recognized uniquely by a complex known as the U11/U12 di-snRNP containing 2 small nuclear RNAs (snRNAs: U11, U12) and 8 minor spliceosome-specific proteins that do not bind to major introns (Bai et al, 2021; de Wolf et al, 2021; Will et al, 2004; Will et al, 1999). Recently, a cryo-EM reconstruction yielded major insights into the architecture of a 13-subunit U11 snRNP revealing the structure of a core complex comprising 5 interacting minor spliceosome proteins (ZMAT2, SNRNP25, SNRNP35, SNRNP48 and PDCD7) and the mechanism of 5'ss recognition by the U11 snRNP (Zhao et al, 2025). Upon recognition and binding to minor introns, the U11/U12 di-snRNP is rearranged by the addition of further unique components: the U4ATAC and U6ATAC snRNAs and 5 additional proteins, creating an activated complex (B') that stabilises the conformation of the catalytic core (Bai et al, 2021). From thereon, the downstream events of minor splicing are catalysed by components shared with the major splicing machinery (Will et al, 1999).

Despite their low frequency in the genome, the importance of minor introns is indicated by their evolutionary conservation in early eukaryotes (Russell et al, 2006), fungi, land plants and animals. Intriguingly, minor introns are not distributed randomly throughout the genome; instead, they are over-represented in sets of genes that perform essential cellular functions such as DNA replication, RNA processing, including transcription and splicing, RNA quality control and translation (Doggett et al, 2018). Of interest in the context of cancer, minor introns are found in genes with established roles in

[1]Walter and Eliza Hall Institute of Medical Research, Parkville, VIC 3052, Australia. [2]Department of Medical Biology, University of Melbourne, Parkville, VIC 3052, Australia. [3]Physiology and Neurobiology Department, University of Connecticut, Storrs, CT 06269, USA. [4]Department of Cellular and Molecular Medicine, University of Copenhagen, Copenhagen, Denmark. [5]Cancer and Inflammation Laboratory, Olivia Newton-John Cancer Research Institute, Heidelberg, VIC 3084, Australia. [6]La Trobe University School of Cancer Medicine, Heidelberg, VIC 3084, Australia. [7]Institute for Molecular Bioscience, University of Queensland, St Lucia, QLD 4072, Australia. [8]Institute for Systems Genomics, University of Connecticut, Storrs, CT 06269, USA. ✉E-mail: doggett.k@wehi.edu.au

oncogenic signalling pathways, including the proto-oncogenes, *BRAF* and *RAF1*, and 11 out of 14 mitogen-activated protein kinase (MAPK) family genes, including *ERK*, *JNK*, *p38* and their respective isoforms.

One of the direct consequences of impaired minor splicing is the accumulation of aberrant pre-mRNA transcripts exhibiting features such as minor intron retention (IR) and exon skipping (Lotti et al, 2012) that are often retained in the nucleus. In addition, the generation of frameshifts and premature stop codons in the coding sequence may lead to the production of truncated proteins or the degradation of improperly processed mRNAs by nonsense mediated decay (NMD) (Middleton et al, 2017). Severe consequences may result from the incorrect expression of minor intron containing genes (MIGs), including impaired cell growth, proliferation and survival, particularly during development (Argente et al, 2014; Baumgartner et al, 2018; Doktor et al, 2017; Elsaid et al, 2017; Horiuchi et al, 2018; Madan et al, 2015).

Our interest in minor splicing arose from a focused genetic screen in zebrafish, which yielded several mutants with impaired growth of the developing liver, pancreas and intestine. We demonstrated that one of these mutants harboured a single nucleotide variation in both copies of *rnpc3*, encoding a 65 kDa RNA-binding, protein component of the minor spliceosome (Markmiller et al, 2014). In mice, we showed that *Rnpc3* is essential for pre-implantation development and that induced recombination of both alleles of the *Rnpc3* locus in adults severely impairs the homeostasis of the gastrointestinal epithelium, hematopoietic compartment and thymus (Doggett et al, 2018). Our results indicated a heightened requirement for minor splicing in tissues undergoing rapid, continuous cell cycling, compared to quiescent tissues, and prompted us to think that minor splicing may be an Achilles' heel of all highly proliferative tissues, including cancers.

The first publication to provide unequivocal evidence for this hypothesis demonstrated the value of minor splicing impairment in therapy-resistant prostate cancer (Augspach et al, 2023). In this study, minor splicing efficiency was impaired by siRNA-mediated knockdown of *U6ATAC* snRNA in multiple prostate cancer cell lines and patient-derived organoids. The authors found that disrupting minor splicing was more effective at reducing cancer cell survival than current state-of-the-art therapies (Augspach et al, 2023). Our study complements these in vitro findings by demonstrating that heterozygous *Rnpc3* expression is sufficient to reduce tumour burden in a variety of in vivo cancer models from zebrafish and mice, including those driven by mutant Kras oncogenes. We also used the human lung adenocarcinoma cell line, A549, to show that reduced expression of *RNPC3*, and two other minor spliceosome-specific components, *PDCD7* and *U12 snRNA*, causes minor intron retention and alternative splicing. Accumulation of aberrant pre-RNA transcripts then leads to DNA replicative stress, an increase in DNA damage, TP53 stabilisation and activation of a TP53-dependent transcriptional program that promotes cell cycle arrest and apoptosis, thereby restricting tumour burden.

## Results

### Heterozygous loss of *rnpc3* reduces liver overgrowth in a *kras*^*G12V* model of hepatocellular carcinoma

Zebrafish and mice carrying a single loss-of-function *rnpc3/Rnpc3* allele develop normally, achieve sexual maturity, and exhibit a

normal lifespan (Doggett et al, 2018; Markmiller et al, 2014). To determine the impact of *rnpc3* heterozygosity in a cancer setting, we chose a zebrafish model of hepatocellular carcinoma (HCC) in which rapid growth and proliferation of hepatocytes is driven by a mutant *kras*^*G12V* transgene. In this transgenic line, *Tg(fabp10:rt-TA2s-M2;TRE2:EGFP-kras*^*G12V*), hereafter *TO(kras*^*G12V*), doxycycline induces the hepatocyte-specific expression of *EGFP-kras*^*G12V* causing hepatocyte hyperplasia and an increase in liver volume (Chew et al, 2014; Morgan et al, 2023), which we quantified by two-photon microscopy. We used a second transgenic zebrafish line, *2-CLiP* or *LiPan* (Korzh et al, 2008), which exhibits constitutive, hepatocyte-specific expression of dsRed, but no oncogenic transgene, to acquire fluorescent images of *kras*^*G12V*-negative controls.

First, we established that heterozygous *rnpc3* larvae at 7 days post-fertilisation (dpf) contained less *rnpc3* mRNA (~40% reduction) than *rnpc3*^*+/+* siblings, as expected (Fig. 1A). We then determined whether the mean volume of livers from *2-CLiP* larvae was affected by *rnpc3* genotype. We found that irrespective of whether the *2-CLiP* larvae were WT or HET for *rnpc3*, the shape and volume of the livers were unchanged (red livers; Fig. 1B,C), demonstrating that heterozygous expression of *rnpc3* is sufficient for normal liver growth during development.

In *TO(kras*^*G12V*) larvae on a WT *rnpc3* background, the dox-induced expression of *kras*^*G12V* transgene increased mean liver volume by >5-fold compared to *2-CLIP* livers; however, in *rnpc3* HETS this increase was pared back significantly (to threefold), indicating that a ~40% reduction in *rnpc3* expression restricted mutant *kras*^*G12V*-driven hyperplasia. One possibility for this observation is that *rnpc3* heterozygosity impaired liver overgrowth in dox-induced *TO(kras*^*G12V*) larvae by interfering, either directly or indirectly, with the production of GTP-activated Ras proteins. We tested this by comparing the amount of EGFP-KrasG12V-GTP protein in lysates from separate pools of *rnpc3*^*+/+* and *rnpc3*^*+/−* larvae using an active Ras pull down assay followed by western blot (Baker and Rubio, 2021). We observed robust levels of EGFP-KrasG12V-GTP protein, which were not affected by the *rnpc3* genotype (Fig. EV1A) ruling out a direct effect of *rnpc3* heterozygosity on the level of the EGFP-KrasG12V-GTP protein.

We also looked at hepatocyte hyperplasia in *TO(kras*^*G12V*) adults (3.5–4 months of age) by treating them daily for 7 d with fresh 20 mg/L dox (Morgan et al, 2023). We observed a large increase in liver weight (expressed as a percentage of total body weight) in dox-treated adults compared to untreated adults. The %age weight of the liver in *rnpc3* WT adults was >12%, compared to only 2% in untreated adults (Fig. EV1B). Remarkably, the %age weight of the liver fell to 7% in *rnpc3* HET adults, showing again that whereas *rnpc3* heterozygosity is sufficient for normal liver growth in both larvae and adults, it is rate-limiting for mutant *kras*^*G12V*-driven hepatocyte hyperplasia.

To determine whether the reduced liver volume observed in *rnpc3* HETS was caused by changes in cell cycle progression, we treated live zebrafish larvae (7 dpf) with the thymidine analogue, N-ethynyl-2'-deoxyuridine (EdU; 2 μM) for 2 h, followed by 1 h in fresh egg water, and counted the number of hepatocytes in S-phase. In *TO(kras*^*G12V*) larvae expressing WT *rnpc3*, the %age of EdU-positive hepatocyte nuclei was approximately 31%, compared to 19% in *rnpc3*^*+/−*;*TO(kras*^*G12V*) larvae (Fig. 1D,E), equating to a 40% reduction in the number of hepatocytes in S-phase. To determine whether cell death contributed to the reduced liver volume

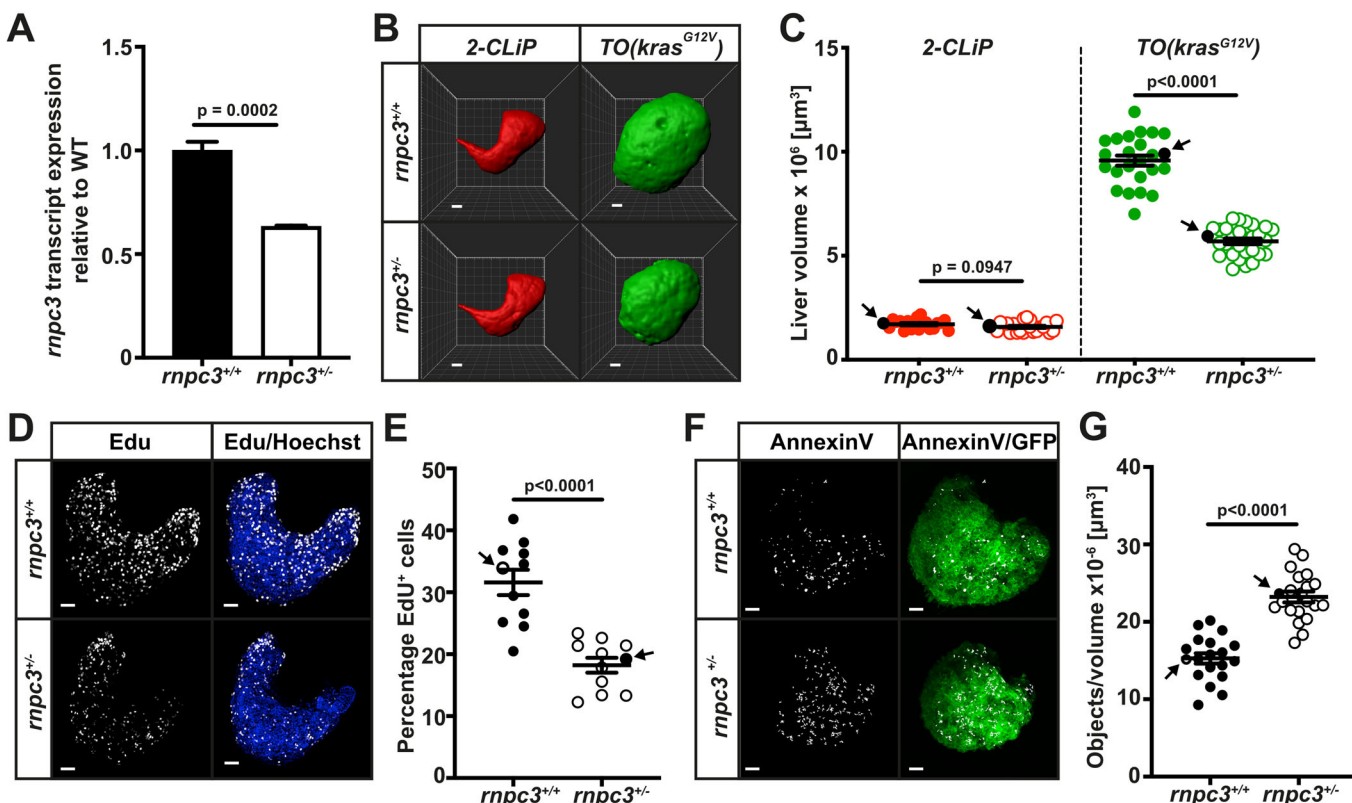

**Figure 1. Heterozygous loss of *rnpc3* reduces tumour burden in a *kras^G12V*-driven zebrafish model of hepatocellular carcinoma (HCC).**

(A) RT-qPCR analysis of *rnpc3* mRNA extracted from independent pools of *rnpc3^+/−* larvae aged 7 days post-fertilization (dpf) compared to *rnpc3^+/+* larvae ($n = 3$ biological replicates). (B) Representative Imaris three-dimensional reconstructions of *2-CLiP* and dox-treated *TO(kras^G12V)^T/+* livers of 7 dpf larvae of the indicated *rnpc3* genotype. Scale bar is 25 μm. (C) Liver volume in *2-CLiP* (red symbols, $n = 26$ or $n = 29$) and *TO(kras^G12V)* transgene (green circles, $n = 24$ or $n = 28$) zebrafish at 7 dpf. Black arrows on the graphs indicate the data points (black symbols) that correspond to the representative images shown in (B). (D, E) EdU (white dots) and Hoechst 33342-positive hepatocyte nuclei ($n = 11$) at 7 dpf. Black arrows on the graphs indicate the data points that correspond to the representative images shown in (D). Scale bar is 50 μm. (F, G) Foci of AnnexinV-mKate fluorescence in *TO(kras^G12V)* livers ($n = 19$ and 20) at 7 dpf. Black arrows on the graphs indicate the data points that correspond to the representative images shown in (F). Scale bar is 50 μm. Data are represented as mean ± SEM. Significance was assessed using a Student's $t$ test, $P < 0.05$. Source data are available online for this figure.

observed in *rnpc3* HETS, we introduced an *annexinV-mkate* cell death reporter transgene into the *TO(kras^G12V)* model to mark cells undergoing apoptosis (Hall et al, 2019). We observed a 40% increase in the number of Annexin-mKate fluorescent foci in *rnpc3* HETS, compared to *rnpc3* WT (Fig. 1F,G). These data indicate a vulnerability of *kras^G12V*-expressing hepatocytes to *rnpc3* heterozygosity that results in a decreased number of hepatocytes in S-phase of the cell cycle alongside an increased number of hepatocytes undergoing cell death, both contributing to a reduction in hyperplastic growth.

### *rnpc3* heterozygosity combines with *kras^G12V* to activate a Tp53 DNA damage response that restricts hepatocyte hyperplasia

Due to its known role in promoting cell cycle arrest and apoptosis, we investigated whether the tumour suppressor protein Tp53 was activated in response to *rnpc3* heterozygosity. To do this, we measured the levels of Tp53 protein in lysates of micro-dissected livers expressing the *kras^G12V* transgene. We observed no Tp53 signals from livers not expressing the *kras^G12V* transgene (denoted *TO(kras^G12V)^+/+*), or the remaining body after

liver removal, irrespective of *rnpc3* genotype (Fig. 2A). However, we did detect a weak Tp53 signal in extracts of *rnpc3* WT *kras^G12V*-expressing livers, denoted *TO(kras^G12V)^Tg/+*, suggestive of hyperplastic hepatocytes undergoing mild oncogenic stress (Fig. 2A). We detected a much higher (>2.4-fold) Tp53 signal in *rnpc3* HET livers, indicating that hyperplastic hepatocytes with only one functional *rnpc3* allele were under more severe stress. To test whether Tp53 was necessary for the reduction in hepatocyte hyperplasia in *rnpc3^+/−;TO(kras^G12V)* livers, we repeated the experiment in the presence and absence of functional Tp53. To do this, we introduced two *tp53^M214K* alleles (hereafter *tp53^m/m*) into the *TO(kras^G12V)^Tg/+* model of HCC. (Berghmans et al, 2005). *tp53^m* alleles carry a missense mutation in the DNA-binding domain of Tp53, rendering the Tp53 protein non-functional. Compared to *rnpc3^+/+;TO(kras^G12V)^Tg/+* livers expressing WT *tp53* alleles (Fig. 2B,C, blue circles), mean liver volume was increased by 33% on a *tp53^m/m* background (Fig. 2B,C, red squares), indicating that Tp53 normally restricts liver overgrowth in the zebrafish HCC model. Consistent with Fig. 1B,C, we observed a 40% decrease in mean liver volume in *rnpc3* HETS compared to *rnpc3* WT (Fig. 2B,C, blue open circles). However, in the absence of functional Tp53, *rnpc3* heterozygosity reduced liver volume by only 8% compared to larvae with WT *rnpc3* (Fig. 2B,C, open red squares). This demonstrates that the capacity of

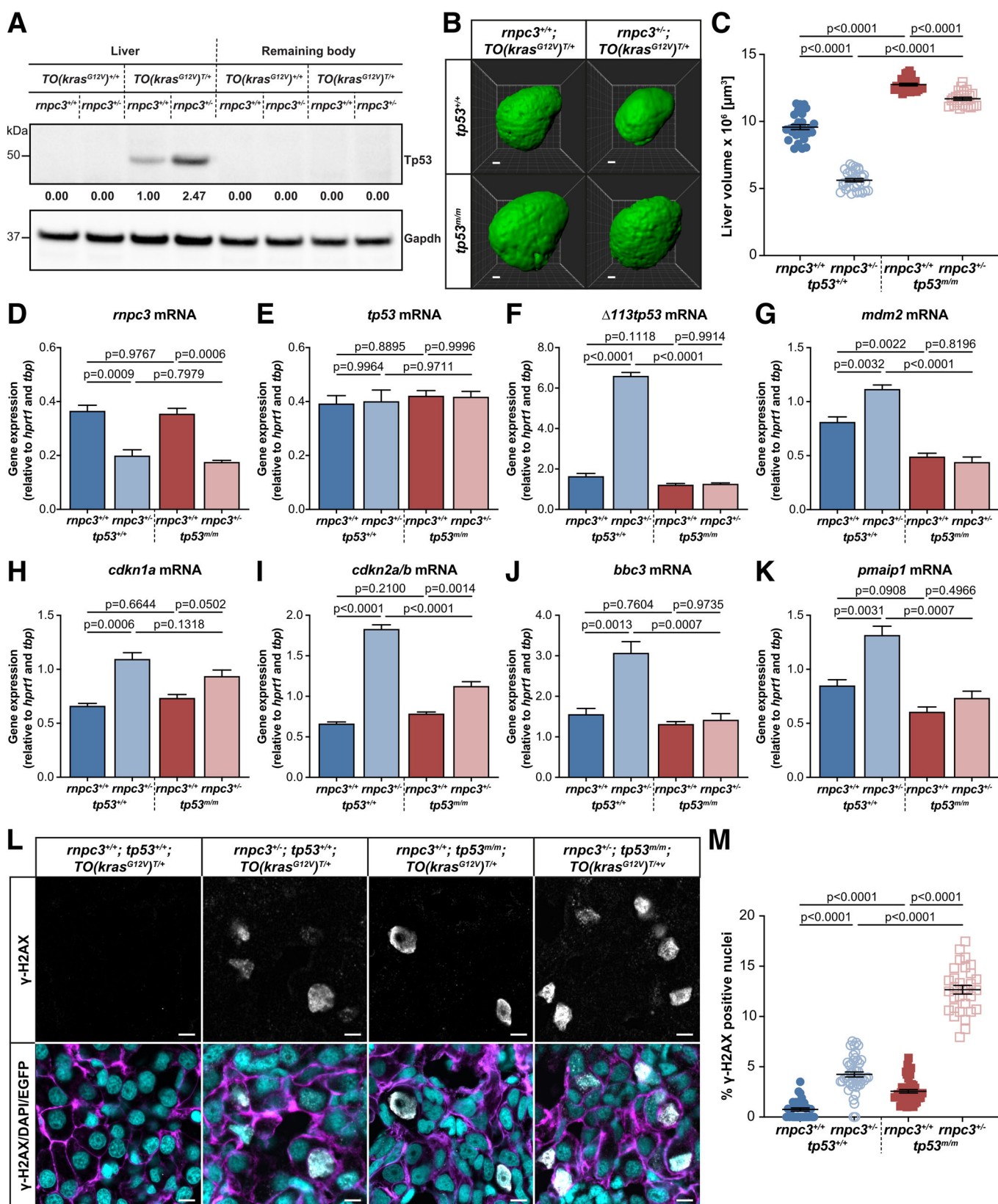

**Figure 2.    *rnpc3* heterozygosity combines with *kras^{G12V}* to activate a Tp53 DNA damage response that restricts tumour burden.**

(A) Tp53 Western blot in lysates of 7 dpf *TO(kras^{G12V})* larvae of the indicated *rnpc3* genotype. Values shown are normalized by reference to the Gapdh loading control and compared with the Tp53 signal in lane 3, which was set at 1. (B) Representative three-dimensional reconstructions of 7 dpf *TO(kras^{G12V})^{T/+}* livers of the indicated *rnpc3* and *tp53* genotypes. Scale bar is 25 μm. (C) Impact of *rnpc3* heterozygosity and homozygous *tp53* mutation on liver volume in 7 dpf *TO(kras^{G12V})^{T/+}* larvae. Data represented as mean ± SEM, $n = 28, 32, 27$ and 24. Significance was tested using a one-way ANOVA with Tukey's multiple comparisons test. Scale bar is 25 μm. (D–K) RT-qPCR analysis of gene expression in 7 dpf *TO(kras^{G12V})^{T/+}* dissected livers of the indicated *rnpc3* and *tp53* genotypes. Data are expressed as mean ± SEM, $n = 3$ biological replicates. (L) Representative Airyscan imaging of liver cryosections of liver from 7 dpf *TO(kras^{G12V})^{T/+}* larvae of the indicated *rnpc3* and *tp53* genotype stained with γ-H2AX antibody (white) marking DNA double-strand breaks, DAPI (cyan) marking DNA, and EGFP-Kras^{G12V} (magenta) marking the cell membrane. Scale bar is 5 μm. (M) Quantification of the percentage of hepatocytes positive for γ-H2AX. Data are expressed as mean ± SEM, $n = 32, 47, 46$ and 30. Significance was tested using a one-way ANOVA with Tukey's multiple comparisons test. Source data are available online for this figure.

heterozygous *rnpc3* to restrain liver overgrowth is heavily dependent on the availability of WT Tp53.

Tp53 is a transcription factor with the potential to employ a large number of target genes to fulfil its tumour suppressor role. We used RT-qPCR to determine which Tp53 target genes were activated in *kras^{G12V}*-expressing livers and whether their expression was dependent on the presence or absence of Tp53 function. First, we found that *tp53* mRNA expression was not altered by *rnpc3* genotype (Fig. 2D,E), indicating that post-transcriptional mechanisms were responsible for the differential levels of Tp53 protein we observed in larvae containing *kras^{G12V}*-expressing livers that were either WT or HET for *rnpc3* (Fig. 2A). We demonstrated upregulated mRNA expression of two canonical Tp53 target genes, *Δ113tp53* and *mdm2* (4.0-fold and 1.4-fold, respectively), two cell cycle arrest genes, *cdkn1a* and *cdkn2a/b* (encoding p21 and p14^{ARF}/ p16^{INK4A}, respectively) and two Bcl2 family genes that promote mitochondrial apoptosis, *pmaip1* and *bbc3* (Fig. 2F–K). On a WT *tp53* background, these six genes were significantly upregulated in the livers of *rnpc3* HETs, compared to *rnpc3* WT livers; however, the enhanced expression of the cell cycle and apoptosis genes was muted in *tp53^{m/m}* larvae, consistent with a requirement for Tp53 in restricting liver overgrowth in *rnpc3* HETS.

As well as regulating cell cycle and apoptosis, responding to DNA damage is a key tumour suppressive function of Tp53. To see if DNA damage was evident in the zebrafish HCC model, we used an antibody to γ-H2AX to look for DNA double-strand (ds) breaks in DAPI-stained cryosections of liver (Fig. 2L). We found that only 0.8% hepatocyte nuclei that were WT for both *rnpc3* and *tp53* contained γ-H2AX foci (Fig. 2M). However, in the absence of Tp53 function, the frequency of γ-H2AX foci increased about threefold (2.5%). In *rnpc3* HETS, the percentage of γ-H2AX positive hepatocytes was fivefold greater (4%) than in *rnpc3* WT hepatocytes, showing that the combination of *kras^{G12V}* and heterozygous *rnpc3* expression increased DNA damage. Moreover, in the absence of WT Tp53, 12.7% of hepatocytes in *rnpc3* HETs were positive for γ-H2AX, equating to a 5-fold increase over their frequency in *rnpc3* WT hepatocytes. Collectively, these data from the zebrafish model of HCC demonstrate that *rnpc3* HET hyperplastic hepatocytes are prone to DNA damage, and that this damage is exacerbated in the absence of WT Tp53 (Fig. 2L,M).

## Heterozygous loss of *Rnpc3* reduces tumour burden in a *Kras^{G12D}*-driven mouse model of lung adenocarcinoma

To test whether similar effects were observed in mouse models of cancer, we took *Rnpc3* heterozygous mice (Doggett et al, 2018) and confirmed they exhibited a 50% decrease in *Rnpc3* mRNA by RT-

qPCR in two organs of interest, lung and stomach (Fig. 3A). We induced lung adenocarcinoma in *Kras^{LSLG12D/+}* *Rnpc3* WT and HET mice (Jackson et al, 2001). In these mice, selective expression of oncogenic *Kras^{G12D}* in lung epithelial cells is achieved upon intranasal delivery of adenoviral Cre recombinase (AdCre) (DuPage et al, 2009). In *Rpnc3* WT, *Kras^{G12D}* mice, histological examination of the lungs 180 d after induction of oncogenic *Kras^{G12D}* expression, revealed the presence of multifocal preneoplastic epithelial lesions, classified as either atypical adenomatous hyperplasia (AAH, arrows) or more advanced papillary adenomas and micro-adenocarcinomas (arrowhead; Fig. 3B), reminiscent of early to intermediate stages of the human disease. These lesions stained robustly for pERK1/2 and the *Rnpc3*-encoded protein, 65K (Fig. 3B'). In contrast, the lungs from *Rpnc3^{+/−}*;*Kras^{G12D}* mice exhibited smaller lesions than lungs from *Rnpc3^{+/+}*;*Kras^{G12D}* mice (compare Fig. 3B,C). Histopathological analysis confirmed that most of these lesions were AAH (arrows, Fig. 3C) with few papillary adenomas/adenocarcinomas. The smaller lesions also expressed pERK and 65K (Fig. 3C'). Quantification of tumour burden revealed a significant decrease in the percentage of total lung area occupied by hyperplastic lesions in *Rnpc3* HET mice compared to *Rnpc3* WT mice (Fig. 3D).

We also crossed the Kras^{G12D} lung cancer model onto *Rnpc3^{lox/lox}* mice (Doggett et al, 2018). In this setting, intranasal administration of AdCre recombines the conditional *Rnpc3^{lox/lox}* and *Kras^{G12D}* alleles at the same time in the same cells. Histological examination of the lungs 90 d after AdCre delivery revealed a significant reduction in the tumour area and the number of AAH lesions in cells harbouring recombined *Rnpc3^{lox/lox}*;*Kras^{G12D}* alleles, compared to cells containing *Rnpc3^{+/+}*;*Kras^{G12D}* alleles (Fig. EV2A–D). These data demonstrate that constitutive and conditional reduction of *Rnpc3* expression restricts the abundance of *Kras^{G12D}* expressing lung epithelial cells during lung tumourigenesis.

## Heterozygous loss of *Rnpc3* reduces tumour burden in a STAT3-driven model of gastric adenocarcinoma

To explore whether heterozygous *Rnpc3* reduces tumour burden more broadly, we employed a mouse model of gastric cancer in which adenomas develop due to a tyrosine (Y) to phenylalanine (F) mutation at codon 757 of the cytokine receptor GP130, leading to persistent activation of STAT3. In *Gp130^{F/F}* mice, the development of adenomas in the glandular epithelium of the stomach is spontaneous and 100% penetrant by 100 d of age (Jenkins et al, 2005; Tebbutt et al, 2002). Mice with a *Rnpc3^{+/−}*;*Gp130^{F/F}* genotype exhibited a significant reduction in total adenoma weight compared to *Rnpc3^{+/+}*;*Gp130^{F/F}* mice at both 100 and 180 d, with a more marked reduction in the weight of adenomas harvested from the proximal glandular epithelium (corpus; arrowheads) compared

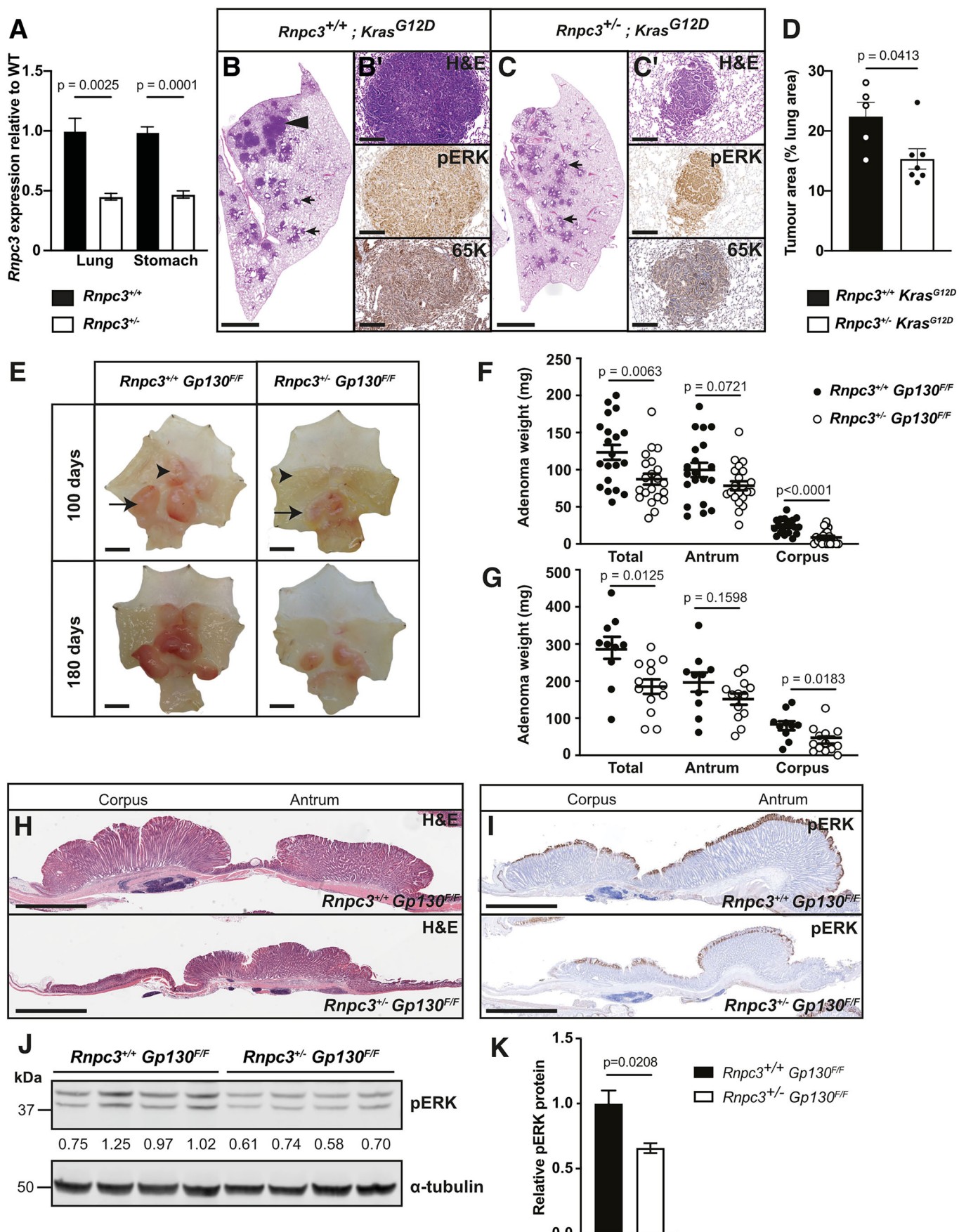

◀ **Figure 3.** *Rnpc3* **heterozygosity reduces tumour burden in a mouse model of lung adenocarcinoma and gastric adenoma.**

(A) RT-qPCR analysis of *Rnpc3* mRNA extracted from 160-day-old WT and *Rnpc3* heterozygous mouse stomach and lung tissue ($n = 4$, 2 males and 2 females). Results are expressed as mean ± SEM. Significance was assessed using multiple unpaired *t* tests. (B, C) Representative hematoxylin and eosin-stained lung sections 180 days after intranasal administration of adenoviral Cre recombinase (AdCre). Arrows and arrowhead indicate foci of atypical adenomatous hyperplasia (AAH) and adenoma, respectively. These lesions are pERK and 65 K positive. Scale bar in (B, C) is 2 mm. Scale bar in (B′, C′) is 200 μm. (D) Quantification of hyperplasia in sections of lung, expressed as a percentage of total lung area. Results are expressed as mean ± SEM, $n = 5$ or 7 per genotype. Significance was assessed using a Student's *t* test. (E) Representative adenomas in the corpus (arrowhead) and antral (arrow) regions of the glandular stomach of mice. Scale bar is 5 mm. (F) Total adenoma weight at 100 d of age, $n = 20$ or 21 per genotype. Data are expressed as mean ± SEM. (G) Total adenoma weight at 180 d of age, $n = 10$ or 14 per genotype. Data are expressed as mean ± SEM. Significance was assessed with a Student's *t* test with Welch's correction. (H) Histological sections of the glandular stomach stained with H&E at 100 d. (I) Immunocytochemical localization of pERK1/2 indicates active MAPK signalling at the luminal surface of adenomas in 100 d old *Gp130^{F/F}* mice. Scale bar in (H, I) is 2 mm. (J) Western blot analysis of pERK1/2 proteins in antral adenomas from four individual mice/genotype at 100 d. Values shown are normalized by reference to the α-tubulin loading control and relative to *Rnpc3^{+/+}*; *Gp130^{F/F}* samples. (K) Quantification of pERK1/2 protein abundance shown in (J). Data are expressed as mean ± SEM, $n = 4$. Significance was assessed with a Student's *t* test. Source data are available online for this figure.

to the distal glandular epithelium (antrum; arrows, Fig. 3E–G). Indeed, approximately 40% of *Rnpc3^{+/−}*;*Gp130^{F/F}* mice displayed a complete absence of corpus adenomas at 100 d (Fig. 3E,F). We also observed pERK1/2 staining at the luminal edge of adenomas in both the corpus and antral regions (Fig. 3H,I), indicating that the pro-proliferative MAPK pathway was active at the periphery of the growing adenomas. However, there was a 35% decrease in the abundance of pERK1/2 protein in the adenomas of *Rnpc3* HETs compared to *Rnpc3* WT adenomas (Fig. 3J,K), indicating attenuation of the MAPK pathway in cells exhibiting heterozygous expression of *RNPC3*.

To determine whether inducing loss of *Rnpc3* expression in established gastric adenomas also reduced growth (compared to constitutive heterozygosity), we crossed mice carrying conditional *Rnpc3^{lox}* alleles with *Gp130^{F/F}* mice also carrying a *Trefoil factor 1 (Tff1)*-CreERT2 BAC transgene that confers tamoxifen (TMX)-inducible, gastric epithelium-selective, Cre recombinase activity (Thiem et al, 2016a). We used oral gavage to deliver TMX to mice on days 56 and 57, well after gastric adenoma formation had initiated (Putoczki et al, 2013) (Fig. EV2E). We found that recombination of *Rnpc3^{lox}* alleles produced a reduction in total adenoma burden, again with a more marked effect on the corpus region than the antrum, compared to TMX-treated mice not carrying the *Tff1*-CreERT2 BAC transgene (Fig. EV2F,G).

Antral adenomas harvested from TMX-treated *Tff1*-CreERT2;*Rnpc3^{lox/lox}*;*Gp130^{F/F}* mice contained 50% less *Rnpc3* mRNA than antral adenomas from TMX-treated *Rnpc3^{lox/lox}*;*Gp130^{F/F}* (no Cre-transgene) controls (Fig. EV2H). PCR analysis of individual adenomas from TMX-treated *Tff1*-CreERT2;*Rnpc3^{lox/lox}*;*Gp130^{F/F}* mice, identified cells with either a *Rnpc3^{lox/lox}* or *Rnpc3^{lox/Δ}* genotype, but no cells with a *Rnpc3^{Δ/Δ}* genotype, where *Δ* represents a deleted (null) allele (Fig. EV2I). This indicates that cells evading recombination of both *lox* alleles survived, while *Rnpc3^{Δ/Δ}* cells died. We surmise from this that the reduction in adenoma burden observed in TMX-treated *Tff1*-CreERT2;*Rnpc3^{lox/lox}*;*Gp130^{F/F}* mice is likely due to the combined effect of two induced genotypes: the *Rnpc3^{lox/Δ}* (heterozygous) genotype causing reduced proliferation and the *Rnpc3^{Δ/Δ}* (homozygous null) genotype causing cell death. These data echo our previous experiments indicating that a full complement of *Rnpc3* expression is required to support the full growth potential of cancer cells.

## Disruption of the *Rnpc3* locus in AML cells causes impaired minor splicing and prolonged survival of mice

Having shown that *Rnpc3* heterozygosity limits the growth of liver, lung and gastric hyperplasia/adenomas, we investigated whether hyperproliferative blood cancers were sensitive to *Rnpc3* expression as well. We used a tractable model of blood cancer driven by a retroviral construct encoding the human MLL-ENL leukemogenic oncoprotein fused to GFP (Schoch et al, 2003; Zuber et al, 2009) (Fig. 4A). Foetal liver cells harvested from mice harbouring conditional *Rnpc3^{lox/lox}* alleles and a *UBC-CreERT2* transgene (Doggett et al, 2018) were transformed with retroviral MLL-ENL-GFP to generate a population of cells that was used to reconstitute the hematopoietic system of irradiated WT mice (Fig. 4A). As each mouse reached the ethical endpoint of the experiment (disease latency = 42–81 d), AML cells were harvested from the bone marrow and spleen and transplanted into unirradiated WT hosts where they generated a more aggressive AML with a shorter latency (17–55 d; Fig. 4A). These AML cells were harvested from the bone marrow and were either treated with 4-hydroxy tamoxifen (4-OHT) in vitro to assess the impact of recombining the *Rnpc3^{lox/lox}* locus on the efficiency of minor splicing and cell viability in culture, or transplanted for the third time into recipient WT mice and treated with TMX 13 and 14 d later to determine the impact of recombining the *Rnpc3^{lox/lox}* locus on these aggressive AML cells in vivo.

*Rnpc3^{lox}* alleles in AML cells carrying the *UBC-CreERT2* transgene underwent almost complete recombination after treatment with 4-OHT in vitro (Fig. 4B). This resulted in a 95% loss of *Rnpc3* mRNA expression at 72 h (Fig. 4C) and a marked increase in apoptosis of AnnexinV^+ and PI^+ cells at 96 and 120 h, as shown by flow cytometry analysis (Fig. EV3A,B). The impact of loss of *Rnpc3* expression on the efficiency of minor splicing in these secondary AML cells in vitro was assessed by RT-qPCR using primers designed to amplify retained minor introns. We analysed transcripts relevant to DNA repair (*Parp1*), MAPK signalling (*Braf*, *Mapk1*, *Rasgrp2*) and cell cycle progression (*Ccnt2*, *Ccnk*, *Cdk5*, *E2f1*, *Cdc45*) and *Vps16*. In all cases, transcripts from AML cells with a recombined *Rnpc3* locus exhibited elevated minor intron retention compared to cells in which the *Rnpc3* locus remained intact (Fig. 4D).

Next, we recombined the *Rnpc3* locus in vivo. In the absence of TMX treatment, recipient mice harbouring AML cells with the *Rnpc3^{lox/lox}*;*UBC-CreERT2* genotype exhibited a median survival of 19 d. However, recombining the *Rnpc3^{lox/lox}* locus at 13 and 14 d with TMX extended the median survival to 29 d (52% increase, Fig. 4E). Similarly, when WT recipient mice were transplanted with *Rnpc3^{lox/−}*;*UBC-CreERT2* AML cells and treated with TMX, the median survival of these mice was extended by 55% (34 d compared

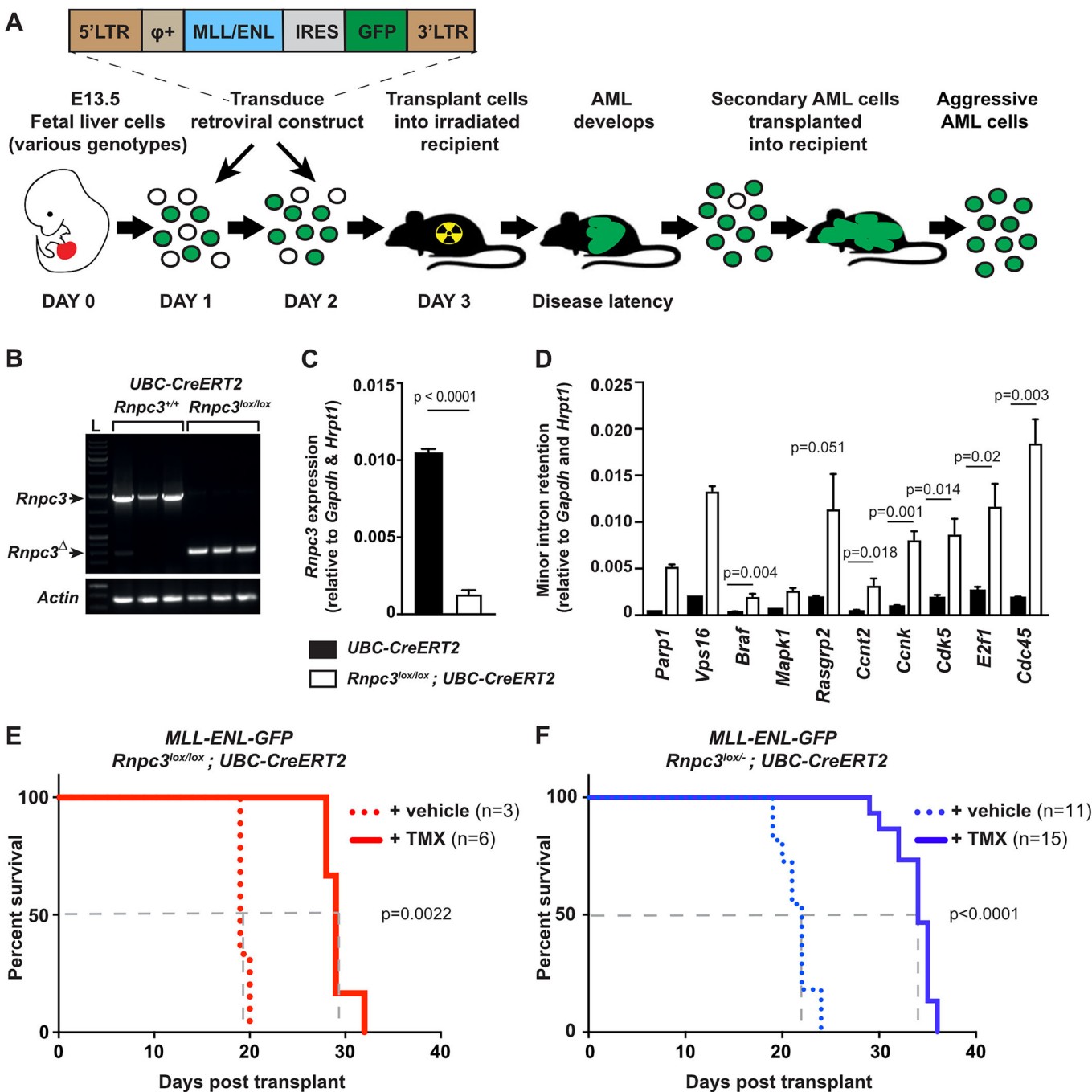

**Figure 4. Disruption of the *Rnpc3* locus in AML cells causes impaired minor splicing and prolonged survival of mice.**

(A) Schematic diagram depicting the AML experimental workflow. (B) PCR for *Rnpc3* recombination on AML cells treated in vitro for 72 h with 4-OHT. L Ladder. (C) RT-qPCR of *Rnpc3* transcripts on AML cells treated in vitro for 72 h with 4-OHT. Data are expressed as mean ± SEM, $n = 3$. (D) RT-qPCR of amplicons containing retained minor introns. Results are expressed as the mean of three independent secondary AML cell lines per genotype ± SEM, significance was assessed using a two-tailed, unpaired Students' *t* test, except for *Parp1, Vps16* and *Mapk1* where $n = 2$. (E) Kaplan–Meier plot of female mice harbouring tertiary transplants of *Rnpc3*<sup>lox/lox</sup>;*UBC-CreERT2* AML cells, treated with TMX or vehicle 13 and 14 days later ($n = 6$ and 3, respectively, one experiment). (F) Kaplan–Meier plot of female mice harbouring tertiary transplants of *Rnpc3*<sup>lox/−</sup>;*UBC-CreERT2* AML cells, treated with TMX or vehicle 13 and 14 days later. ($n = 15$ and 11, respectively, two independent experiments combined). Significance was assessed with a Mantel–Cox test. Source data are available online for this figure.

to 22 d; Fig. 4F). To minimise the possibility that treatment of mice with TMX had a confounding influence on disease latency, we performed additional controls (Fig. EV3C). Ultimately, all mice developed splenomegaly (Fig. EV3D) and succumbed to AML with an immature myeloid (Mac-1⁺/Gr-1⁻) phenotype (Fig. EV3E). Genomic analysis of the TMX-treated tertiary transplanted AML cells revealed the presence of both un-recombined ($Rnpc3^{lox}$) and deleted ($Rnpc3^{\Delta}$) alleles, indicating that TMX treatment did not achieve 100% recombination of the $Rnpc3$ locus in AML cells in vivo (Fig. EV3F). Only the un-recombined $Rnpc3^{lox}$ allele was detected in $Rnpc3^{lox/-}$;$UBC$-$CreERT2$ AML cells derived from TMX-treated mice, suggesting that AML cells that acquired a $Rnpc3^{\Delta/-}$ genotype died. Together, these data show that disrupting $Rnpc3$ expression in highly proliferative AML cells impairs the efficiency of minor splicing, leading to death of AML cells and improved survival of recipient mice.

## Knockdown of *RNPC3* impairs minor splicing and slows the growth of human A549 cells

Next, we carried out in vitro studies in a human lung adenocarcinoma-derived cell line, A549, with the aim of providing a mechanistic explanation of how reduced expression of *RNPC3* causes a decrease in tumour burden in the various in vivo cancer models. We chose A549 cells because they carry a $KRAS^{G12S}$ mutation, similar to the introduced $kras^{G12V}$ and $KRAS^{G12D}$ mutations harboured by the genetically-engineered zebrafish and mouse cancer models, respectively. We transfected A549 cells with two siRNAs independently targeted to *RNPC3* (siRNAs #18 and #19) to achieve levels of depletion between 50–100%. The control siRNA for these experiments was a non-targeted (NT) sequence that does not induce degradation of cellular transcripts. To determine whether knocking-down another unique component of the minor spliceosome also played a role in minor splicing, we additionally knocked down *PDCD7* expression in A549 cells (two independent siRNAs, #5 and #6).

We transfected A549 cells with siRNAs and after 72 h measured *RNPC3* mRNA expression levels by RT-qPCR (Fig. 5A). Over this time-course, the knockdown in *RNPC3* expression was approximately 70% leading to the aberrant splicing of a small set of well-characterised 'model' minor introns (Fig. EV4C), either through intron retention (IR) as seen for *VSP16* and *TTC23* or alternative splicing (AS) as seen for *VSP35*, *NCBP2* and *E2F3*. In response to these aberrant splicing events, we observed a 30% reduction in the percentage confluency of A549 cells (Fig. 5B). We obtained very similar results with *PDCD7* knockdown in A549 cells (Fig. EV4).

## Transcriptome analysis of A549 cells reveals widespread disruption of minor intron splicing in response to *RNPC3* knockdown

Next, we inspected the integrity of minor intron splicing transcriptome-wide using RNAseq. By reference to the Minor Intron Database (MiDB; https//midb.pnb.uconn.edu), we found that of the 755 minor introns annotated in the human MiDB, 389 were detected in A549 cells. Of these 389 detected MIGs, 158 (40.6%) were aberrantly spliced in response to *RNPC3* knockdown and the other 231 (59.4%) were spliced normally. Of the 158 affected MIGs, 134 (34.4%) exhibited significantly elevated IR, as

seen in *MAPK12* and *ATG3*, 14 (3.6%) exhibited significantly elevated AS, as seen in *NAA60* and *VPS35* and 10 (2.6%) exhibited elevated levels of both IR and AS (Figs. 5C–E and EV5A). This set of 158 aberrantly expressed MIGs was enriched for several GO terms that are relevant to the growth and proliferation of cancer cells, including, cellular response to stress, snRNA processing, MAPK activity and nucleotide excision repair (Fig. EV5B; Dataset EV1). In multiple cases, major (U2) introns that flanked retained minor introns also exhibited elevated intron retention (mis-splicing index_{ret}) in response to *RNPC3* knockdown (Fig. EV5C), consistent with previous observations that exon-bridging interactions between the components of the major and minor spliceosomes are impaired in response to *RNPC3* loss (Olthof et al, 2021).

## *RNPC3* knockdown in A549 cells selectively disrupts minor intron splicing over major intron splicing

To establish whether aberrant splicing in A549 cells in response to *RNPC3* knockdown was selective for minor introns, we re-analysed our A549 RNAseq dataset using an algorithm known as IRFinder (Middleton et al, 2017). This provided a transcriptome-wide assessment of IR across all introns (both major and minor). We then employed differential intron retention analysis to identify 245 high-confidence introns that were differentially retained between si*RNPC3* and siNT transfected A549 cells. Of these, 185 were major introns and 60 were minor introns (Fig. EV5D,E; Dataset EV2). Of the 599 minor introns detected, 60 (10%) exhibited increased IR and not one displayed less IR than in siNT-treated cells (Fig. EV5D,E). In contrast, of the 179,463 major introns detected, only 148 (0.08%) exhibited increased IR in *RNPC3* knockdown cells and 0.02% exhibited reduced IR (Fig. EV5D). This equates to an approximately 125% enrichment of minor intron retention over major intron retention (10% versus 0.08%) in response to *RNPC3* knockdown. In contrast, the median retention level of all major introns increased by only 5% in response to *RNPC3* knockdown (Fig. EV5F). Of the retained major introns, one-third was found in transcripts containing one or more minor introns and almost two-thirds of these occupied positions flanking minor introns, independent of whether the minor intron was retained or not. This is consistent with the data we generated above when we interrogated the same RNA-seq data with the boutique, minor intron-specific algorithm (Fig. EV5C,F).

However, we did notice that in terms of precise numbers, the output from the two algorithms was different, in that the boutique algorithm identified 158 minor introns that were affected by IR and/or AS out of a total of 389 minor introns detected (40%), whereas IRFinder identified 60 retained introns out of 599 minor introns detected (10%). However, both sets of analyses were in agreement that *RNPC3* knockdown exerted a selective effect on the splicing of minor introns over major introns.

To determine the impact of the dysregulation of minor splicing on gene expression globally, we analysed the abundance of all the differentially expressed genes (DEGs) in our A549 RNAseq dataset. Of the 24,644 transcripts detected, 583 (2.4%) were transcribed from MIGs (Fig. 5F,G). Of these 583 MIGs, 138 (24%) were downregulated and 32 (5.5%) were upregulated (Fig. 5G), leaving 413 (70%) that were not differentially expressed. Of the differentially expressed MIGs, *RNPC3* knockdown resulted in an

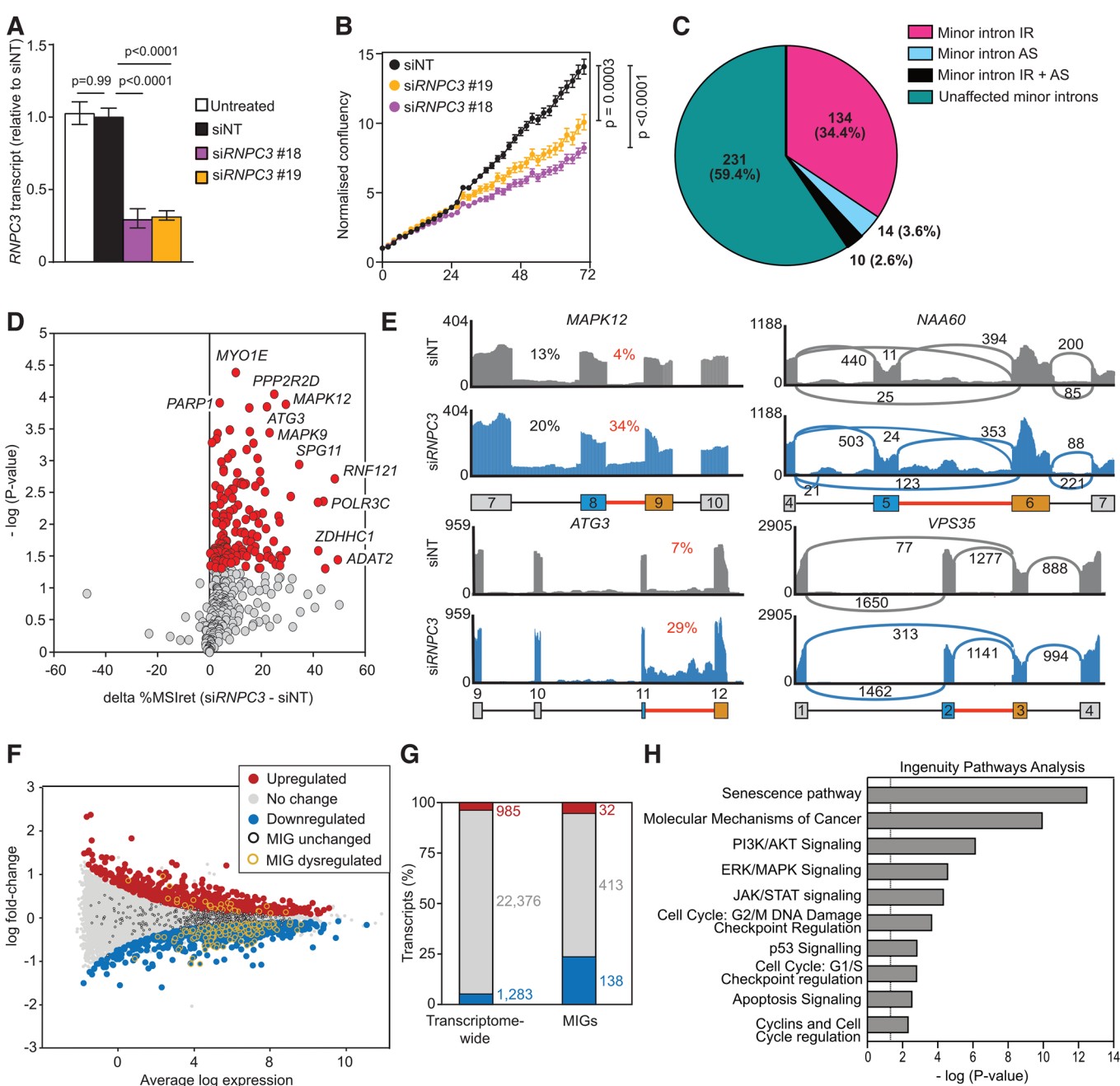

**Figure 5. *RNPC3* knockdown impairs the growth of A549 cells and causes defects in the splicing of minor introns resulting in differential expression of genes enriched in cancer related pathways.**

(A) RT-qPCR analysis of *RNPC3* transcripts in A549 cells treated with si*RNPC3* #18 and #19 for 72 h. Data are represented as mean ± SEM ($n = 4$ biological replicates). (B) Quantification of A549 cell growth over 72 h treatment with non-targeting (NT) or 2 independent *RNPC3* siRNAs. Data are represented as mean ± SEM ($n = 3$, 25 images per well, every hour). Significance was assessed by two-way ANOVA with Dunnett's multiple comparisons test. (C) Pie chart displaying percentage distribution of aberrant minor intron splicing events in si*RNPC3*-treated cells. (D) Volcano plot of MIG transcripts exhibiting significant minor intron retention (solid red circles) in si*RNPC3*-treated cells. Significance was assessed by Welch's *t* test. For an intron to be called as significantly retained in si*RNPC3* samples, we required the following, FDR ≦ 0.05, 100% coverage of the intron in si*RNPC3* samples, ≥10% IRratio in si*RNPC3* and ≥ 5% IRratio over siNT samples. Likewise, for an intron to be significantly retained in siNT samples, FDR ≦ 0.05, 100% coverage across the intron in siNT samples, ≥ 10% IRratio in siNT cells and ≥ 5% IRratio over si*RNPC3* samples ($n = 4$ replicates per genotype). (E) Example RNAseq read coverage plots showing *RNPC3* knockdown increases minor intron retention in *MAPK12* and *ATG3* and produces alternative splicing in *NAA60* and *VPS35*. (F) Scatter plot of differential gene expression between si*RNPC3* and siNT-treated A549 cells. Of the 24,061 expressed transcripts, 2268 are DEGs (1286 downregulated shown in blue, 985 upregulated shown in red. The dysregulated MIGs are circled in yellow). (G) Differential gene expression analysis of MIGs compared to all transcripts. (H) Selection of significantly enriched canonical pathways identified by IPA analysis of all affected genes (DEGs plus IR and AS affected MIGs). -log (*P*) values > 1.3 (vertical line) signify significant enrichment. Significance was assessed by Fisher exact test. Source data are available online for this figure.

approximately 4.4-fold greater propensity for downregulation of expression compared to upregulation (Fig. 5F,G).

In total, >40% of expressed MIGs were affected by *RNPC3* knockdown, due to a combination of differential gene expression, intron retention and/or aberrant splicing (Fig. EV6A,B). We took this group of affected genes (DE-MIGs + IR MIGs + AS MIGs) and used Ingenuity Pathway Analysis (IPA) to identify 'Canonical pathways' significantly enriched with our set of affected genes. This revealed multiple terms relevant to this study such as Senescence pathway, Cell cycle DNA damage checkpoints, Tp53 signalling and the ERK/MAPK, PI3K/AKT and JAK/STAT signalling pathways that drive cell proliferation and survival (Fig. 5H; Dataset EV1).

### Disrupted minor splicing in A549 cells causes accumulation of aberrant MIG transcripts, DNA damage, TP53 activation, cell cycle arrest and senescence

Our data show that a 70% depletion in the levels of mRNA transcribed from the *RNPC3* locus in A549 cells results in the accumulation of aberrant MIG transcripts, changes in the expression of genes typically enriched in cancer cells and impaired growth and survival (Figs. 5, EV5, and EV6). To gain insights into the cellular and molecular mechanisms underlying these events, we disrupted minor splicing in A549 cells using antisense oligonucleotides (ASO) designed to target U12 snRNA, which is another unique and indispensable component of the minor spliceosome.

We conducted RT-PCR analysis of RNA harvested from A549 cells treated for 72 h with a U12 ASO and a previously published non-targeted ASO (NT ASO) as control (Younis et al, 2013). We examined the impact of the U12 ASO on the expression of several well-characterised MIGs (Fig. EV7A) and MIGs with roles in cell cycle and transcription regulation (E2F1-3), initiation of chromosomal DNA replication (CDC45) and the resolution of replicative stress (TRAIP). We found that U12 ASO treatment disrupted the splicing of the minor introns in all pre-mRNA transcripts, either through IR (E2F1, 2 and TRAIP), AS (E2F3) and decreased MIG expression, likely due to NMD of aberrantly-processed transcripts, as seen for CDC45 (Fig. 6A). To determine whether these perturbations affected the integrity of the cells, we used flow cytometry and immunocytochemistry with fluorescent antibodies to identify double-stranded DNA breaks. With FACS, we observed a five-fold increase in the number of phospho-histone γH2AX (Ser139)-positive A549 cancer cells in response to the U12 ASO (Figs. 6B,C and EV7B) and a marked increase in phospho-53BP1 (Ser1778) (Fig. 6D), which like γH2AX, is phosphorylated upon DNA damage, triggering their recruitment to double-strand breaks. These data indicate that disrupted MIG splicing in A549 lung cancer cells leads to a marked increase in DNA damage.

As shown in Fig. 2, in the context of the zebrafish model of HCC, the stabilisation and accumulation of TP53 in response to DNA damage unleashes a transcriptional program designed to reduce cell growth and proliferation through pathways that can induce apoptosis, cell cycle arrest and senescence. To see if this is reiterated in A549 cells, we used Western blot analysis to determine whether the levels of TP53 were changed in response to impaired minor splicing. This revealed robust TP53 signals in the nuclear compartment of A549 cells treated with U12 ASO and very faint signals in the cytoplasm (Fig. 6E). Only very weak signals were obtained from A549 cells that had been treated with the NT ASO,

irrespective of cellular compartment. These data suggest that aberrant splicing of MIGs leads to DNA damage and the accumulation of TP53 (Fig. 6E). We also found that U12 ASO-treated cells showed a three-fold increase in the number of senescent cells compared to treatment with the NT ASO (Fig. 6F,G). There was also a doubling of the percentage of cells in G2, concomitant with a small but significant decrease in the percentage of cells in S phase (Fig. 6H,I). These perturbations in cell cycle phases were reminiscent of separate experiments we performed with the Incucyte Live Cell Analysis system, where we saw a marked stalling in the %age confluency of A549 cells from 24 h onwards over the course of a 72 h treatment with both 5 and 10 nM U12 ASO (Fig. 6J).

## Discussion

Cancer of the digestive organs remains a leading cause of mortality worldwide, necessitating the discovery of novel therapeutic targets. In our study, we utilised a forward genetic screen in zebrafish to identify *rnpc3*, a gene crucial for rapid digestive organ growth during development. Remarkably, heterozygous *rnpc3* reduced liver overgrowth in a zebrafish model of hepatocellular carcinoma (HCC), underscoring the potential of zebrafish genetic screens in identifying new cancer therapy targets. To extend our findings to mammals, we introduced *Rnpc3* heterozygosity into a variety of mouse cancer models and again observed significantly reduced tumour growth. These results establish *Rnpc3* as a relevant therapeutic target across species and cancer types. To explore the molecular mechanisms underlying these observations, we turned to the human lung cancer cell line A549, which is known for its functional TP53 pathway. This model allowed us to study *TP53*-mediated responses to DNA damage and cell cycle regulation, providing deeper insights into the therapeutic potential of targeting *RNPC3*. However, while we were successful in demonstrating the wide applicability of our results across a variety of in vivo and in vitro cancer models, there are also disadvantages to our multifaceted approach. For example, if we had focused more strongly on a subset of our in vivo cancer models, we may have been successful in demonstrating a formal link between *Rnpc3* heterozygosity and aberrant minor splicing, which is absent from our current study.

Although the existence of minor splicing has been recognised for almost 30 years, a broad appreciation of its biological significance is lacking. This is surprising, considering that the process is indispensable for vertebrate development and is critical for the correct expression of hundreds of genes, many of which play central roles in the growth, division and survival of cells. Only recently has a consensus emerged that in adult animals, the process is required to support the behaviour of continuously cycling cells, including stem cells in self-renewing tissues (Baumgartner et al, 2018; Doggett et al, 2018). In this study, our aim was to examine the impact of impairing minor splicing efficiency in the context of cancer. To do this, we crossed the zebrafish and mouse models of *Rnpc3* deficiency (Markmiller et al, 2014; Doggett et al, 2018) onto a variety of constitutive and inducible tumour-prone backgrounds. These experiments showed that heterozygous expression of *Rnpc3* is sufficient to markedly restrict the growth and proliferation of cancers driven by a variety of pro-proliferative, pro-survival

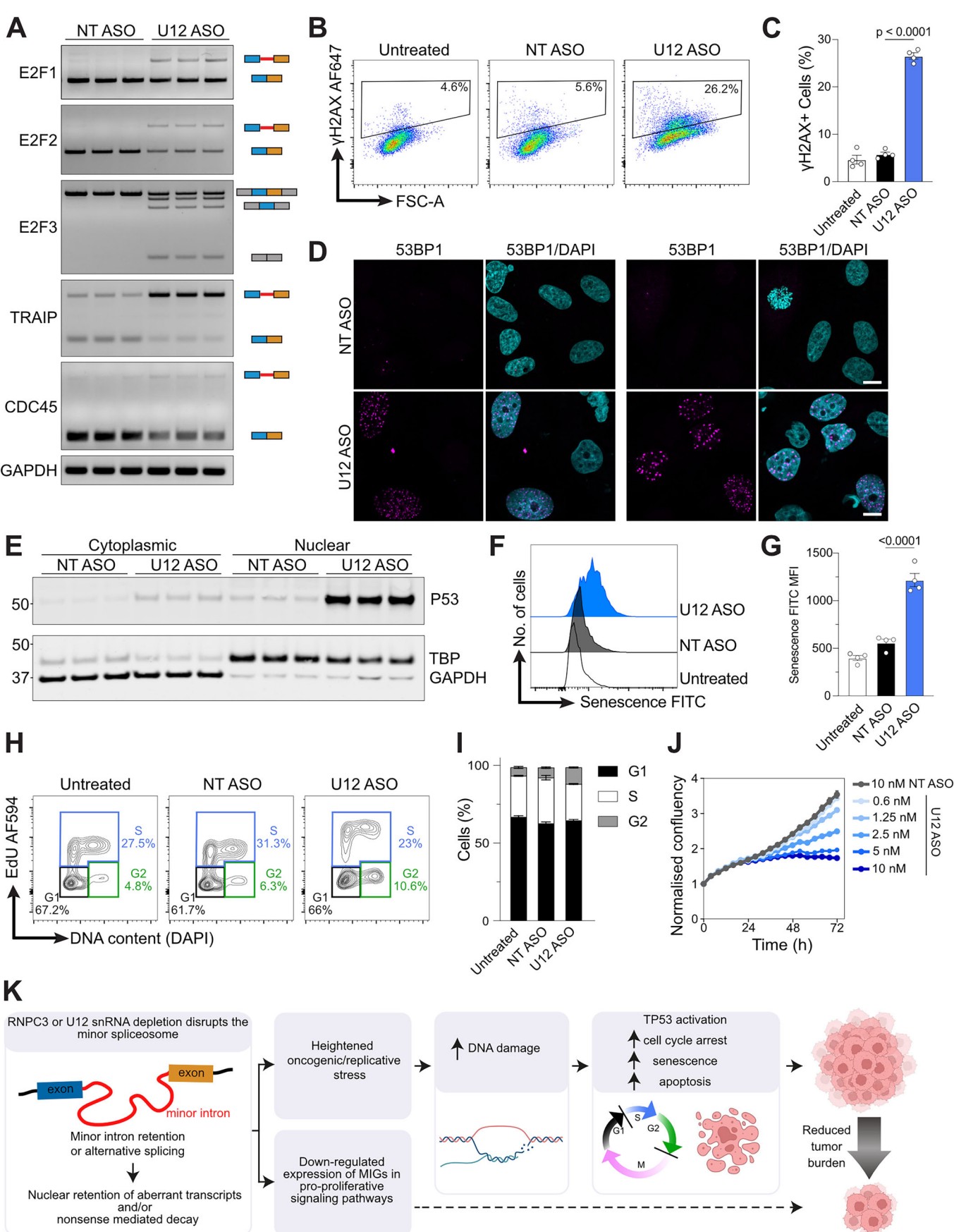

◄ **Figure 6. Minor class splicing knockdown induces DNA damage and cell cycle arrest in human cancer cell lines.**

(A) RT-PCR analysis of cell cycle related MIG splicing changes in A549 cells after 72 h treatment with 10 nM ASO. Schematic depictions of the obtained amplicons are shown on the right with the minor intron in red and the upstream and downstream exons coloured blue and orange, respectively. Exons not separated by a minor intron are grey. (B) Representative FACS plots of γH2AX expression from A549 cells after 72 h treatment with 10 nM ASO. (C) Quantification of γH2AX FACS analysis. Data are represented by the mean ± SEM ($n = 4$). Significance was assessed by one-way ANOVA with Tukey's multiple comparison test. (D) Two representative images of 53BP1 staining in A549 cells 72 h after 10 nM ASO treatment. Scale bar is 10 μm. (E) P53 immunoblot of nuclear and cytoplasmic fractions from A549 cells exposed to 10 nM ASO for 72 h ($n = 3$ biological replicates/condition). (F) Flow cytometry histograms of FITC senescence probe in A549 cells 72 h after 10 nM ASO treatment. (G) Quantification of senescence FACS analysis. MFI = median fluorescence intensity. Data are represented by mean ± SEM ($n = 4$). Significance was assessed by one-way ANOVA with Tukey's multiple comparison test. (H) Representative FACS plots of cell cycle analysis from A549 cells after 72 h treatment with 10 nM ASO. Proliferating cells stained for incorporated EdU against total DNA content measured by DAPI. (I) Quantification of FACS cell cycle analysis. Data are represented by mean ± SEM ($n = 4$). Significance was assessed by one-way ANOVA with Tukey's multiple comparison test. U12 ASO-treated samples were significantly different to NT ASO samples at two cell cycle stages: S $P = 0.0009$, G2 $P < 0.0001$. (J) Quantification of A549 cell growth over 72 h treatment across a titration of U12 ASO (0.625 nM–10 nM) compared to 10 nM non-targeted (NT) ASO. (K) Schematic diagram depicting the sequence of molecular and cellular events linking minor splicing disruption with reduced tumour burden. Data are represented as mean ± SEM ($n = 4$, 16 images per well, every hour). Source data are available online for this figure.

pathways (KRAS/MAPK and STAT3), without any impact on healthy tissues.

To understand the cell and molecular mechanism(s) underlying our in vivo observations, we looked at the impact of impaired minor splicing on A549 cells, derived from a human lung adenocarcinoma. Collectively our data show that impaired splicing of MIGs in oncogene-fuelled cells leads to the generation of pre-mRNA transcripts containing retained minor introns and alternatively spliced RNAs. We hypothesise that there are at least two mechanisms that contribute to a reduction in cancer growth. In the first, we think that the accumulation of aberrantly spliced MIG transcripts in the nucleus is 'blind' to their cancer relevance, meaning that any aberrantly spliced MIG transcripts can contribute to steric hindrance and collisions between the DNA replication, transcription and splicing machineries as they compete for the genome in rapidly proliferating cells (Bermejo et al, 2012; Kotsantis et al, 2018), producing replicative stress and DNA damage, irrespective of their functional role in the cell. The increase in DNA damage activates the tumour suppressive properties of Tp53, triggering a transcriptional program that increases cell cycle arrest, senescence and apoptosis, and a decrease in tumour burden.

In the second, we think there may be scenarios where aberrant expression of individual MIGs contribute directly to disabling cancer signalling pathways, including MIGs that encode the majority of MAPK family members and the E2F3 family of cell cycle genes. Since the genes encoding BRAF, RAF1 and 11 out of 14 MAPKs contain minor introns, we thought that the pro-proliferative activity of the KRAS/RAF/MAPK pathway may rely heavily on efficient minor splicing, as indicated by the dashed arrow in Fig. 6K. However, our RNA-seq results do not provide definitive evidence that mis-splicing of one or more BRAF/RAF1/MAPK transcripts is responsible for the decreased growth and survival of A549 cells. Instead, our transcriptome-wide gene expression data suggest strongly that loss of minor splicing integrity is a cancer cell vulnerability that triggers changes in the expression of a different set of cancer-relevant genes, including those involved in DNA damage repair, replication and transcription. Our schematic diagram (Fig. 6K) encapsulates both these possibilities.

Working with A549 cells helped us to circumvent the problem of not being able to vary the degree of minor splicing inhibition in the in vivo tumour models where gene dosing can only be reduced genetically by 50% or 100% (or not at all). This makes extrapolation of our findings to a clinical situation difficult, since we predict that

a therapeutically effective dose of *RNPC3*/65K inhibition would likely lie somewhere within the 50–100% range. To pursue this notion further, we inhibited the expression of *RNPC3*/65K beyond the equivalent of a 50% reduction in gene dosage by transfecting A549 cells with siRNAs designed to target *RNPC3*. We showed that a 70% reduction in *RNPC3* mRNA expression impaired the integrity of minor splicing and produced a concomitant reduction in cancer cell growth. Similar results were obtained with ASOs, where we demonstrated a dose-dependent inhibition of A549 cell growth. This suggests that, in the future, small molecule inhibitors of minor spliceosome function, with favourable pharmacokinetic properties, could be developed and applied in a titratable fashion against a broad spectrum of hitherto difficult-to-treat solid and haematological malignancies.

The inhibition of splicing for the purpose of cancer therapy is not a new idea. It first attracted attention when it was discovered that the genomes of certain cancers frequently carry point mutations in the general splicing factor genes, *SF3B1*, *U2AF1*, and *SRSF2*. It was particularly conspicuous that 20–30% of patients with myelodysplastic syndrome, a disease which frequently progresses to chronic lymphocytic leukemia, chronic myelomonocytic leukemia and AML (Madan et al, 2015; Yoshida et al, 2011; Liu et al, 2020) harbour mutations in *SF3B1*. Cancer cells bearing these mutations, which usually cause a change in function (rather than a loss of function) of the encoded proteins, are extremely sensitive to further genetic or chemical disruption of splicing (Lee et al, 2016; Obeng et al, 2016). For example, human myeloid and leukemia cells engineered to carry heterozygous mutations in both *SF3B1* and *SRSF2* are not viable (Lee et al, 2018).

Importantly, we found no evidence that MIGs have the capacity to play a direct role as cancer-causing oncogenes. Rather, they contribute to synthetic lethal interactions with oncogenes which limit tumour burden. This molecular mechanism is highly amenable to therapeutic intervention and has proved to be extremely successful in the clinic. The most noteworthy example is the synthetic lethal interaction between *BRAC1/2* mutations and inhibition of PARP1 protein expression by PARP1 inhibitors in breast and ovarian cancers. Interestingly, PARP1 is itself a MIG, raising the possibility that minor splicing inhibitors could also interact indirectly with BRAC1/2 mutations to reduce tumour burden.

Another potential advantage of targeting a protein like 65K is the fact that it is an essential component of a multiprotein complex. This may mean that drugs targeted to it are less vulnerable to the types of resistance mechanisms that thwart drugs targeted to

**Table 1.  Primer sequences.**

|  | Gene | Forward | Reverse |
|---|---|---|---|
| Zebrafish | | | |
| Genotyping | *rnpc3* | GAC GCA GGC GCA TAA AAT CAG TC | AGG CGT TCG TAA TGT TCA GG |
| | *Tp53* (WT) | AGC TGC ATG GGG GGG AT | GAT AGC CTA GTG CGA GCA CAC TCT T |
| | *Tp53* (Mutant) | AGC TGC ATG GGG GGG AA | GAT AGC CTA GTG CGA GCA CAC TCT T |
| RT-qPCR | *hrpt1* | GAG GAG CGT TGG ATA CAG A | CTC GTT GTA GTC AAG TGC AT |
| | *b2m* | GCG GTT GGG ATT TAC ATG TTG | GCC TTC ACC CCA GAG AAA GG |
| | *tbp* | CAG GCA ACA CAC CAC TTT AT | AAG TTT ACG TGG ACA CAA AT |
| | *rnpc3* | AGG CCC TGA AGG AAA CCA AT | TCA ACC AGG GCA GTC ACT TCA |
| | *cdkn2a/b (p16)* | CGA GGA TGA ACT GAC CAC AGC | CAA CAG CCA AAG GTG CGT TAC |
| | *cdkn1a (p21)* | CAA GCC AAG AAG CGT CTA GTG | AAC GGT GTC GTC TCT GGT TC |
| | *tp53* | TCC ACT CTC CCA CCA ACA TC | GGG AAC CTG AGC CTA AAT CC |
| | *Δ113tp53* | ATA TCC TGG CGA ACA TTT GG | ACG TCC ACC ACC ATT GAC AC |
| | *mdm2* | TGA CAA AGA AAC TGG TAA GA | AAA CAT AAC CTC CTT CAT GGT |
| | *bbc3 (puma)* | GAT GCC TTC AGC TTG GAC | GCC TGG ACA CTT CCT GTT CT |
| | *pmaip1 (noxa)* | ATG GCG AAG AAA GAG CAA AC | TCA TCG CTT CCC CTC CAT TTG |
| Mouse | | | |
| Genotyping | *Cre* | GGG ATT GCT TAT AAC ACC CTG TTA CG | TAT TCG ATC ATC AGC TAC ACC AG |
| | *Actin* | TGA GAA GCT GGC CAA AGA GAA GGG TTA C | GTG ACC TGT TAC TTT GGG AGT GGC AAG C |
| | *Rnpc3* | AGA GAA CAG TAT TTT AGT TCC AAG ATA TGC | CCT CAC ACA GCA TGG CTG AGA AGG |
| | *Neo/LacZ* (Lar3) | CAA CGG GTT CTT CTG TTA GTC C | *Rnpc3* specific primer |
| | *Rnpc3:* Un-recombined | TGG ATG ACA TGT GGA AAT GAT AA | AAT ACC CAA AAC ATG TAT TCA ACA |
| | *Rnpc3:* Recombined | AGA GAA CAG TAT TTT AGT TCC AAG ATA TGC | CAA TGC TAC ACC AAG TAA CT |
| | *Kras-LSL* | CCT TTA CAA GCG CAC GCA GAC TGT AGA | AGC TAG CCA CCA TGG CTT GAG TAA TGC A |
| | *Gp130*<sup>F/F</sup> (S319) | CTG AAT GAA CTG CAG GAC GA | |
| | *Gp130*<sup>F/F</sup> (MI012/14) | CAA GTG TTC TCA AGG TCC GAG TCC AC | TGA AGC ACT CGT CTT TA GC |
| | *Tff1-Cre* | CTG TCT GAG CAG GCA GTG TAA G | GGG ACA CAG CAT TGG AGT CAG A |
| RT-qPCR | *Gapdh* | CAA CTC ATC AAA GAT TGT CAG CAA | TAC TTG GCA GGT TTC TCC AGG C |
| | *Hrpt1* | GCC CCA AAA TGG TTA AGG TT | CAA GGG CAT ATC AAA CAA CA |
| | *Rnpc3* | CCA CCA TCA AGC ACA ATC CTC GCA | TCG GTC AAA TGT GTG GGC A |
| | *Parp1*_U12_intron | CGA CAC CTG CCT GCT GTA TAA TGA | AGG CAA GTT GTC CTG GGA TCT AAG AG |
| | *Vps16*_U12_intron | TCT CAC CTA CAC CCA GTA TCC CTA TG | ATA AGG CCT GTT TGC AGG GA T |
| | *Braf*_U12_intron | ACA CGC CAA GTC AAT CAT CCA CAG | GGG TGG TTC AGA CTT CGC AGA CC |
| | *Mapk1*_U12_intron | GCG CTT CAG ACA TGA GAA CA | GCA GTG GCA GCA GCT AAC TT |
| | *RasGrp2*_U12_intron | GCC TTA ACC CTC CTT TA CCT | TGA CCA GTG ACA CGT TTT CA |
| | *Ccnt2*_U12_intron | GGA GTG GAA GCG GAT GAA GA | CAG GGC CGC AAC TAG AAA A |
| | *Ccnk*_U12_intron | GGG CTC GCT TCA TCT TTG AT | GAC TGC ATG CTT GTC GAC AG |
| | *Cdk5*_U12_intron | GCA TCT TTG CAG AGC TGG CTA | AAG AGG TGA GGG GT GTA GGC |
| | *E2f1*_U12_intron | CAA CTG CAG GAG AGT GAG C | CAA TGC ACC CTG ACC AAT CC |
| | *Cdc45*_U12_intron | TCC CGT CAT AAC CAC CGA AA | ACG AGT ACT GCC TAA ACC CC |
| Human | | | |

**Table 1.** (continued)

| | Gene | Forward | Reverse |
|---|---|---|---|
| RT-PCR | RNPC3 | TCT GCT AGA CCA AAA CAA GAT CC | AGC TGT TAC GCA CAG TTC CAT AG |
| | PFDN5 | GCT AGA AAT GCT CAA GAA CCA GCT GG | AGG GCT GTG AGC TGC TGA ATC TTC |
| | TRAIP | TAA GAA CAG GCT TCG ATG GG | TTA ACA GGC AAT GGG CGG AT |
| | TTC23 | GCC AGA CAA ATC CTC GCC AA | CAC ATT GTA GCA GCT CCT TTG A |
| | VPS16 | GTG TGC TCA ATG CTG TTC GG | CCT GGA TGG TGA GCT GCT TAT |
| | VPS35 | GGC TGT GAA GGT CCA GTC ATT CCA A | CAC ACC ACG GCA CAT TTC TAC CAA ATC |
| | NCBP2 | ATG TCG GGT GGC CTC CTG AAG | AGC AAA TTC AAC AGG CCA AAG GAG TGT TT |
| | CDC45 | AGA ACA CAC TCT CCG TGG AC | TGA ACC TGG CTG CGG TAT AG |
| | E2F1 | TGT CGT CGA CCT GAA CTG GG | AGT CAG TGG CCT TGT TCT CC |
| | E2F2 | CGA GTC AGA GGA TGG GGT CC | TCC TCT GGG CAC AGG TAG AC |
| | E2F3 | ATG GGC CCT TGG GTA CTT GCC AAA T | GGA TTT GAA CAA GGC AGC AGA AGT GC |
| | GAPDH | CCA TGA CAA CTT TGG CAT TG | CCT GCT TCA CCA CCT TCT TG |
| RT-qPCR | RNPC3 | TCT GCT AGA CCA AAA CAA GAT CC | AGC TGT TAC GCA CAG TTC CAT AG |
| | PDCD7 | ACG CAT CAT CTT CAG CGA CT | ACT GTC GGA AAG CTT CCA AG |
| | GAPDH | CAA GAA GGT GGT GAA GCA G | CAG CGT CAA AGG TGG AG |
| | CCND1 | GCC CAG CAG AAC ATG GAC C | GTG GGT GTG CAA GCC AGG T |

redundant components of signalling pathways. Our interrogation of mutation data in the curated set of non-redundant studies (http://www.cbioportal.org/) (Cerami et al, 2012; de Bruijn et al, 2023; Gao et al, 2013) shows that *RNPC3* is rarely mutated in cancer. Indeed, out of a total of 105,260 samples in this dataset, only 116 (0.11%) contain a potentially debilitating *RNPC3* mutation. Focusing the analysis on the cancer types investigated in our study, we found no *RNPC3* mutations in 1950 AML samples, and only 0.1% (3 out of 3038) of HCC samples, 0.11% (6/5312) of lung adenocarcinomas and 0.32% (2/630) of gastric adenocarcinomas contain a mutation of unknown significance. This rarity of *RNPC3* mutations suggests that cancers generally do not have alternative mechanisms to circumvent 65K deficiency.

In summary, we propose that cancer cells that express mutant KRAS proteins and other strong oncogenes require efficient minor splicing and unencumbered expression of MIGs to support the molecular pathways they depend on. We believe that minor splicing inhibitors could offer a non-genotoxic solution to cancer therapy across a broad spectrum of cancer settings, with a viable therapeutic window and the potential to evade resistance mechanisms.

# Methods

### Reagents and tools table

| Reagent/resource | Reference or source | Identifier or catalog number |
|---|---|---|
| **Experimental models** | | |
| Zebrafish: *Rnpc3*<sup>zm00416857Tg/+</sup> | Markmiller et al, 2014 | RRID: ZFIN_ZDB-GENO-140806-3 |
| Zebrafish: Tg(fabp10:rtTA2s-M2;TRE2:EGFP-kras<sup>G12V</sup>) | Chew et al, 2014 | RRID: ZFIN_ZDB-ALT-151022-1 |

| Reagent/resource | Reference or source | Identifier or catalog number |
|---|---|---|
| Zebrafish: Tg(fabp10:dsRed;ela3l:GFP)<sup>gz12</sup> | Korzh et al, 2008 | RRID: ZFIN_ZDB-ALT-090424-3 |
| Zebrafish: Tg(ubiq:secAnnexinV-mKate) | Hall et al, 2019 | |
| Zebrafish: *Tp53*<sup>zdf1</sup> | Berghmans et al, 2005 | RRID: ZFIN_ZDB-ALT-050428-2 |
| Mouse: *Rnpc3*<sup>+/−</sup>; *Rnpc3*<sup>lox/lox</sup> | Doggett et al, 2018 | RRID: MGI_6276269 |
| Mouse: UBC-CreERT2 | Ruzankina et al, 2007 | RRID: IMSR_JAX:007001 |
| Mouse: Kras-LSL-G12D | Tuveson et al, 2004 | RRID: IMSR_JAX:008180 |
| Mouse: Gp130<sup>Y757F/Y757F</sup> | Tebbutt et al, 2002 | |
| Mouse: Tff1-CreERT2 | Thiem et al, 2016b | RRID: MGI:6257036 |
| Mouse: C57BL/6 | WEHI bioservices | RRID: MGI:3028467 |
| Cell line (Human): Phoenix-Eco | ATCC | RRID: CVCL_H717 |
| Cell Line (Human): A549 | Cell Bank Australia | RRID: CVCL_0023 |
| **Recombinant DNA** | | |
| MSCV-MLLENL-IRES-GFP | Lavau et al, 2000 | |
| Adenovirus: Ad5-CMV-Cre | University of Iowa Gene Transfer Core Facility | Cat#VVC-U of Iowa-5 |
| **Antibodies** | | |
| WB & IHC: Rabbit monoclonal anti-phospho-p42/44 | Cell Signaling Technologies | Cat#4370; RRID: AB_2315112 |
| WB: Mouse monoclonal anti-alpha-tubulin (DM1A) | Cell Signaling Technologies | Cat#3873; RRID: AB_1904178 |

| Reagent/resource | Reference or source | Identifier or catalog number |
|---|---|---|
| WB: Mouse monoclonal anti-Tp53 (9.1) | AbCam | Cat#ab77813; RRID: AB_10864112 |
| WB: Rabbit polyclonal anti-GAPDH (14C10) | Cell Signaling Technologies | Cat#2118; RRID: AB_561053 |
| WB: Mouse monoclonal anti-Tp53 (DO-1) | Santa Cruz | Cat#sc-126; RRID: AB_628082 |
| WB: Rabbit polyclonal TATA binding protein | Abcam | Cat#ab28175; RRID: AB_778239 |
| WB: Goat anti-mouse HRP | Agilent | Cat#P0447; RRID: AB_2617137 |
| WB: Goat anti-rabbit HRP | Agilent | Cat#P0448; RRID: AB_2617138 |
| WB: IR Dye 680LT Donkey Anti-rabbit IgG | LI-COR | Cat#925-68023; RRID: AB_2716687 |
| WB: IR Dye 800CW Donkey Anti-mouse IgG | LI-COR | Cat#925-32212; RRID: AB_2716622 |
| IHC: Rabbit polyclonal anti-65 K | AbCam | Cat#Ab90090; RRID: AB_2042822 |
| IF: Rabbit polyclonal anti gamma-H2AX (Ser139) | Gift from James Amatruda | RRID:AB_297813 |
| IF: Rabbit anti-phospho 53BP1 (ser1778) | Cell Signaling Technologies | Cat#2675; RRID: AB_490917 |
| IF: Goat anti-rabbit AF488 | Thermo Fischer Scientific | Cat#A11034; RRID:AB_2576217 |
| IF & FACS: Rabbit monoclonal anti gamma-H2AX (Ser139) (20E3) | Cell Signaling Technologies | Cat#9718; RRID: AB_2118009 |
| IF & FACS: Donkey anti-rabbit AF647 | Thermo Fischer Scientific | Cat#A31573; RRID:AB_2536183 |
| FACS: AnnexinV Alexa Fluor-647 | Life Technologies | Cat#A23-204; RRID: AB_2341149 |
| FACS: Anti-mouse Mac-1-PE (M1/70) | WEHI | RRID: AB_1019241 |
| FACS: Anti-mouse Gr-1-APC (RB6-8C5) | WEHI | RRID: AB_2621610 |
| **Oligonucleotides and other sequence-based reagents** | | |
| PCR Primers | This Study | Table 1 |
| Human ON-TARGETplus siRNAs to RNPC3 | Dharmacon Inc | Cat#J-021646-18-0010 Cat#J-021646-19-0010 |
| Human ON-TARGETplus siRNAs to PDCD7 | Dharmacon Inc | Cat#J-012096-06-0010 Cat#J-012096-05-0010 |
| Non-targeting control siRNA | Dharmacon Inc | Cat#D-001810-01-20 |
| U12-ASO | Integrated DNA Technologies | |
| NT-ASO | Integrated DNA Technologies – based on sequence from (Younis et al, 2013) | |
| **Chemicals, enzymes and other reagents** | | |
| Polybrene | Sigma | Cat#H9268 |
| Fugene®HD | Promega | Cat#E2311 |
| DharmaFECT 1 | Dharmacon Inc | Cat#T-2001-03 |

| Reagent/resource | Reference or source | Identifier or catalog number |
|---|---|---|
| Lipofectamine RNAiMAX Transfection Reagent | Thermo Fischer Scientific | Cat#13778075 |
| Doxycycline | Sigma | Cat#D9891 |
| mIL-3 | Peprotech | Cat#213-13 |
| mIL-6 | WEHI | |
| TPO | WEHI | |
| mSCF | WEHI | |
| mFlt-3 | WEHI | |
| 4-OHT | Sigma | Cat#H7904 |
| Tamoxifen | Sigma | Cat#T5648 |
| Neomycin | Sigma | Cat#N1876 |
| cOmplete Protease Inhibitor | Roche | Cat#11836170001 |
| PhosSTOP | Roche | Cat#04906837001 |
| Amersham ECL Western Blotting Detection kit | Cytiva | Cat#RPN2108 |
| Trisure | Bioline | Cat#38033 |
| EdU | Invitrogen | Cat#C10340 |
| Hoescht 33342 | Thermo Fisher Scientific | Cat#62249 |
| Prolong Diamond Antifade reagent with DAPI | Thermo Fisher Scientific | Cat#P36962 |
| Fluorogold | Sigma | Cat#39286 |
| Propidium Iodide | Sigma | Cat#P4864 |
| **Commercial kits** | | |
| RNeasy Micro Kit | QIAGEN | Cat#74004 |
| SuperScript III First Strand Synthesis System | Thermo Fisher Scientific | Cat#18080051 |
| SensiMix Sybr Hi-ROX Kit | Bioline | Cat#QT605-05 |
| Click-iT Edu Alexa Fluor 647 imaging kit | Thermo Fisher Scientific | Cat#C10340 |
| Click-iT Edu Flow Cytometry Assay kit | Thermo Fisher Scientific | Cat#C10646 |
| Cell Event Senescence Greed Detection Kit | Thermo Fisher Scientific | Cat#C10850 |
| VectaStain Elite ABC kit HRP | Vector Laboratories | Cat#PK-6100 |
| Liquid Diaminobenzidine substrate Chromogen system | DAKO | Cat#K3468 |
| Active Ras Pull-down & Detection kit | Thermo Fischer Scientific | Cat#16117 |
| NE-PER Nuclear and Cytoplasmic Extraction Kit | Thermo Fischer Scientific | Cat#78833 |
| TruSeq Stranded mRNA Total Library Preo with Ribo-Zero Gold rRNA depletion | Illumina | Cat#20020598 |
| **Software** | | |
| Prism v7.03 | GraphPad Software | https://www.graphpad.com |
| Imaris | | https://www.bitplane.com |
| FIJI | | https://fiji.sc |

| Reagent/resource | Reference or source | Identifier or catalog number |
|---|---|---|
| Arivis Vision 4D software | | https://www.arivis.com |
| Ilastik | | https://www.ilastik.org |
| FlowJo v10.1 | BD bioscience | RRID:SCR_008520 |
| Incucyte S3 | Essenbioscience | RRID:SCR_023147 |
| LinRegPCR v11.0 | | https://www.genetargetsolutions.com.au |
| Rsubread v2.4.3 | Liao et al, 2019 | RRID:SCR_016945 |
| Limma v3.46.0 | Ritchie et al, 2015 | RRID:SCR_010943 |
| edgeR v3.32.1 | Robinson et al, 2010 | RRID:SCR_012802 |
| Voom | Law et al, 2014 | |
| Pheatmap v1.0.12 | | RRID:SCR_016418 |
| IRFinder v1.2.0 | Middleton et al, 2017 | |

## Animal models

All procedures performed on zebrafish and mice were conducted with the approval of the Animal Ethics Committees of the Walter and Eliza Hall Institute of Medical Research, The University of Melbourne and the Parkville Branch of the Ludwig Institute for Cancer Research, Australia (2015.020 and 2019.014). All mice were maintained on a C57BL/6 background. Zebrafish were maintained on the Tubingen TL background and at 28 °C on a 12 h light/12 h dark cycle according to standard husbandry procedures. Details of mice and zebrafish strains used in this study are available in the reagents and tools table. All genotyping primer sequences can be found in Table 1.

## Inducing HCC in zebrafish larvae and adults

We induced mutant Kras expression in $TO(kras^{G12V})$ zebrafish larvae (Chew et al, 2014) by treatment with 20 µg/ml doxycycline (Sigma, #D9891) at 2 dpf in egg water with 0.003% 1-Phenyl-2-thiourea (PTU; Sigma, #P7629) to suppress pigmentation. Egg water was changed at 5 dpf and fresh doxycycline (20 µg/ml) added. To quantitate liver volume, zebrafish larvae were anaesthetized with benzocaine (200 mg/L; Sigma, #PHR1158) and mounted in 1% agarose. Image acquisition was performed using an Olympus FVMPE-RS multiphoton microscope with a ×25 objective and Olympus FV30-SW software. Excitation wavelengths for GFP and dsRed were 840 nm and 1100 nm, respectively. Emission was detected at 550 nm and 580 nm, respectively. For volumetric analysis of whole livers, Z-stacks with step-size 2 µm, were imported into ImageJ (1.49 v) or Imaris software.

We induced mutant Kras expression in $rnpc3^{+/+}$; $TO(krasG12V)^{T/+}$ and $rnpc3^{+/-}$; $TO(krasG12V)^{T/+}$ male zebrafish at 3–4 months of age by daily treatment with 20 mg/L dox for 7 d, during which time the tank water was changed and fresh dox administered every day. Controls of the same genotype were not treated with dox. After 7 d of treatment, zebrafish were euthanized and weighed pre- and post-dissection of the intact livers.

## Cell death analysis in zebrafish

To assess apoptosis, 7 dpf $TO(kras^{G12V})$;annexinV-mkate zebrafish larvae were fixed in 4% paraformaldehyde (Thermo Fisher, #28906) and the livers collected by microdissection. Image acquisition was performed using a Zeiss LSM 880 microscope with a 20x objective and ZEN software. Excitation wavelengths for mKate and GFP were 560 nm and 900 nm, respectively. Liver volume was quantified and 3D segmentation of the AnnexinV-mKate signals was performed in FIJI.

## Cell cycle analysis in zebrafish

Live zebrafish larvae (7 dpf) were incubated in 2 mM EdU in egg water for 2 h followed by a further incubation in fresh egg water for 1 h. Larvae were euthanized using benzocaine (1000 mg/L) prior to removal of the liver by dissection. EdU labeling was carried out using the Click-iT Edu Alexa Fluor 647 (AF647) imaging kit (Invitrogen, #C10340) according to the manufacturer's instructions. Livers were co-stained with Hoechst 33342 (1:250; Thermo Fisher, #62249). Image acquisition was performed using an Olympus FVMPE-RS multiphoton microscope with excitation wavelengths of 950 nm and 1160 nm for Hoechst 33342 and AF647, respectively. The numbers of Hoechst 33342 and EdU-positive cells were quantified using Arivis Vision4D software.

## Mouse lung adenocarcinoma model

8–16-week-old $Kras$-LSL-G12D mice (Tuveson et al, 2004) were exposed to adenoviral-Cre (AdCre) in the lung airway epithelia. AdCre:CaPi coprecipitates were prepared (Fasbender et al, 1998) and delivered intranasally to individual mice at a dose of $2.5 \times 10^8$ plaque forming units in 50 µl MEM under isoflurane anesthesia. This treatment served to recombine both the $Kras$-LSL-G12D locus and loxP-flanked Rnpc3 alleles when present.

To analyze the degree of hyperplasia in the lungs of $Kras^{LSLKrasG12D}$ mice, animals were anesthetized, their lungs inflated (250 mm-$H_2O$ pressure) and fixed by cardiac perfusion of phosphate-buffered 4% paraformaldehyde. Lungs were harvested, soaked overnight at 4 °C in the same fixative followed by embedding in paraffin. Histological sections (4 µm) were cut and stained with hematoxylin and eosin (H&E). Three left and right lung lobe sections with at least 100 µm between them were imaged using a Panoramic Scan II. Quantification of tissue area was performed using FIJI. Segmentation was carried out using Ilastik (Interactive learning and segmentation toolkit; https://www.ilastik.org). The area and number of tumor lesions from six slides/mouse were averaged to give one data point per mouse. Differences in lung tumor grade were based on criteria established by Nikitin and colleagues (Nikitin et al, 2004).

## Mouse gastric cancer model

$Gp130^{Y757F/Y757F}$ mice in which gastric adenoma development occurs spontaneously and with 100% penetrance by 100 d (Jenkins et al, 2005; Tebbutt et al, 2002) were used to investigate gastric cancer. We disrupted the Rnpc3 locus in the glandular epithelium of the stomach by crossing $Rnpc3^{lox/lox}$;$Gp130^{Y757F/Y757F}$ mice with Tff1-CreERT2 mice (Thiem et al, 2016b). We administered tamoxifen (Sigma, St. Louis, MO; 30 mg/ml) to adult $Rnpc3^{lox/lox}$;$Gp130^{Y757F/}$

*Y757F*;*Tff1-CreERT2* mice by oral gavage in two consecutive daily doses (150 µl). Stomachs were collected from mice euthanized at 100 d or 180 d of age, opened longitudinally, washed three times in PBS with vigorous shaking and pinned out on silicone-coated plates and photographed. Gastric adenomas were resected, weighed and either snap frozen for later molecular analysis or fixed in 10% buffered formalin solution (pH 7.4) overnight, prior to embedding in paraffin for immunohistochemical analysis.

## Generation and propagation of mouse AML cells

Phoenix cells were cultured in Dulbecco's DMEM (Gibco, #11885084) supplemented with 10% fetal bovine serum (FBS; GE Healthcare Bio-Sciences, #SH30088.03), 10% $CO_2$. The MSCV MLL-ENL IRES GFP retroviral construct was obtained from Dr. Stefan Glaser (first described in (Lavau et al, 2000)) and transduced into Phoenix cells using 4 µl of Fugene (Promega, #E2311) per µg of plasmid DNA. Viral supernatants harvested 24 and 48 h later (Swift et al, 2001). Cells dissociated from E13.5 (embryonic day 13.5) mouse livers were cultured at 37 °C with 10% $CO_2$ in DMEM with 20% FBS, 100 ng/ml murine stem cell factor (mSCF), 50 ng/ml murine thrombopoietin (mTPO), 10 ng/ml murine interleukin-6 (mIL-6) and 10 ng/ml murine FMS-like tyrosine kinase 3 (mFlt-3), all produced in-house.

Fetal liver cells were virally transduced with MSCV MLL-ENL IRES GFP by spin infection with 4 µg/ml polybrene (Sigma, #H9268) on two consecutive days as described in (Bilardi et al, 2016). In all, $1 \times 10^6$ cells were transplanted into 6–8 weeks old sub-lethally γ-irradiated (7.5 Gy) C57BL/6 female mice, 24 h prior to transplantation by tail-vein injection. Blood was collected from the retro-orbital plexus and cell counts were obtained with an Advia 2120 hematological analyzer to monitor disease onset. As each mouse reached the ethical endpoint and was euthanized, primary AML cells were harvested from the bone marrow and spleen and filtered through 100 µm cell strainers to generate single cell suspensions. Red blood cell lysis was performed on spleen cell preparations. Primary AML cells were expanded by transplanting into secondary recipient mice (no prior irradiation). Secondary transplant cells were harvested from the spleen and bone marrow of at least three independent recipient mice per genotype and used to analyze the impact of disrupting the *Rnpc3* locus in vitro and in vivo. AML cells were cultured in DMEM with 10% FBS and mIL-3 (6 ng/ml, Preprotech, #213-13). *UBC-CreERT2* (Ruzankina et al, 2007) mediated recombination of the *Rnpc3* locus in vitro was achieved using treatment with 200 nM 4-OHT (Sigma, #H7904). Cre-mediated recombination of the *Rnpc3* locus in vivo was achieved using 30 mg/ml TMX (Sigma, #T5648) administered by oral gavage on day 13 and 14 following transplant.

## AML fluorescence-activated cell sorting

To determine the burden and phenotype of AML at the time of euthanasia, live peripheral blood and/or bone marrow cells were collected and stained with FluoroGold (Sigma, #39286), anti-Gr-1 (RA6-8C5) and anti-Mac-1 (M1/70) antibodies (Walter and Eliza Hall Institute Monoclonal Antibody Facility). GFP fluorescence was used to detect AML cells expressing the MLL-ENL fusion protein. The viability of AML cells in vitro following treatment with 200 nM 4-OHT for timed intervals was assessed using

AnnexinV-Alexa Fluor 647 (Life Technologies, #A23-204) and 4 µg/ml propidium iodide (Sigma, #P4864) exclusion staining. Briefly, cells were washed once with balanced salt solution (150 mM NaCl, 3.7 mM KCl, 2.5 mM $CaCl_2$, 1.2 mM $MgSO_4$, 7.4 mM HEPES, NaOH, 1.2 mM $KH_2PO_4$ and 0.8 mM $K_2HPO_4$) containing 5% FBS and resuspended in the same medium containing the two reagents. Data was collected on an LSR-II flow cytometer (BD Biosciences) and cell viability analyzed using FlowJo v10.1 software (FlowJo LLC).

## Knockdown of minor spliceosome components in human A549 lung adenocarcinoma cells

Human lung adenocarcinoma, A549 cells were authenticated using small tandem repeat (STR) profiling (CellBank Australia, Report #19-338) and were cultured in RPMI (Gibco, Cat#11875093) medium supplemented with 1× GlutaMAX (Gibco, #35050061) and 5% FBS.

Minor class splicing knockdown was achieved by two methods siRNA or ASO.

siRNA—A549 cells were transfected with 50 nM of two independent human ON-TARGETplus siRNAs to *RNPC3* or *PDCD7* and a non-targeting (NT) control siRNA using Dharma-FECT 1 (Dharmacon Inc., #T-2001-03) according to the manufacturer's protocol.

ASO—25mer antisense oligonucleotides with chemically modified (MethoxyEthoxy) bases to improve stability (underlined in the sequences below) were purchased from Integrated DNA Technologies (IDT). The control NT-ASO (5′-CCT CTT ACC TCA GTT ACA ATT TAT A-3′) was described previously (Younis et al, 2013). The U12-ASO: (5′-TCG TTA TTT TCC TTA CTC ATA AGT T-3′) was designed to target bases in Stem Loop 1 (SL1) close to the 5′ end of U12 snRNA. A549 cells were transfected with ASOs (final concentration 39 pM-10 nM) using Lipofectamine RNAiMAX Transfection Reagent (Thermofisher, #13778150).

All transfections were carried out in at least triplicate and treated as independent replicates thereafter.

## Proliferation and cell cycle analysis in human cancer cells

A549 cells were incubated immediately after transfection in an IncuCyte S3 live cell analysis system (Essenbioscience) and imaged every hour for 72 h. Confluency over the time course was normalized to the first time point for each image acquired 1 h post-transfection.

## FACS analysis of cell cycle and DNA damage in human cancer cells

All FACS analysis was carried out on single cell suspensions of A549 cells 72 h after transfection with 10 nM ASO. The Click-iT plus EdU flow cytometry assay kit (Invitrogen, #C10646) following the manufacturer's instructions was used to assess cell cycle, with cells incubated with 10 µM EdU 2 h prior to harvest. The CellEvent senescence green detection kit (Invitrogen, #C10850) following the manufacturer's instructions was used to assess senescence. 1:100 phospho-histone H2A.X (Ser139; Cell Signaling Technologies, #9718) was used to stain DNA damage, followed by 1:200 donkey anti-rabbit AF647. For all assays 10,000 viable cells were collected on a BD FACSymphony and analysed using FlowJo v10.1 software (FlowJo LLC).

## Immunoblotting

Samples of mouse and zebrafish tissues were lysed in RIPA buffer (20 mM Hepes, pH 7.9, 150 mM NaCl, 1 mM $MgCl_2$, 1% NP40, 10 mM NaF, 0.2 mM $Na_3VO_4$, 10 mM β-glycerol phosphate). Nuclear protein extracts were obtained from A549 cells using NE-PER Nuclear and Cytoplasmic Extraction Reagents (Thermo Fisher Scientific, #78833) as per the manufacturer's protocol. All buffers were supplemented with cOmplete Protease and PhosSTOP inhibitors (Roche, #11836170001 & #04906837001). Nuclear protein lysates were treated with 50 ng/μL DNase I (Worthington Biochemical, #NC9199796), incubated for 30 min on ice and cleared by centrifugation at 13,000 rpm for 20 min at 4 °C. The protein concentration of samples was determined by BCA protein assay (Thermo Fisher Scientific, #23227). In all, 25–50 μg of protein per lane were resolved on NuPAGE Novex Bis-Tris 4–12% polyacrylamide gels (Invitrogen, #NP0321BOX). Mouse and human samples were transferred to Immobilon-FL PVDF membranes (Millipore, #IPFL00010) and blocked in Odyssey Blocking Buffer (LI-COR, #927 40000). Mouse gastric polyp blot was incubated with anti-phospho-p44/42 MAPK (ERK1/2) (Thr202/Tyr204, 1:1000; Cell Signaling Technology, #4370) and anti-alpha tubulin (DM1A) (loading control) (1:2000; Cell Signaling Technology, #3873). A549 nuclear/cytoplasmic protein membrane was incubated with primary antibodies anti-Tp53 (DO-1) (1:500; Santa Cruz, #sc-126), anti-GAPDH (14C10) (1:2000; Cell Signaling Technology, #2118) and anti-TATA Binding Protein (1:2000; Abcam, #ab28175). Membranes were incubated with secondary antibodies: IRDye 680LT donkey anti-rabbit (1:10,000; LI-COR, Cat#925-68023) and IRDye 800CW donkey anti-mouse (1:10,000; LI-COR, #925-32212) and scanned using an Odyssey infrared imaging system (Li-COR). Zebrafish samples were transferred to nitrocellulose blotting membranes (Amersham, #10600003), blocked with 5% BSA and incubated with primary antibodies anti-Tp53 (9.1), (1:500; Abcam, #ab77813) and anti-GAPDH (14C10) (1:1000; Cell Signaling Technology, #2118). Secondary antibodies goat anti-mouse HRP (1:5000; DAKO, #P0447) and goat anti-rabbit HRP (1:5000; DAKO, #P0448) were incubated with membranes and signals developed using Amersham ECL Western Blotting Detection Kit (Cytiva, #RPN2108) and imaged on a Chemidoc Touch (Biorad). Relative protein abundance was calculated based on normalized integrated intensity.

## Active Ras pull-down and detection

Total lysate was isolated from 7 dpf zebrafish larvae and equal amounts of protein (50 mg) were used as input for the pull-down of activated GTP-Ras proteins using the Active Ras Pull-Down and Detection kit (ThermoFisher, #16117) according to the manufacturers' instructions.

## Immunohistochemistry

Immunohistochemical analysis was carried out on unstained histological sections of mouse tissue using either anti-65K antibody (1:250; Abcam, #ab90090) or anti-pERK (1:1000; Cell Signaling Technologies, #4370). Prior to staining, antigen retrieval was performed on the sections by heating to 100 °C (microwave) in 10 mM pH 6 sodium citrate buffer. Endogenous peroxidase activity was inhibited with 3% $H_2O_2$ (v/v; Thermo Fisher Scientific, #H325-100). Non-specific antibody binding was blocked using 10% goat serum (w/v; Sigma, #G9023). Primary antibody binding was performed overnight at 4 °C in a humidified container. Sections

were incubated with anti-rabbit biotinylated secondary antibody and Vectastain Elite ABC HRP reagent (Vector Laboratories, #PK-6100) for 30 min at RT. Signals were detected with the Liquid Diaminobenzidine substrate Chromogen System (DAKO, #K3468) prior to counterstaining with haematoxylin. All images were captured using an upright Nikon Eclipse 90i or a Panoramic Scan II.

## Cryosectioning and immunofluorescence microscopy analysis

Dissected zebrafish livers were fixed in 4% PFA overnight at 4 °C and washed with PBS/0.1% Tween 20 before incubation in 30% sucrose in PBS overnight at 4 °C. Livers were aligned in a tissue mold, embedded in OCT and frozen on dry ice. Livers were sectioned at 10 μm intervals using a Thermofisher Scientific Microm HM550 cryostat. Sections were washed with PBS before blocking with 10% FCS in PBS/0.3% Triton X-100. Incubation with γ-H2AX (1:1000; gift of James Amatruda) antibody was performed at 4 °C overnight, followed by 1 h at RT with anti-rabbit AF647 (1:500; Thermofisher Scientific, #A31573). Prolong Diamond Antifade Mountant with DAPI (Thermofisher #P36962) was used for slide mounting.

A549 cells were seeded on chamber slides (Lab-TEK II) and treated with 10 nM ASO for 72 h. Cells were fixed for 15 mins in 4% PFA at RT, washed twice with PBS and then blocked for 2 h with 1% w/v BSA, 1% v/v FCS, 0.5% v/v Triton-X100 in PBST. Slides were incubated with 1:100 phospho-53BP1 (Ser1778; Cell Signaling Technology, #2675) or phospho-histone H2A.X (Ser139; Cell Signaling Technology, #9718) in 0.2% w/v BSA, 0.5% v/v Triton-X100 in PBST at 4 °C overnight, followed by 1 h at RT with anti-rabbit AF488 (1:500; Thermofisher Scientific, #A11034) and DAPI 1 ng/ml.

The Zeiss LSM880 Fast Airyscan Confocal microscope was used for image acquisition and image analysis was performed in ImageJ.

## RNA analysis by RT-PCR and RT-qPCR

Total RNA was extracted from independent pools of dissected zebrafish livers using the RNeasy Micro Kit (QIAGEN, #74004). Total RNA was extracted from human and mouse cells and tissues using TRIsure™ reagent (Bioline, #38033). RNA integrity was assessed using either a High Sensitivity RNA ScreenTape assay (Agilent, #5067-5579) on a 2200 TapeStation or on a 2100 Bioanalyzer (Agilent). cDNA was generated from 1–10 μg RNA using the Superscript III First Strand Synthesis System (Invitrogen, #18080051) and oligo(dT) priming according to the manufacturer's instructions. RT-quantitative PCR (RT-qPCR) was performed using a SensiMix SYBR kit (Bioline, #QT605-05) on an Applied Biosystems ViiA™7 Real-Time PCR machine. Zebrafish expression data were normalized by reference to *hrpt1*, *b2m* and *tbp*; mouse expression data were normalized by reference to *Gapdh* and *Hprt1* expression, and human to *GAPDH* and *CCND1* expression. LinRegPCR V11.0 was used for baseline correction, PCR efficiency calculation and transcript quantification analysis (Ramakers et al, 2003). To detect spliced transcripts by RT-PCR, primers were designed to amplify sequences spanning exon-exon borders. To detect minor intron retention by RT-qPCR, primers were designed to hybridise to sequences in an upstream (5′) or downstream (3′) exon and their adjacent minor intron (Primer sequences; Table 1).

Relative expression levels were calculated by the $2^{-\Delta\Delta Ct}$ method and all results were expressed as the mean ± SEM of at least three independent biological replicates.

## RNAseq of A549 cells

A549 cells were transfected for 72 h with 50 nM si*RNPC3* ($n = 4$ independent dishes) and 50 nM non-targeting (NT) siRNA ($n = 4$ independent dishes). Total RNA was extracted from cells using TRIsure™ reagent (Bioline, #38033) and treated with DNase 1. cDNA library preparation and RNAseq was performed by the Australian Genome Research Facility (AGRF, Melbourne, Australia). RNA integrity and concentration was assessed with a 2100 Bioanalyzer (Agilent) and 100 ng samples of total RNA were used for cDNA library preparation (TruSeq Stranded mRNA Total Library Prep with Ribo-Zero Gold rRNA depletion; Illumina, #20020598). cDNAs were sequenced with an Illumina NovaSeq 6000, yielding >40 M stranded paired-end reads (150 bp) per sample. RNAseq data were deposited at the GEO with Accession ID GSE190943.

## Differential expression analysis of A549 cells by RNAseq

All reads were aligned to the human genome, build hg38, using the Rsubread (Liao et al, 2019) software package and the align function (v2.4.3). In all cases at least 93% of all fragments (read pairs) mapped to the genome. All fragments overlapping genes were summarized into counts using Rsubread's featureCounts function. Genes were identified using Gencode annotation to the human genome (v37). Differential expression analyses between the si*RNPC3* and siNT conditions were then undertaken using the limma (v 3.46.0) (Ritchie et al, 2015) and edgeR (v 3.32.1) (Robinson et al, 2010) software packages.

Prior to analysis, all genes labeled 'To Be Experimentally Confirmed' (TEC) were removed. Expression based filtering was then performed using edgeR's filterByExpr function with default parameters. A total of 24,644 genes remained. Sample composition was then normalized using the TMM method (Robinson and Oshlack, 2010). To identify differentially expressed genes between the conditions, the data was first transformed to log-counts per million (logCPM) with associated precision weights using voom (Law et al, 2014). Differential expression was then assessed using linear models and robust empirical bayes moderated t-statistics (limma-voom pipeline) (Phipson et al, 2016). To increase precision, the linear models incorporated a correction for replicate. The false discovery rate (FDR) was controlled below 5% using the Benjamini and Hochberg method. The heatmap of the minor intron genes was generated using the pheatmap software package (v 1.0.12). Limma's removeBatchEffect function was first applied to the logCPM data prior to generating the heatmap to remove the replicate batch effect.

## Pathway analysis

To identify biological functions that could be affected by aberrant minor intron splicing, all MIGs with significantly ($P < 0.05$) elevated minor intron retention were submitted to g:Profiler. To identify pathways, diseases and functions affected upon si*RNPC3*, all affected genes (differentially expressed genes + MIGs with elevated retention and/or AS) were used for Ingenuity Pathway Analysis (IPA). Terms with $P < 0.05$ were curated.

## Intron retention and alternative splicing analysis

To determine levels of intron retention (IR) and alternative splicing (AS) caused by *RNPC3* knockdown in A549 cells, RNAseq reads were aligned to the hg38 genome using Hisat2 with the –trim-5 1 option enabled. Minor intron retention and de novo alternative splicing events were then studied as described previously (Olthof et al, 2019). Minor intron retention levels are reported as a mis-splicing index ($MSI_{ret}$), which calculates the reads aligned to the 5′ and 3′ exon-intron boundaries as a ratio of the total number of reads spanning the canonical exon-exon junction. Other forms of alternative splicing (e.g., exon skipping) were quantified by calculating the ratio of spliced reads supporting an aberrant exon-exon junction over the total number of spliced reads aligning to the canonical exon-exon junction. Alternative splicing levels are reported as $MSI_{AS}$. Significance of AS and IR events was assessed using a two-tailed $T$ test assuming unequal variance.

For analysis of global intron retention (both major and minor), we used IRFinder v1.2.0 following the default human genome pipeline (Middleton et al, 2017). Differential IR (DIR) was called using an Audic and Claverie test. To ensure robust calling of DIR, only those significant DIR introns marked as "clean" and with uniform or non-uniform coverage were considered. For an intron to be called as significantly retained in si*RNPC3* samples, we required the following: FDR ≤ 0.05, 100% coverage of the intron in si*RNPC3* samples, ≥10% IR ratio in siRNPC3 and ≥ 5% IR ratio over NT samples. Likewise, for an intron to be significantly retained in NT samples, we required FDR ≤ 0.05, 100% coverage of the intron in NT samples, ≥10% IR ratio in NT cells and ≥5% IR ratio over si*RNPC3* samples. Minor introns were called based on the MIDB hg38 database (Olthof et al, 2019).

## Quantification and statistical analysis

Data are expressed as mean ± SEM unless indicated otherwise and the number of biological replicates indicating samples from individual animals for each experiment are stated in the figure legends. No blinding was performed. $P$-values were calculated using Student's $t$ tests (two-tailed, followed by Welch's correction) when comparing two groups, or tested by ANOVA followed by Tukey's post-hoc test when comparing multiple groups. Survival data were plotted as Kaplan–Meier curves with significance calculated by Mantel–Cox log-ranked test. Chi-square goodness of fit test was used to evaluate whole transcriptome analysis and determine if MIG changes were statistically significant. All analysis was performed using GraphPad Prism V7.03 (GraphPad software) and $P ≤ 0.05$ was considered statistically significant.

# Data availability

The RNAseq dataset generated in this study has been deposited in GEO (GSE190943).

The source data of this paper are collected in the following database record: biostudies:S-SCDT-10_1038-S44319-025-00511-8.

# Peer review information

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

## Acknowledgements

The authors thank Tyson Blanch, Cameron Mackey, Dora McPhee, Mark Greer, Lysandra Richards and Elizabeth Grgacic (zebrafish husbandry), Andrew Naughton, Melanie Asquith, Mel Pritchard, Faye Dobrowski, Emily Sutherland and Leanne Johnson (mouse husbandry) and Qian Du, Julia Griesbach, Charlotte Burstroem and Samantha Eccles (technical assistance). KJM was supported by an Australian Government Research Training Program Scholarship. The work was supported in part by National Institute of Neurological Disorders and Stroke (R01NS102538 to RNK). The majority of this  work was supported by the National Health and Medical Research Council of Australia (Project Grants 1024878 and 1161336 to JKH), Ludwig Institute for Cancer Research, a Victorian State Government Operational Infrastructure Support grant and the Australian Government NHMRC Independent Research Institutes Infrastructure Support Scheme (IRIISS).

## Author contributions

**Karen Doggett**: Conceptualization; Data curation; Formal analysis; Supervision; Investigation; Methodology; Writing—original draft; Writing—review and editing. **Kimberly J Morgan**: Data curation; Formal analysis; Methodology. **Anouk M Olthof**: Data curation; Software; Formal analysis; Investigation; Methodology; Writing—review and editing. **Stephen Mieruszynski**: Data curation; Formal analysis; Validation; Investigation; Writing—review and editing. **Benjamin B Williams**: Conceptualization; Data curation; Formal

analysis; Investigation; Methodology. **Alexandra L Garnham**: Formal analysis. **Michael J G Milevskiy**: Formal analysis; Investigation; Methodology. **Lachlan Whitehead**: Formal analysis. **Janine Coates**: Formal analysis; Investigation. **Michael Buchert**: Investigation; Methodology. **Robert J J O'Donoghue**: Investigation; Methodology. **Thomas E Hall**: Resources. **Tracy L Putoczki**: Investigation; Methodology. **Matthias Ernst**: Resources; Methodology. **Kate D Sutherland**: Resources; Methodology. **Rahul N Kanadia**: Conceptualization; Software; Supervision; Methodology; Writing—review and editing. **Joan K Heath**: Conceptualization; Resources; Data curation; Formal analysis; Supervision; Funding acquisition; Methodology; Writing—original draft; Project administration; Writing—review and editing.

Source data underlying figure panels in this paper may have individual authorship assigned. Where available, figure panel/source data authorship is listed in the following database record: biostudies:S-SCDT-10_1038-S44319-025-00511-8.

## Disclosure and competing interests statement

The authors declare no competing interests.

# Expanded View Figures

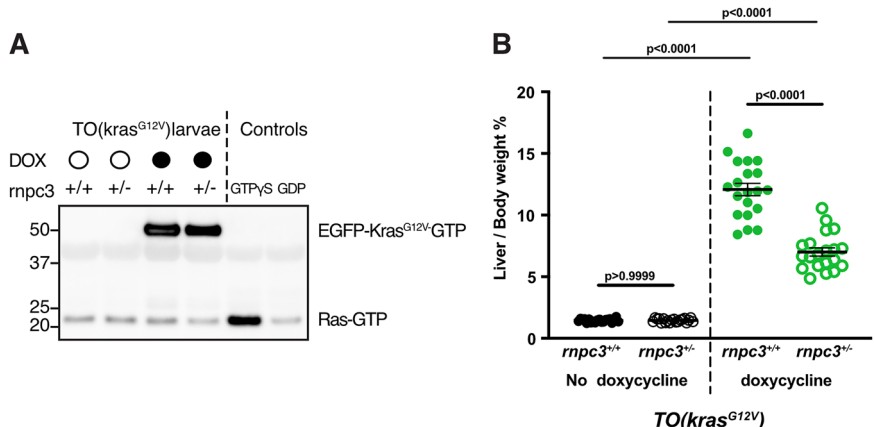

**Figure EV1. Heterozygous loss of *rnpc3* reduces tumour burden in a *kras^G12V*-driven zebrafish model of hepatocellular carcinoma (HCC).**

(A) Western blot of active Ras-GTP protein following active Ras pull-down from *TO(kras^G12V)* larvae of the indicated *rnpc3* genotype. (B) *rnpc3^+/+* and *rnpc3^+/−* liver/body mass ratio (%) of adult male *TO(kras^G12V)* (3.5–4 months of age) with and without dox treatment. *n* = 20, 2 pooled independent experiments. Mean ± SEM, significance was assessed by one-way ANOVA with Tukey's multiple comparison test.

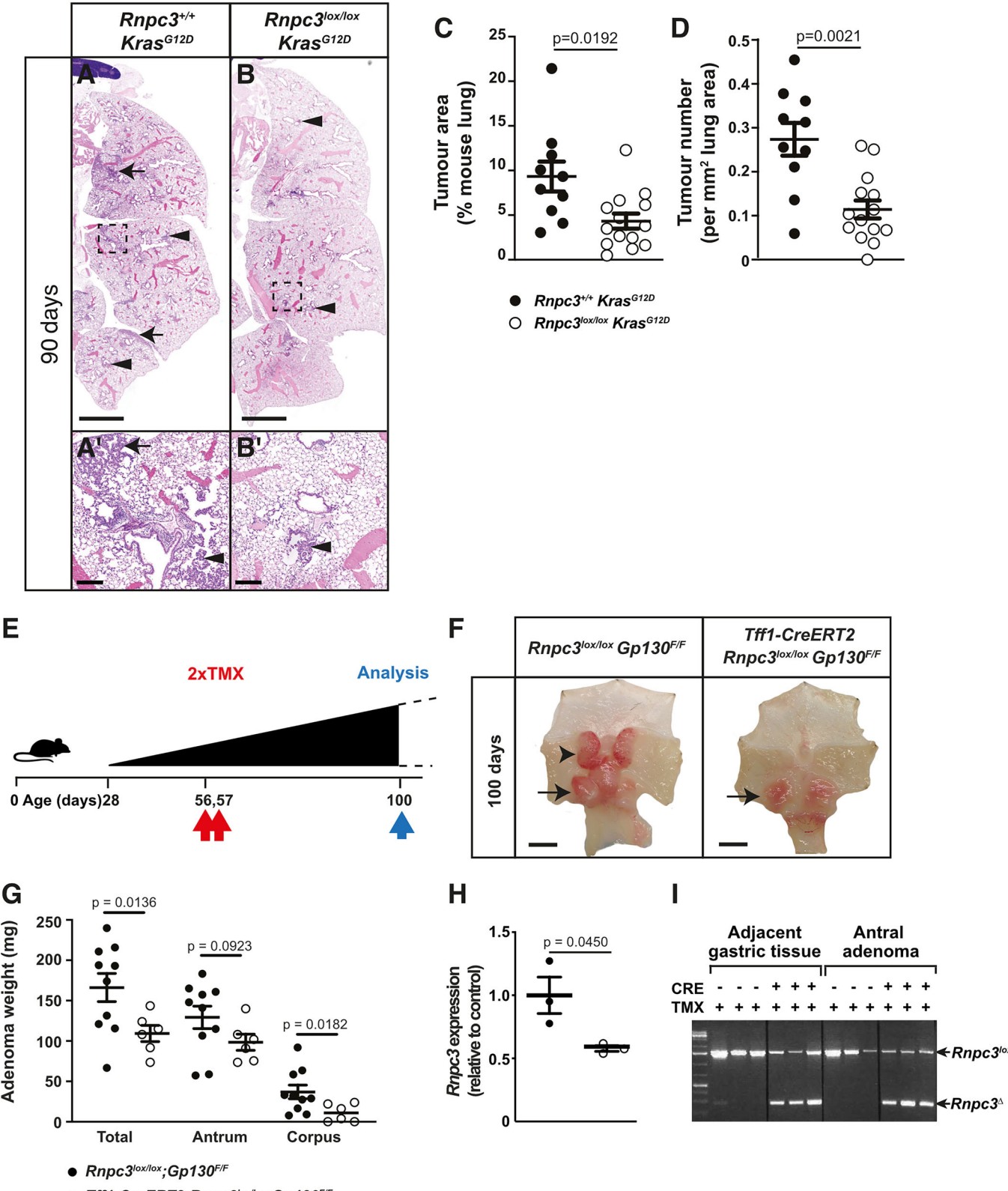

**Figure EV2.** **Recombination of *Rnpc3^lox* alleles in mice reduces tumour burden in *Kras^G12D*-driven lung adenocarcinoma and a *Gp130^F/F* gastric adenoma model.**

(A, B) Histological analysis of lung hyperplasia. Areas of AAH (atypical adenomatous hyperplasia; arrows) and bronchial hyperplasia (arrowheads) were observed in the lungs 90 d after adenoviral Cre recombinase administration. Scale bar is 2 mm. (A', B') Higher magnification images of region in dashed box. Scale bar is 200 μm. (C, D) Quantitation of the frequency and area of the hyperplastic lesions. Data are expressed as mean ± SEM, $n = 10$ or 14 per genotype. Significance was assessed using an unpaired Student's *t* test with Welch's correction. (E) Schematic diagram of gastric adenoma formation in *Gp130^F/F* mice and treatment with tamoxifen (2 x TMX) on days 56 and 57 (red arrows) to induce recombination of *Rnpc3^lox* alleles. (F) Representative stomachs with adenomas in the corpus (arrowhead) and antrum (arrow). Scale bar is 5 mm. (G) Quantitation of adenoma weight. Data are expressed as mean ± SEM, $n = 6$ or 10 per genotype. Significance was assessed using an unpaired Student's *t* test with Welch's correction. (H) RT-qPCR analysis of *Rnpc3* mRNA in single antral adenomas from three independent mice per genotype. Data are expressed as mean ± SEM. Significance was evaluated with a two-tailed Student's *t* test. (I) PCR of genomic DNA for *Rnpc3* alleles in normal glandular stomach and adenomas harvested from TMX-treated, *Tff1-CreERT2;Rnpc3^lox/lox;Gp130^F/F* mice, $n = 3$.

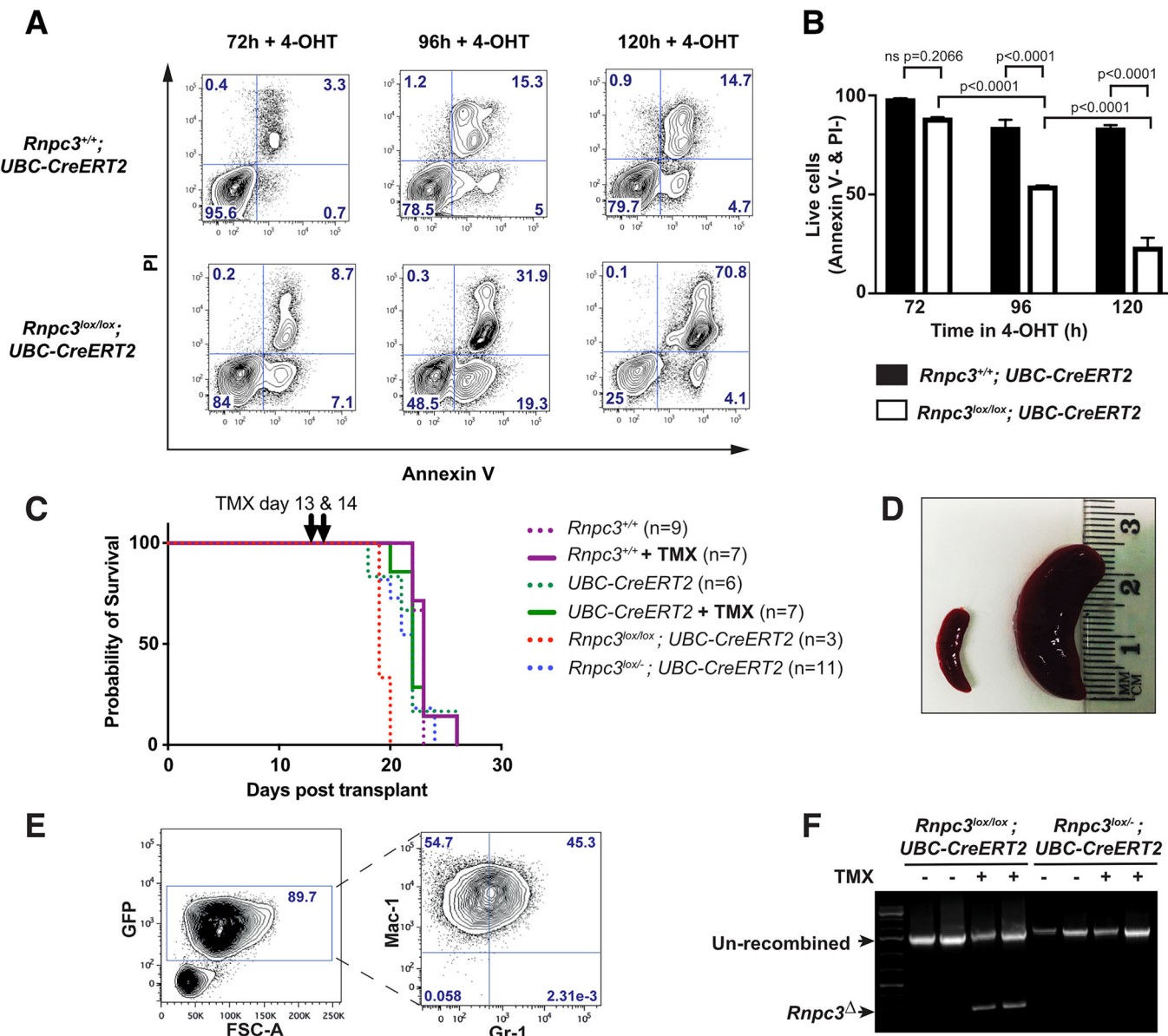

**Figure EV3. Recombination of the *Rnpc3* locus in MLL-ENL AML cells in vitro reduces cell viability.**

(A) Representative fluorescence-activated cell sorting (FACS) plots of cell viability using AnnexinV and PI staining. Numbers in each quadrant indicate the percentage of cells in each category. (B) Quantitation of the percentage of viable (AnnexinV⁻, PI⁻) cells over time enumerated by FACS. Percentage viable cells expressed as mean ± SEM, *n* = 3 for all cohorts. Significant differences were assessed with a one-way ANOVA with Tukey's multiple comparison test. (C) Kaplan–Meier survival plot of mice harbouring tertiary transplants of MLL-ENL AML cells with all control genotypes in the presence (solid line) and absence (dotted line) of TMX treatment 13 and 14 d (black arrows) following transplantation. There is no significant difference in median survival (19–23 d) between all control cohorts (i.e. no *Rnpc3lox* alleles and no TMX treatment), *n* = 311. Significance was assessed with a Mantel–Cox test. (D) Spleen from a WT mouse not transplanted with AML cells next to a spleen harvested from a mouse transplanted with *UBC-CreERT2;Rnpc3lox/lox* AML cells and treated with TMX. (E) Example FACS analysis of *UBC-CreERT2;Rnpc3lox/lox* AML cells. (F) Genomic analysis of the *Rnpc3* locus in tertiary transplanted AML cells.

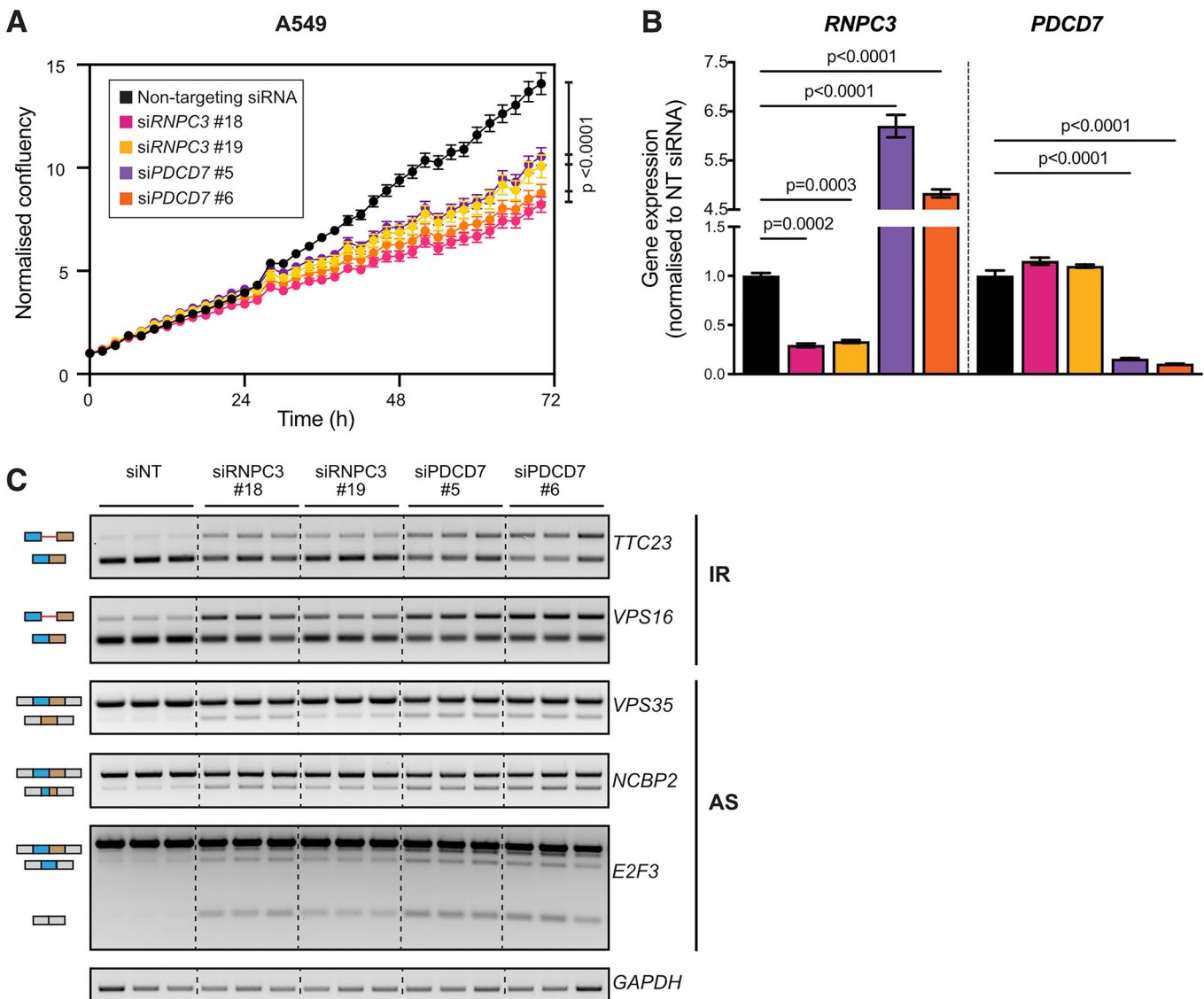

**Figure EV4.** **Analysis of cell proliferation and aberrant splicing of MIGs in A549 cells treated with si*RNPC3* and si*PDCD7* for 72 h.**

(A) Quantification of A549 cell growth over 72 h treatment with non-targeting (NT), two independent *RNPC3* or 2 independent *PDCD7* siRNAs. Data are represented as mean ± SEM ($n = 3$, 25 images per well, every hour). Significance was assessed by 2way ANOVA with Dunnett's multiple comparisons test. Note: the siNT and si*RNPC3* data are the same as shown in Fig. 5B. (B) RT-qPCR analysis of *RNPC3* and *PDCD7* mRNA in A549 cells after 72 h treatment with siRNAs. Data are represented as mean ± SEM ($n = 3$), significance was tested using a one-way ANOVA with Tukey's multiple comparisons test. (C) RT-PCR analysis of example MIG splicing changes, identified in A549 cells treated with si*RNPC3* or si*PDCD7* for 72 h. Schematised IR and AS events are represented on the left with the minor intron in red and the flanking upstream and downstream exons in blue and orange, respectively. Exons not separated by a minor intron are grey.

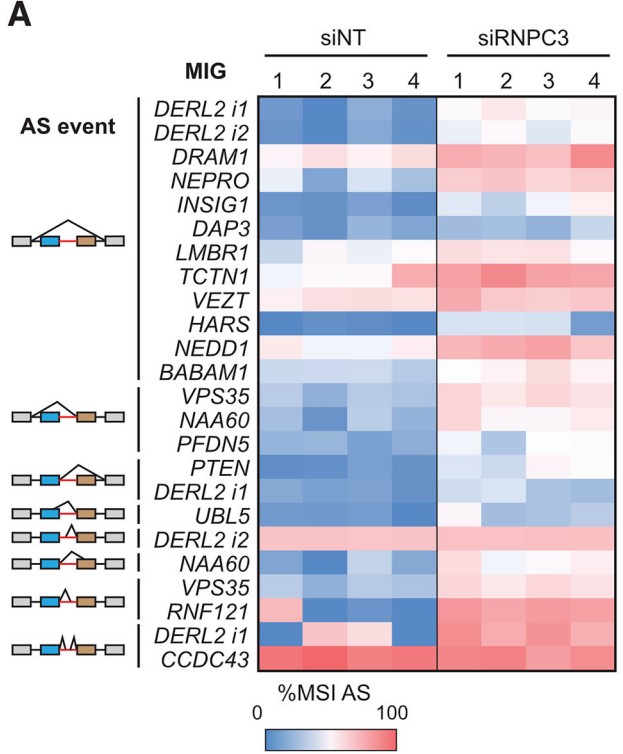

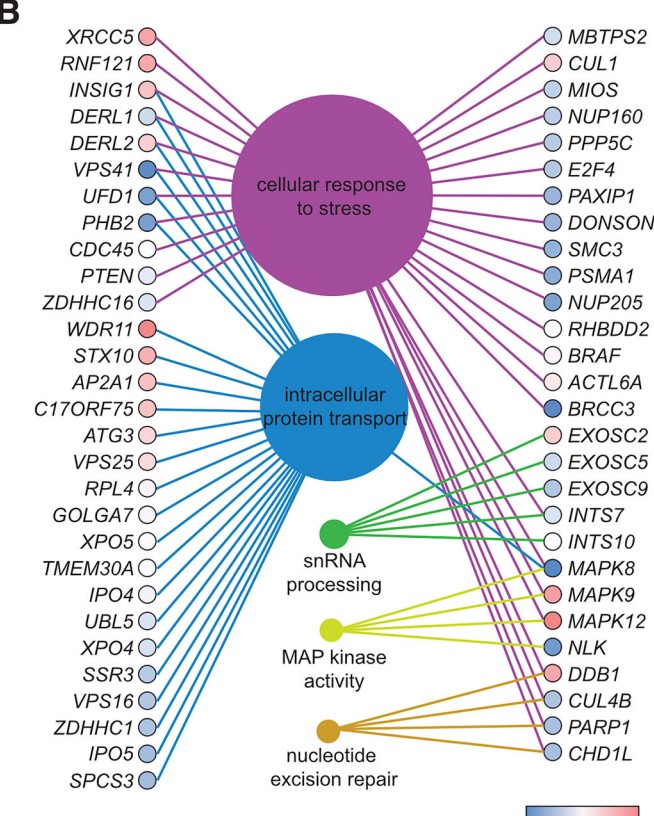

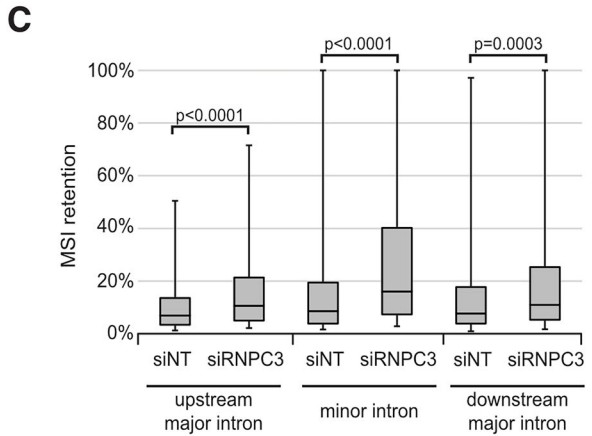

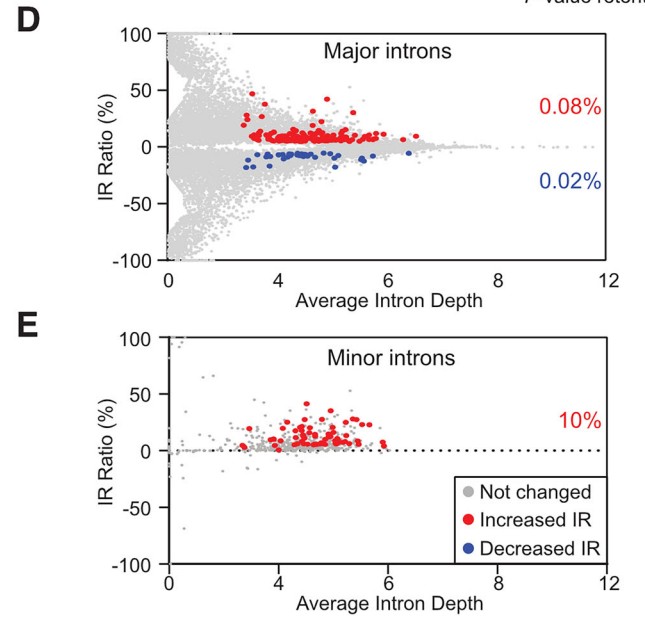

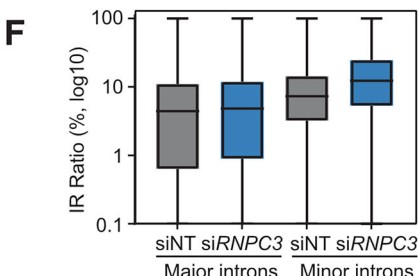

◀ **Figure EV5. RNAseq analysis of A549 cells treated with siRNPC3 for 72 h reveals defects in MIG splicing.**

(A) Heatmap of RNAseq identified MIG AS events. (B) Selected enriched GO terms of MIGs with significant intron retention. Significance was assessed by Welch's t test. (C) Box plot of intron retention at minor introns and their upstream and downstream major introns. Significance was assessed by Mann–Whitney U test, n = 4 replicate RNAseq samples/genotype. (D, E) Transcriptome-wide IRFinder analysis showing median IR for all introns. (D) Major introns showing significantly upregulated IR (solid red circles) and significantly downregulated IR (solid blue circles). (E) Minor introns showing significantly upregulated IR (solid red circles). For an intron to be called as significantly retained in siRNPC3 samples, we required the following, FDR JQ 0.05, 100% coverage of the intron in siRNPC3 samples, 10% IRratio in siRNPC3 and 5% IRratio over NT samples. Likewise, for an intron to be significantly retained in NT samples, FDR JQ 0.05, 100% coverage across the intron in NT samples, 10% IRratio in NT cells and 5% IRratio over siRNPC3 samples. (F) Box plot of intron retention (% IRratio log10) across 234,057 major and 599 minor introns. Significance was assessed by paired t test.

**A**

DE minor intron genes

siNT    siRNPC3

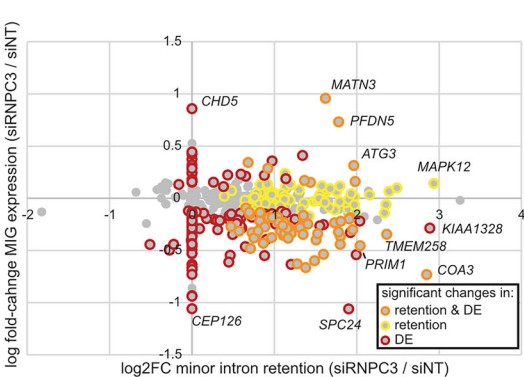

**B**

◀ **Figure EV6. A549 cells treated for 72 h with *RNPC3* siRNA exhibit differential expression of genes and MIGs that are enriched in cancer.**

(A) Heatmap of all identified differentially expressed MIGs, showing logCPM corrected for replicate and scaled for each gene. (B) Log-fold change plot of all MIGs highlighting those that exhibit significant changes in intron retention, differential expression or both when *RNPC3* is knocked down.

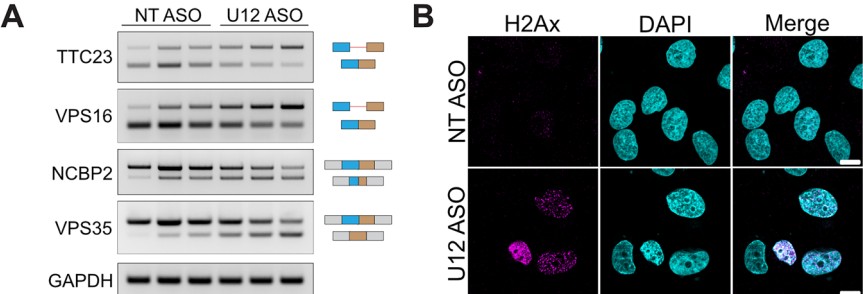

**Figure EV7.** **A549 cells treated for 72 h with U12 ASO exhibit aberrant MIG splicing and markers of DNA damage.**

(**A**) RT-PCR analysis of MIG splicing changes in A549 cells after 72 h treatment with 10 nM ASO. Schematic depictions of the obtained amplicons are shown on the right with the minor intron in red and the upstream and downstream exons coloured blue and orange, respectively. Exons not separated by a minor intron are grey. (**B**) Representative images of γH2Ax (Ser139) staining in A549 cells 72 h after 10 nM ASO treatment. Scale bar is 10 μm.

