## [Peer Review File · EMBO Reports]

Inhibition of the minor spliceosome restricts the growth of a broad spectrum of cancers

Karen Doggett, Kimberly Morgan, Anouk Olthof, Stephen Mieruszynski, Benjamin Williams, Alexandra Garnham, Michael Milevskiy, Lachlan Whitehead, Janine Coates, Michael Buchert, Robert O'Donoghue, Thomas Hall, Tracy Putoczki, Matthias Ernst, Kate Sutherland, Rahul Kanadia, and Joan Heath

Corresponding author(s): Karen Doggett (doggett.k@wehi.edu.au)

Review Timeline:

Submission Date:	14th Feb 25
Editorial Decision:	18th Feb 25
Revision Received:	24th Apr 25
Editorial Decision:	20th May 25
Revision Received:	1st Jun 25
Additional Correspondence from the Editor:	6th Jun 25
Author's Response:	12th Jun 25
Accepted:	16th Jun 25

Editor: Esther Schnapp

Transaction Report: This manuscript was transferred to EMBO reports following peer review at EMBO Molecular Medicine.

Referee #1 (Remarks for Author):

This manuscript is focused on the role of minor splicing in cancer progression and its potential use as a therapeutic vulnerability. This is an attractive idea, yet is not particularly novel, since this has been proposed for prostate cancer (PCa) (see Novelty below the in Specific comments section).

The authors use a variety of experimental systems, including zebrafish, mouse and human cancer models. First, Doggett and co-authors investigate the impact of *rnpc3*, which encodes a 65kDa RNA-binding protein of the minor spliceosome, in a zebrafish model of hepatocellular carcinoma (HCC) driven by an inducible mutant *krasG12V* transgene. These experiments revealed a vulnerability of *krasG12V*-expressing hepatocytes to a reduction in RNPC3. The authors also showed that *rnpc3* heterozygosity also results in a decreased number of hepatocytes in S-phase, concomitantly with an increase in cell death. The authors went on to show that *rnpc3* heterozygosity in zebrafish is accompanied by an increase in DNA damage and activation of Tp53-dependent transcriptional program, which leads to cell cycle arrest, senescence and apoptosis. This seems to be the most original part of this study.

Next, Doggett and colleagues expand their findings to test the effect of a reduced doses of RNPC3 to mouse models of lung adenocarcinoma and gastric adenoma. Along the lines of their findings in zebrafish, the authors found that heterozygous loss of *Rnpc3* reduces tumour burden in both mouse models.

The authors next moved to yet another experimental system and investigate the effects of disrupting the *Rnpc3* locus in AML cells, as a way of comparing the therapeutic vulnerability in solid tumour models with hyperproliferative blood cancer. They extend their observations to test the effect of RNPC3 in the growth and minor splicing of human A549 cells. In another twist of event, the authors finally investigate changes in minor splicing, but this time they do so using antisense oligonucleotides targeting U12 snRNA (see point 2, related to controls in Specific comments).

There is a lot of work presented here, and this manuscript certainly has potential. The authors put forward an attractive model where interfering with the minor spliceosome may offer a therapeutic vulnerability in cancer

Specific comments

Novelty

1- Overall, this is an interesting study, yet not entirely novel. There is an influential paper published in Mol Cell, that shows that minor intron splicing is a therapeutic vulnerability have nominated MiS as a vulnerability for lethal PCa and potentially other cancers (PMID: 37295433; N&V PMID: 37327771). Somehow, the authors decided not to reference this paper, which so far, is the most compelling evidence for a role of minor splicing in cancer. The authors should clearly explain the degree of novelty of their study considering this previous paper.

2- It would be an important control to have a knock-down (and/or heterozygosity) of another component of the minor spliceosome, to rule out other gene-specific effects that could operate independently of defects in minor splicing. This could also be important taking into consideration effects seen in A549 cells in Fig 5C, where RNCP3 only affect splicing of 40% of minor introns.

- The fragmented nature of this manuscript could be considered positive, due to the exploration of different cancer settings; however, it is not clear why the authors investigate global effects of RNCP3 in human A549 cells and not in any of the mouse models, used in previous figures.

Referee #2 (Comments on Novelty/Model System for Author):

The models are thorough and the relevance is here - there are some experimental/mechanistic issues to address.

Referee #2 (Remarks for Author):

In this manuscript by Doggett et al., the authors investigate the role of the RNCP3 gene, a component of the minor spliceosome (MiS), in cancer using zebrafish, mouse, and human models. Through both in vivo and in vitro approaches, they demonstrate that reduced

RNPC3 activity hinders cancer cell proliferation and diminishes tumor burden. Using the KrasG12V allele to drive cancer in zebrafish, they show that *rnpc3* heterozygosity creates vulnerabilities in KrasG12V-expressing cells and induces DNA damage. These findings are further validated through several mouse models of cancer, reinforcing the dependency on RNPC3.

The data presented is clean, and the findings are compelling. The manuscript provides a strong argument for the therapeutic potential of targeting the minor spliceosome in cancer, particularly through RNPC3 disruption. However, some areas would benefit from clarification and additional investigation, as these points are critical for fully developing the story. One major issue is the claim of senescence. The data provided is insufficient to conclusively state that the affected cells are senescent. Since senescence is a pleiotropic phenotype, further investigation is needed to better understand this response to RNPC3 depletion. Additionally, while the findings are convincing, the manuscript lacks mechanistic connections for several aspects of the story. For example, if RNPC3 loss causes DNA damage and activates tp53, why doesn't the DNA damage response resolve the damage? Are the same minor intron genes affected across the different tumor models, or does each model show disruption in a unique set of genes? Do RNPC3-deficient tumor models exhibit DNA damage? Is cell death or senescence occurring in these models? RNAseq analysis across tumor models would help elucidate whether the same or different minor intron genes (MIGs) are affected. Addressing these questions would help strengthen the narrative and provide a more complete understanding of the mechanisms involved. In addition to these key questions, there are several additional points that should be addressed.

Major:

1. How does minor splicing deficiency cause DNA damage?

The authors show DNA damage markers (e.g., γ H2AX) in RNPC3-deficient cells but do not clarify whether this involves replication stress, transcription-replication conflicts, or other mechanisms. The staining in Figure 6 suggests double-strand breaks but does not exclude single-strand breaks.

2. Why is RNPC3 loss more impactful in certain tissue regions?

In the gut, RNPC3 loss has a stronger effect in the corpus than the antrum. The basis for this regional specificity is unclear.

3. Does Ras signaling in physiological settings depend on MIGs?

While the authors focus on KrasG12V models, understanding whether MIGs are essential for wild-type Ras signaling in normal tissues would strengthen the relevance of their findings.

4. Are the same MIGs affected across tumor models?

Do zebrafish, mouse, and human models share common MIG vulnerabilities, or does the splicing disruption target distinct genes in each context?

5. What is the fate of RNPC3-deficient cells?

Figures 6H and 6I suggest reduced proliferation, increased G2 phase, and no change in G1 phase. It remains unclear whether these cells progress through mitosis, enter senescence, or undergo apoptosis.

6. Why does the DNA damage response fail to repair the lesions?

If RNPC3 loss activates tp53, why is DNA damage unresolved in these cells? This mechanistic gap needs exploration.

7. Is there evidence of DNA damage in tumors reduced by RNPC3 loss?

The link between reduced tumor burden and DNA damage remains speculative.

Demonstrating this connection would strengthen the therapeutic rationale.

8. Comparative analysis with U6atac knockdown (KD):

Previous work (e.g., Augspach et al., Molecular Cell, 2023) shows that U6atac KD disrupts MiS activity, leading to cell-cycle arrest and reduced tumor burden. Comparing RNPC3 inhibition to U6atac KD would clarify whether RNPC3 offers unique therapeutic advantages.

Minor:

1. Abstract:

The abstract is too general. It should specify the key findings, including the role of RNPC3 in KrasG12V-driven cancers and the evidence for DNA damage and reduced proliferation.

2. Introduction:

Provide a detailed explanation of MiS in cancer and how splicing defects could lead to DNA damage, replication stress, and other anti-cancer phenotypes.

3. Figures and Legends:

Include more details, such as:

- o Fish age in Figure 1.
- o Clarify the arrow in Figure 1E (presumably pointing to the same structure as 1C).
- o Label age in EV1B.
- o Specify sample types and genetic backgrounds in Figure 2C-K.

4. Quantify EdU-positive nuclei and apoptosis rates:

Compare WT and HET larvae in zebrafish models for more robust conclusions.

5. Avoid possessive language ("our model"):

Replace terms like "our zebrafish HCC model" with "the zebrafish HCC model."

6. RNAseq of *rnpc3*^{-/-}; *tp53*^{-/-}; Ras livers:

A broader transcriptomic analysis would be more informative than the selected gene panels in Figure 2D-K.

7. Address absence of DNA damage in WT RNPC3 Ras models:

RAS overexpression typically induces replication stress and DNA damage. Why is this absent in WT RNPC3 models? Is it time-point dependent?

8. EV Figure 6A/B:

Suggests co-regulation of major and minor introns in certain genes. This deserves further investigation, particularly in RNPC3 KD cells.

9. Minor splicing inhibitors:

Evaluate inhibitors like E7107 or H3B-8800, which have shown efficacy in hematologic malignancies, as a potential comparison for RNPC3 inhibition.

10. Clarify tp53 expression in tp53 mutant zebrafish:

The authors state that tp53 gene levels remain unaffected in tp53m/m fish. Clarifying whether this is true at the protein level would help interpret the results.

11. Statistical analyses:

Include statistical comparisons in Figures 2C and 6H/I to validate observed differences in phase distribution and gene expression.

12. Figure Consistency:

Standardize y-axis scaling across figures (e.g., confluency data in Figures 5B and 6J).

13. Total cell count in Figure 6J:

Verify whether changes in cell-cycle phases correspond to an overall reduction in cell number.

Referee #3 (Remarks for Author):

In this study, Doggett et al. investigated the role of minor splicing disruption in cancer inhibition. Specifically, the authors reduced RNPC3, a protein component unique to the minor spliceosome, in a variety of zebrafish, mouse and human cancer models and showed that *rnpc3/Rnpc3/RNPC3* loss reduced tumour burden in all these cancer settings. They further showed that reduction of *rnpc3/Rnpc3/RNPC3* increased DNA damage and activated a Tp53 DNA damage response in the zebrafish liver cancer model and in the human lung cancer model, induced missplicing of minor introns in the mouse AML and human lung cancer models, and induced cell cycle arrest, senescence and apoptosis in the human lung cancer model. This study suggests that minor splicing inhibition may provide a viable solution for a variety of cancers.

The manuscript is well written. The experiments were well executed and the data clearly presented. Other than a previous study showing the inhibitory effect of reducing a unique RNA component (i.e., U6atac snRNA) of the minor spliceosome on prostate cancer cell growth, this is the first study to show the inhibitory effect of reducing a unique protein component of the minor spliceosome on tumour burden in a number of cancer settings. Therefore, this study provides new insights and should have a big impact on the cancer research field. I think the manuscript will be improved if the authors can address the following points.

Major points:

1. The authors provided rigorous data to convincingly show that heterozygous loss of *rnpc3* reduced hepatocyte hyperplasia in a zebrafish model of hepatocellular carcinoma. However, they did not show any evidence that *rnpc3* reduction led to missplicing of minor introns in this zebrafish cancer model. To strengthen the main claim of this study, the authors need to perform RT-PCR analyses on a few target minor introns to confirm that ~40% reduction of *rnpc3* is able to induce minor intron retention/alternative splicing.

2. The authors need to provide evidence that mice with a *Rnpc3*^{+/-} genotype indeed have reduced levels of *Rnpc3* in the mouse models of lung and gastric cancers. It is well known that negative autoregulatory feedback is prevalent in splicing regulation. Some splicing factors can use a negative autoregulatory feedback to maintain constant levels when their own transcription is reduced. Therefore, it is important for the authors to confirm that *Rnpc3* levels are reduced in the heterozygous *Rnpc3*^{+/-} mice. In addition, the authors need to perform RT-PCR analyses on a few target minor introns to confirm that minor introns are retained/alternatively spliced in *Rnpc3*^{+/-} mice.

Minor points:

1. On page 15, the authors wrote "133 (34.2%) exhibited significantly elevated IR" in the last paragraph. However, on page 16, the authors wrote "60 (10%) exhibited increased IR" in the last paragraph. These two percentage numbers are so different that the authors need to clearly explain this discrepancy. Why didn't the authors use their own RNA-seq data for the transcriptome-wide analysis with IRFinder?

2. Figs. 4E and 4F, mice with *Rnpc3* lox/lox;UBC-CreERT2 AML cells survived longer than mice with *Rnpc3* lox/-;UBC-CreERT2 AML cells (29 d vs. 22 d). This result seems to contradict with the authors' conclusion. Can the authors provide an explanation for this?

3. Is RNPC3 upregulated in human lung cancer?
4. Page 17 third paragraph, "Fig. 7A, B" should be "Fig. EV7A, B".
5. Page 23 first paragraph, "Fig. 7L" should be "Fig. 6K".
6. In the Abstract, "correct expression of 755 genes" should be "correct splicing of 755 minor introns", because some genes have more than one minor intron.

Referee #4 (Remarks for Author):

The manuscript titled "Minor splicing is a therapeutically viable target for the treatment of a broad spectrum of cancers" by Doggett et al. presents findings regarding the potential therapeutic targeting of the minor spliceosome-specific component, RNPC3, in cancer treatment. The study highlights how targeting RNPC3 can impair tumor growth through mechanisms involving splicing disruption, DNA damage, and activation of TP53-dependent pathways. There are several concerns regarding the experimental design and data interpretation that need to be addressed:

Specific Comments:

1. The experiments shown in Fig. 1D-E should be performed using a *kras* wild-type (WT) setting to confirm the conclusions drawn.
2. The statement that "cell cycle arrest and apoptosis are strongly suggestive of activation of the tumor suppressor functions of TP53" (page 7) is overly simplistic. Numerous regulators control cell cycle and apoptosis. A more comprehensive or genome-wide analysis is necessary to identify the key regulators involved in this context.
3. The observation that *kras* WT liver samples do not express *tp53* (Fig. 2A) raises questions about the zebrafish model's relevance. Does this imply that WT liver lacks *tp53* expression in zebrafish?

If so, this model may not be ideal for studying TP53-associated DNA damage responses, given the significant differences from human and mouse models.

4. Fig. 2B does not appear to be cited in the manuscript text.

5. To support the conclusions, global gene expression analyses are required to examine the roles of cell cycle, apoptosis, and DNA damage response pathways.
6. Protein expression levels of 65K (encoded by Rnpc3) and pERK in the AHH or adenoma lesion in Fig. 3B seem higher than those of Rnpc^{+/+} counterpart in Fig. 3A. The differences shown in Fig. 3 are, if any, very subtle, which may significantly harm the authors' narrative.
7. The authors stated that "Our findings above, where we employed three different solid tumour models, suggest that hyperproliferative cancers require efficient minor splicing to proliferate and survive" (in page 12). However, (minor) splicing was not analyzed at all.
8. The critical comparison in Fig. 4E-F is the survival difference between Rnpc3 lox/lox and Rnpc3^{+/+} mice treated with TMX. This data is missing. Additionally, the manuscript lacks evidence of AML development in recipients (At least Mac-1 and c-kit staining and HE staining of blood/bone marrow smear should be provided for the diagnosis).
9. The manuscript states that "aberrant splicing of MIGs leads to DNA damage and the stabilization of TP53" (page 19). However, TP53 protein stability was not analyzed, which undermines this conclusion.
10. The current title is overly generalized, as the study focuses solely on RNPC3. Furthermore, the findings contradict the known loss-of-function mutations in ZRSR2 in myeloid leukemia, which suggest opposite roles for the minor spliceosome in hematologic malignancies. The title should be revised, and the conclusions should be appropriately tempered to reflect the study's scope.

Dear Karen,

Thank you for the transfer of your manuscript to EMBO reports, and for your revision plan.

I discussed your plan with referee 1, who overall agrees with it, while noting that the many different experimental systems used are not ideal.

Given the overall supportive response, I would like to invite you to revise your manuscript along the lines you suggest, with the understanding that the referee concerns must be fully addressed and their suggestions taken on board. Please address all referee concerns in a complete point-by-point response. Acceptance of the manuscript will depend on a positive outcome of a second round of review. It is EMBO reports policy to allow a single round of major revision only and acceptance or rejection of the manuscript will therefore depend on the completeness of your responses included in the next, final version of the manuscript.

We realize that it is difficult to revise to a specific deadline. In the interest of protecting the conceptual advance provided by the work, we recommend a revision within 2 months (21st April 2025). Please discuss the revision progress ahead of this time with the editor if you require more time to complete the revisions.

- 1) A data availability section providing access to data deposited in public databases is missing. If you have not deposited any data, please add a sentence to the data availability section that explains that.
- 2) Your manuscript contains statistics and error bars based on $n=2$. Please use scatter blots in these cases. No statistics should be calculated if $n=2$.

3) We replaced Supplementary Information with Expanded View (EV) Figures and Tables that are collapsible/expandable online. A maximum of 7 EV Figures can be typeset. EV Figures should be cited as 'Figure EV1, Figure EV2' etc... in the text and their respective legends should be included in the main text after the legends of regular figures.

5) a complete author checklist, which you can download from our author guidelines <https://www.embopress.org/page/journal/14693178/authorguide>. Please insert information in the checklist that is also reflected in the manuscript. The completed author checklist will also be part of the RPF.

6) Please note that all corresponding authors are required to supply an ORCID ID for their name upon submission of a revised manuscript (<https://orcid.org/>). Please find instructions on how to link your ORCID ID to your account in our manuscript tracking system in our Author guidelines <https://www.embopress.org/page/journal/14693178/authorguide#authorshipguidelines>

7) Before submitting your revision, primary datasets produced in this study need to be deposited in an appropriate public database (see <https://www.embopress.org/page/journal/14693178/authorguide#datadeposition>). Please remember to provide a

reviewer password if the datasets are not yet public. The accession numbers and database should be listed in a formal "Data Availability" section placed after Materials & Method (see also <https://www.embopress.org/page/journal/14693178/authorguide#datadeposition>). Please note that the Data Availability Section is restricted to new primary data that are part of this study. * Note - All links should resolve to a page where the data can be accessed. *

- the name of the statistical test used to generate error bars and P values,
- the number (n) of independent experiments (please specify technical or biological replicates) underlying each data point,
- the nature of the bars and error bars (s.d., s.e.m.),
- If the data are obtained from n Program fragment delivered error `Can't locate object method "less" via package "than" (perhaps you forgot to load "than"?) at //ejpvfs23/sites23b/embor_www/letters/embor_decision_revise_and_review.txt line 56.' 2, use scatter blots showing the individual data points.

12) All Materials and Methods need to be described in the main text using our 'Structured Methods' format, which is required for all research articles. According to this format, the Methods section includes a separate Reagents and Tools Table (listing key reagents, experimental models, software and relevant equipment and including their sources and relevant identifiers) and a Methods and Protocols section describing the methods using a step-by-step protocol format. The aim is to facilitate adoption of the methodologies across labs. More information on how to adhere to this format as well as a downloadable template (.docx) for the Reagents and Tools Table can be found in our author guidelines: <https://www.embopress.org/page/journal/14693178/authorguide#structuredmethods>.

An example of a Method paper with Structured Methods can be found here: <https://www.embopress.org/doi/full/10.1038/s44320-024-00037-6#sec-4>

You are able to opt out of this by letting the editorial office know (emboreports@embo.org). If you do opt out, the Review Process File link will point to the following statement: "No Review Process File is available with this article, as the authors have

chosen not to make the review process public in this case."

I look forward to seeing a revised form of your manuscript when it is ready.

Best wishes,
Esther

EMBOR-2025-61344-T**Response to Reviewers' comments from EMBO Molecular Medicine**

We appreciate the detailed and constructive feedback from the 4 reviewers. All express the view that our study is interesting and based on rigorous and convincing experimentation. They do, however, have many suggestions for its improvement, most of which boil down to a thirst for more detail. We agree that the points raised are interesting and relevant; however, most of the comments do not challenge our interpretation of the data or the conclusions we reach. In this "Response to Reviewers" document, we address in detail the four major areas of concern, including those that were raised by more than one reviewer. We have also responded to all the minor comments. Our responses/changes are highlighted in blue text throughout this document and in the manuscript proper. We hope our responses meet the expectations of the reviewers and convince them to recommend acceptance of our revised paper.

1. NOVELTY

Reviewers 1 and 2 are correct to point out that there is a significant paper in the literature that indicates an important role for minor splicing inhibition in therapy-resistant prostate cancer. Augspach and colleagues used siRNA knockdown strategies to demonstrate that minor splicing is a therapeutic vulnerability of prostate cancer cell lines and organoids *in vitro* and raised the possibility that this may be relevant to other cancers. We acknowledge and regret our oversight at not citing this substantial paper, especially since our study, which explores the same topic in a broad spectrum of *in vivo* cancer models, complements the Augspach paper very well. **We have corrected our omission by citing the significance of the Augsprach paper prominently in our revised manuscript (page 5, second paragraph).**

2. ANALYSIS OF MINOR SPLICING IN *IN VIVO* TUMOUR MODELS

Reviewers 2 and 3 are disappointed with our lack of evidence that *Rnpc3* heterozygosity generates aberrant minor splicing in the *in vivo* mouse models we used. We understand their opinion that this would have strengthened the paper.

Unfortunately, for us to address this issue now, we would first have to re-derive our mouse cancer models, which is not a trivial task. This is because part of our Institute's response to the prolonged COVID-related lockdowns in Melbourne was to freeze down the sperm of as many mouse lines as possible, which included all the mouse models described in this paper. While we managed to harvest and fix tissues for histological and immunocytochemical analyses prior to sperm-freezing, no samples were preserved for molecular/splicing studies. Unfortunately, it would likely take at least 18 months of breeding and experimentation to perform the transcriptome-wide analysis of minor splicing required to answer this question.

Our alternative strategy for the transcriptome-wide analysis was to employ an *in vitro* model where we could reduce *RNPC3* expression by >50%, which offered an advantage over the genetic models. We used siRNA-knockdown technology and a well characterized human lung adenocarcinoma-derived cell line, A549, to knockdown *RNPC3* expression by 70%. With this level of *RNPC3* depletion, we generated robust RNA-seq data demonstrating differential mis-splicing of minor introns (Fig. 5B-G) as well as molecular insights linking disrupted minor splicing to the impaired growth and survival of cancer cells (Figs. EV4-6).

Are all MIGs affected?

Based on previous work from ourselves and others, reducing the expression of minor spliceosome components (including *RNPC3*, *PDCD7* and U12 snRNA) does not disrupt the processing of all MIG pre-mRNAs, nor does it affect them all in the same way. For example, some nascent transcripts retain their minor introns and remain in the nucleus, while others are alternatively spliced, giving rise to numerous patterns of exon-exon and exon-intron structure, some of which are tissue-specific (Olthof et al 2019 BMC genomics 20:686). Of the various isoforms, some are transported into the cytoplasm to be translated and potentially degraded by nonsense mediated decay (NMD). Also interesting is that upon disruption of minor splicing, all MIG transcripts appear to have the potential to be substrates for intron retention and aberrant splicing, not just those that encode proteins with a function in DNA damage and cancer. Determining the individual fate(s) of every MIG transcript in response to inhibition of minor splicing would require a great deal of further work that is beyond the scope of this study.

3. MECHANISTIC INSIGHTS – ROLE OF ABERRANT PRE-mRNAs, DNA DAMAGE AND TP53

Reviewers 2 and 4 are interested in the steps involved in generating dsDNA damage and TP53 activation in A549 cells in response to disrupted *RNPC3* or *U12 snRNA* expression. We postulate that the dsDNA damage we observe is caused by de-regulation of physiological DNA replication due to replicative stress (RS). RS generally goes hand-in-hand with oncogene activation and is often a product of transcription-replication conflicts between the large molecular machineries that drive DNA replication, transcription and pre-mRNA processing as they compete for access to the genome (Kotsantis *et al.*, Cancer Discov 2018 8(537-555). In our study, we detected upregulation of γ H2AX in sections of zebrafish *kras*^{G12V} livers from *Rnpc3* HET larvae (Fig. 2L) and 53BP1 and γ H2AX activation in the nuclei of A549 cells treated with an ASO to U12 snRNA (Fig. 6D and Fig. EV7B). Both these markers are definitive indicators of RS (Kotsantis *et al* 2018).

4. CONTROLS

We identified three instances where the reviewers highlight a need for additional controls.

Reviewer 1, point 2: requests a knock-down (and/or heterozygosity) of another component of the minor spliceosome, to rule out other gene-specific effects (of *RNPC3*) that could operate independently of defects in minor splicing.

Response: In our revised paper, we demonstrate that reduced expression of two unique components of the minor spliceosome, *RNPC3* and U12 snRNA, impact similarly on A549 cells to disrupt minor splicing and reduce their growth (Fig. 5, Fig. EV4, Fig. EV5 and Fig. 6A & J). In data not included in our original manuscript, we now show that knockdown of *PDCD7* encoding the unique 59K protein component of the minor spliceosome, produces essentially the same results in A549 cells as knockdown of *RNPC3* and *U12 snRNA*. These data diminish the possibility that *RNPC3* knockdown produces gene-specific effects that operate independently of defects in minor splicing (see **Fig. EV4 A, C**).

Reviewer 1 specific point 4: Quantify EdU-positive nuclei and apoptosis rates: Compare WT and HET larvae in zebrafish models for more robust conclusions.

Reviewer 4 specific point 1: The experiments shown in Figure 1D-E should be performed using a *kras* wild-type (WT) setting to confirm the conclusions drawn.

Response: We think these two comments are referring to the same thing, so we address them both here. As described on page 6 of our paper, and in Fig. 1 B, C, we used the *2-CLiP* line on a *kras*^{G12V(+/+)} negative background as a control for our experiments because in this line, the hepatocytes express only one transgene, dsRed, and no oncogenes. Using the *2-CLiP* line, we collected fluorescent images of livers from *kras*^{G12V(+/+)} negative controls and found that the volume of the liver did not change whether the hepatocytes were WT or HET for *rnpc3*. Thus, normal liver volume during development was not affected by *rnpc3* genotype (Fig. 1B, C) and so we did not perform assays of EdU incorporation and AnnexinV accumulation in this setting.

In the experiments we performed in a cancer setting (dox-induced *kras*^{G12V(T/+)} expression), the only variable was *rnpc3* genotype, and in this case we found that *rnpc3* heterozygosity markedly reduced liver enlargement, prompting us determine whether a decrease in cell proliferation or an increase in apoptosis were involved. To do this, we measured EdU incorporation and AnnexinV accumulation and found that EdU incorporation was decreased and AnnexinV accumulation increased as a result of *rnpc3* heterozygosity. If we have understood the reviewers' suggestions correctly, they think we should also have performed the Edu/AnnexinV analyses on the *2-CLiP* larvae. Regrettably, we cannot see any benefit in doing this because, in our opinion, the most reliable control for the Edu/AnnexinV experiments in the HCC (*kras*^{G12V(T/+)}) context is when the only experimental variable is the *rnpc3* genotype

(WT or HET). We are therefore unsure how performing the Edu/AnnexinV analyses in the *kras*^{G12V}-negative (+/+) 2-CLiP line could have the potential to influence our interpretation of the cancer data.

Reviewer 4 specific point 8: The critical comparison in Figure 4E-F is the survival difference between *Rnpc3* lox/lox and *Rnpc3*+/+ mice treated with TMX. This data is missing. Additionally, the manuscript lacks evidence of AML development in recipients (At least Mac-1 and c-kit staining and HE staining of blood/bone marrow smear should be provided for the diagnosis).

Response:

To increase clarity, we provide AML data from all our tertiary transplant experiments in the Tables below (and in Source data for Fig. 4).

First, we performed the comparison requested by Reviewer 4 (*Rnpc3*+/+ versus *Rnpc3* lox/lox) in the presence of TMX. As expected, the presence of TMX had no impact on disease latency in mice transplanted with *Rnpc3*+/+ AML cells (50% survival, 22d). Recipients of *Rnpc3* lox/lox cells had a median survival of 19d in the absence of TMX, and 29d in the presence of TMX (+10d survival with TMX or +7d survival benefit to *Rnpc3*+/+). These data show that when *Rnpc3* expression is reduced in AML cells, there is an increase in the disease latency in the recipient mice. We also compared the survival of *Rnpc3*+/+ versus *Rnpc3* lox/- AML in the presence and absence of TMX (green rows) and again there was an increase in disease latency when *Rnpc3* expression was reduced (+12d in both comparisons). These data are expressed as Kaplan Meier plots in Fig. 4E, F, and Fig. EV3C.

Rnpc3 genotype	n	TMX (yes/no)	50% survival	Difference in disease latency with TMX
Rnpc3 +/+	6	no	22d	N/A
Rnpc3 +/+	3	yes	22d	0 (Fig. EV3 panel C)
Rnpc3 lox/lox	3	no	19d	N/A
Rnpc3 lox/lox	6	yes	29d	+10d (new Fig. 4E)
Rnpc3 lox/-	11	no	22d	N/A
Rnpc3 lox/-	15	yes	34	+12d (Fig. 4F)
Rnpc3 +/+	6	no	22d	N/A
Rnpc3 +/+	3	yes	22d	0
Rnpc3 lox/lox	6	yes	29d	+7d compared to Rnpc3 +/+ cells
Rnpc3 lox/-	15	yes	34d	+12d

Regarding our original version of Fig. 4, Reviewer 3 asked us to explain why the survival of the mice receiving *Rnpc3* lox/lox cells (+TMX) had a longer median survival than that of mice receiving *Rnpc3* lox/- cells (+TMX). In our original version of Fig. 4E, the total number of mice receiving AML cells with the *Rnpc3* lox/lox genotype was n=12 (no TMX) and n=15 (+ TMX) and the data were collected over two independent experiments. When we separated these mice into two cohorts, we

found that recombining the *Rnpc3* lox/lox locus by TMX in the first cohort of mice increased disease latency of the mice from 19d to 29d (+10d) as shown below, and in new Fig. 4E.

AML genotype FIRST COHORT	N	TMX (yes/no)	50% survival	Difference in longevity with TMX
Rnpc3 lox/lox	3	no	19	
Rnpc3 lox/lox	6	yes	29d	+10d

However, the mice in the second cohort had a much longer disease latency in the absence of TMX (**29d***) than the first cohort (see below, pink rows & Source data Fig. 4E), indicating biological variation between these two experiments. Notwithstanding this, the net increase in disease latency in the presence of TMX was very consistent for the two cohorts of mice (10d vs 11d).

AML genotype SECOND COHORT	N	TMX (yes/no)	50% survival	Difference in longevity with TMX
Rnpc3 lox/lox	9	no	29d*	
Rnpc3 lox/lox	9	yes	40d	+11d

Returning to Reviewer 3's original question, in our previous version of Fig. 4E, where the pooled *Rnpc3* lox/lox data were plotted on a single curve, it appeared that the 50% disease latency was shorter for mice receiving *Rnpc3* lox/- cells (+TMX), compared to mice receiving the *Rnpc3* lox/lox cells (+TMX) because the survival plot for the mice receiving *Rnpc3* lox/lox cells (+TMX) is shifted to the right. However, the increase in median survival for the two cohorts of mice receiving lox/lox cells followed by TMX treatment (+10d and +11d) is essentially the same. So, for the purpose of clarity, we plotted only the data from the first cohort of mice in new Fig. 4E, and provide the raw data for replicate experiments in source data files.

Finally, we provide evidence of AML development using Mac-1 and Gr-1 staining (Fig. EV3E).

POINT-BY-POINT RESPONSES TO THE 4 REVIEWERS:

REVIEWER 1

Specific comments

Novelty

1. Overall, this is an interesting study, yet not entirely novel.

Response: Please see our response under 'NOVELTY' on page 1.

2. It would be an important control to **have a knock-down (and/or heterozygosity) of another component of the minor spliceosome, to rule out other gene-specific effects that could operate independently of defects in minor splicing.** This could also be important taking into consideration effects seen in A549 cells in Fig 5C, where RNCP3 only affected splicing of 40% of minor introns.

Response: As discussed under the heading 'Controls' on page 3 of this document, we now provide data showing very similar effects of *PDCD7* knockdown to *RNPC3* knockdown in A549 cells, arguing against the possibility that our observations are due to any gene-specific properties of *RNPC3* (Fig. EV4).

3. The fragmented nature of this manuscript could be considered positive, due to the exploration of different cancer settings; **however, it is not clear why the authors investigate global effects of RNPC3 in human A549 cells and not in any of the mouse models, used in previous figures.**

Response: The reasons for this are discussed in detail on page 2 of this document.

REVIEWER 2

Key comments

1. **One major issue is the claim of senescence.** The data provided is insufficient to conclusively state that the affected cells are senescent. **Since senescence is pleiotropic phenotype, further investigation is needed to better understand this response to RNPC3 depletion.**

Response: We appreciate there are multiple markers that can be used to indicate senescence. We chose to use a sensitive commercial kit with a senescence-specific fluorescent probe which could be readily quantitated by FACS (Fig. 6F, G). Also, in independent experiments (transcriptome-wide RNA-seq of A549 cells and candidate gene RT-qPCR in the zebrafish HCC model), we generated data consistent with stalled progress through the cell cycle and senescence, including up-regulation of both *cdkn1* and *cdkn2a/b*, encoding p21 and p14^{ARF}/p16^{INK4A}, respectively (Figure 2 H-I).

2. Additionally, while the findings are convincing, the manuscript lacks mechanistic connections for several aspects of the story. For example, if **RNPC3 loss causes**

DNA damage and activates tp53, why doesn't the DNA damage response resolve the damage?

Response: Some DNA damage may be repaired as a result of Tp53 activation, but in the context of constitutive *kras*^{G12V} expression, the high frequency of events causing DNA damage will be unrelenting, making it difficult to resolve. Also, activation of Tp53 protein induces the expression of pro-apoptosis genes to eliminate cells that are irrevocably damaged.

3. Are the same minor intron genes affected across the different tumor models, or does each model show disruption in a unique set of genes?

Response: See middle paragraph on page 2: "Are all MIGs affected?"

4. Do RNPC3-deficient tumor models exhibit DNA damage? Is cell death or senescence occurring in these models? RNA-seq analysis across tumor models would help elucidate whether the same or different minor intron genes (MIGs) are affected.

Addressing these questions would help strengthen the narrative and provide a more complete understanding of the mechanisms involved.

Response: We have demonstrated both DNA damage (Figure 2L) and cell death (Figure 1F) in confocal sections of zebrafish HCC livers from *Rnpc3* WT and HET larvae. We discuss why we did not generate RNA-seq data for the mouse models on page 1 of this document.

In addition to these key questions, there are several additional points that should be addressed.

5. How does minor splicing deficiency cause DNA damage?

The authors show DNA damage markers (e.g., γ H2AX) in RNPC3-deficient cells but do not clarify whether this involves replication stress, transcription-replication conflicts, or other mechanisms. The staining in Fig. 6 suggests double-strand breaks but does not exclude single-strand breaks.

Response: Please see our response under 'Mechanistic Insights' (bottom of Page 2). In short, we agree that dsDNA damage markers such as 53BP1 and γ H2AX are very strong indicators of RS and dsDNA damage, but we cannot preclude additional mechanisms such as ssDNA damage being involved.

6. Why is RNPC3 loss more impactful in certain tissue regions?

In the stomach, *Rnpc3* loss has a stronger effect in the corpus than the antrum. The basis for this regional specificity is unclear.

Response: The reduction in pERK staining at the active front of the corpus and antral adenomas in *Rnpc3* heterozygous mice is consistent with *Rnpc3* loss being more impactful on rapidly proliferating cells than quiescent cells. However, whether

this explains why the corpus adenomas are smaller than their antral counterparts is an open question, as we did not investigate whether markers of proliferating cells such as PCNA were more abundant in the corpus adenomas compared to antral adenomas, nor did we perform experiments to ascertain whether the epithelium in the corpus adenomas is more sensitive to DNA damage.

7. Does Ras signaling in physiological settings depend on MIGs?

While the authors focus on KrasG12V models, understanding whether MIGs are essential for wild-type Ras signaling in normal tissues would strengthen the relevance of their findings.

Response: We expect normal cells to be sensitive to very high levels of minor splicing inhibition, because many essential genes are MIGs. However, the degree of minor splicing inhibition that would be necessary to achieve this would have to be greater than that resulting from *Rnpc3* heterozygosity, as we have consistently shown (Markmiller et al. PNAS 2014; Doggett et al. RNA 2018) that there are no phenotypic differences between *Rnpc3* HETS and *Rnpc3* WTs during development and throughout adulthood.

Also, not all our models are driven by mutant KrasG12V (the gastric cancer model is STAT3 driven, and the AML model is driven by an MLL-ENL fusion protein), so we have not focused on Ras signalling alone. We think one of the strengths of our paper is its exploration of the role of minor splicing across a broad spectrum of cancer models.

8. Are the same MIGs affected across tumor models?

Do zebrafish, mouse, and human models share common MIG vulnerabilities, or does the splicing disruption target distinct genes in each context?

Response: See middle paragraph on page 2: “Are all MIGs affected?”

9. What is the fate of RNPC3-deficient cells?

Figures 6H and 6I suggest reduced proliferation, increased G2 phase, and no change in G1 phase. It remains unclear whether these cells progress through mitosis, enter senescence, or undergo apoptosis.

Response: The data shown in Figures 6H and 6I were obtained from a pool of unsynchronized human A549 cells treated with an ASO targeted to U12 snRNA. This referee queries whether the cells accumulating in G2 in response to U12 snRNA-deficiency progress through mitosis, enter senescence, or undergo apoptosis. Because the cells are unsynchronised, we think that all three of these processes are likely to be in motion, and it would be quite difficult to determine the proportion of cells in each. We think it would likely involve synchronising cells and performing experiments such as single cell or spatial transcriptomics. While a fascinating topic, such experiments are beyond the scope of our current manuscript.

6. Why does the DNA damage response fail to repair the lesions?

If RNPC3 loss activates tp53, why is DNA damage unresolved in these cells? This mechanistic gap needs exploration.

Response: Please see our response to the same point raised by Reviewer 2 on page 8 of this document.

7. Is there evidence of DNA damage in tumors reduced by RNPC3 loss?

The link between reduced tumor burden and DNA damage remains speculative. Demonstrating this connection would strengthen the therapeutic rationale.

Response: We did demonstrate increased DNA damage and reduced tumour volume in the zebrafish HCC model in response to *rnpc3* heterozygosity (Figure 2L). This is described on page 10 of the paper (second paragraph).

8. Comparative analysis with U6atac knockdown (KD):

Previous work (e.g., Augspach et al., Molecular Cell, 2023) shows that U6atac KD disrupts MiS activity, leading to cell-cycle arrest and reduced tumor burden. Comparing RNPC3 inhibition to U6atac KD would clarify whether RNPC3 offers unique therapeutic advantages.

Response: It is possible that depleting *U6atac* and *RNPC3* side-by-side in a variety of cancer cell systems could reveal distinct therapeutic advantages of one over the other. Similarly, knockdown of *PDCD7*, which also inhibits A549 cell growth and survival may offer an additional target for cancer therapeutics. However, determining the relative advantages of each of these players in terms of therapeutic benefit is a big piece of work and is beyond the scope of the current study.

Minor:

1. Abstract: **The abstract is too general.** It should specify the key findings, including the role of RNPC3 in KrasG12V-driven cancers and the evidence for DNA damage and reduced proliferation.

Response: In response to the suggestions of the reviewer, we have rewritten the second half of the abstract.

2. Introduction: **Provide a detailed explanation of MiS in cancer and how splicing defects could lead to DNA damage, replication stress, and other anti-cancer phenotypes.**

Response: We have followed the advice of the reviewer and addressed these points in the last 9 lines of the Introduction (page 5) and again in more detail in the Discussion (page 21, second paragraph).

3. Figures and Legends: Include more details, such as:

o Fish age in Figure 1.

Response: 7dpf is an abbreviation for 7 days post fertilization. This information is now included in the legend to Fig. 1.

o Clarify the arrow in Figure 1E (presumably pointing to the same structure as 1C).

Response: Arrows on the graphs indicate the data points that correspond to the representative images shown, this information is now included in the legend to Fig. 1.

o **Label age in EV1B.**

Response: Age is between 3.5-4 months of age; this information is now included in the **legend to Fig. EV1B.**

o **Specify sample types and genetic backgrounds in Figure 2C-K.**

Response: The livers are from zebrafish larvae aged 7 dpf. The genetic background is Tubingen TL and all our colonies have been in-bred over many generations. **These details have been added to the Methods section under Animal models, page 28.**

4. **Quantify EdU-positive nuclei and apoptosis rates:** Compare WT and HET larvae in zebrafish models for more robust conclusions.

Response: Please see our response in the CONTROLS section (middle of page 3).

5. **Avoid possessive language ("our model"):** Replace terms like "our zebrafish HCC model" with "the zebrafish HCC model."

Response: **We have made this correction throughout the manuscript.**

6. **RNAseq of *rnpc3*^{-/-};*tp53*^{-/-};*Ras* livers:** A broader transcriptomic analysis would be more informative than the selected gene panels in Fig. 2D-K.

Response: We agree that a transcriptome-wide approach provides more information than a candidate gene approach, but for reasons discussed in detail on pages 1-2, we decided to conduct our global RNAseq analysis in A549 cells (Fig. 5).

7. **Address absence of DNA damage in WT RNPC3 Ras models:**

RAS overexpression typically induces replication stress and DNA damage. Why is this absent in WT RNPC3 models? Is it time-point dependent?

Response: The data shown in Figure 2L, M were collected after the zebrafish larvae had been exposed to doxycycline for 5d (i.e. from 2-7dpf). Quantitation of these data at the 7dpf endpoint (Figure 2M) shows that on a WT *rnpc3* background γ H2AX-positive foci are rare, indicating that DNA damage is being tightly controlled by WT Tp53. However, in the context of *rnpc3* heterozygosity, the abundance of γ H2AX-positive foci is significantly higher, meaning that the increased replicative stress

caused by the accumulation of aberrant minor intron-containing transcripts is no longer adequately controlled by WT Tp53. This situation is exacerbated when both *tp53* alleles are mutated, such that no functionally active Tp53 is produced and DNA damage takes place unabated.

8. Fig. EV6A, B (now Fig EV5C, F)

Suggests co-regulation of major and minor introns in certain genes. This deserves further investigation, particularly in RNPC3 KD cells.

Response: We presented the data shown in Fig. EV5C, F to build on previous observations that *PDCD7* (59K) depletion in both HEK293 and A549 cells (Olthof *et al.*, NAR, 2021) causes mis-splicing of major introns in positions flanking minor introns, but generally not elsewhere. The mechanism underlying this phenomenon is based on the exon-definition model, which posits that components of the major and minor spliceosomes must interact with each other to form exon-bridging complexes that are critical for high-integrity splicing. The authors showed that when these interactions were compromised by *PDCD7* loss, the major splicing machinery circumvented the problem by binding to alternative (cryptic), major intron-like splice sites, almost always resulting in exon skipping, intron retention or otherwise aberrantly spliced transcripts. The authors also foreshadowed that the 65K protein encoded by *RNPC3* may also participate in these exon-bridging interactions, and this was reinforced in our current study.

Another explanation could be that loss of 65K, and potentially other minor spliceosome components (including snRNAs), could destroy the structural integrity and function of the minor spliceosome, leaving the major spliceosome to identify and assemble on non-canonical splice sites, in turn causing exon-skipping and the production of an array of alternatively spliced transcript isoforms. While this is fascinating biology, interrogating these potential interactions further is outside the scope of the current paper.

9. Minor splicing inhibitors: Evaluate inhibitors like E7107 or H3B-8800, which have shown efficacy in hematologic malignancies, as a potential comparison for RNPC3 inhibition.

Response: We are not sure exactly what the reviewer has in mind here. Certainly, we are very familiar with E7107, and its clinically relevant derivative H3B-8800, and indeed we used E7107 extensively as a positive control in splicing assays. However, these chemicals are pan-splicing inhibitors and disrupt splicing of both major and minor introns by interacting with components of the SF3B1 complex, which is a general splicing factor.

When compared side-by-side on gel RT-PCRs, E7107 produced alternatively spliced isoforms of both major and minor introns, and loss of virtually all correctly spliced pre-RNA, whereas *RNPC3* inhibition over the same time-course was selective for minor introns and resulted in discrete patterns of alternatively spliced transcripts

often accompanied by correctly spliced pre-RNA. We think the selectivity of minor intron splicing presents a great opportunity to develop small molecules capable of disrupting minor spliceosome activity for the purpose of developing cancer therapeutics, without the toxicity associated with inhibitors that also interfere with major splicing. However, since these observations are somewhat off-topic, we decided not to discuss them in our revised manuscript.

10. Clarify tp53 expression in tp53 mutant zebrafish:

The authors state that tp53 gene levels remain unaffected in tp53m/m fish. Clarifying whether this is true at the protein level would help interpret the results.

Response: We present data in Fig. 2E to show that tp53 mRNA levels remain unaffected in tp53m/m fish. Indeed, tp53 mRNA levels are not affected by *mnp3* genotype either. However, it is well known that *tp53* mRNA levels do not correlate well with the corresponding levels of Tp53 protein. This is because the regulation of tp53/TP53 expression occurs at the post-transcriptional level. In general, *tp53* mRNA is expressed in most cells constitutively, but the accumulation of Tp53 protein is circumvented by activation of *mdm2* and degradation by the ubiquitination pathway. Therefore, to clarify the status of the Tp53 protein in zebrafish livers from larvae of different genotypes, we carried out Western blot analysis (Fig. 2A). Our data show that only in the presence of the activated *kras*^{G12V} transgene (and *mnp3* WT), were small amounts of Tp53 protein detected (lane 3), suggesting that in these conditions, the cells were mildly stressed as a result of oncogene expression. However, when we removed one allele of *mnp3*, the level of Tp53 protein was markedly increased (lane 4), indicating perturbation of minor splicing and a higher degree of replicative stress, leading to accumulation of the Tp53 protein. In turn, this triggered a Tp53-dependent transcriptional program that increased transcription of genes involved in cell cycle arrest, senescence and apoptosis (Fig. 2H-K).

11. Statistical analyses:

Include statistical comparisons in Figures 2C and 6H/I to validate observed differences in phase distribution and gene expression.

Response: We did provide statistical comparisons between genotypes in Figure 2C; however, we did not state the test used in the figure legend. **We have now updated the figure legend by adding a statement that “Significance was tested using a one-way ANOVA with Tukey’s multiple comparisons test”.** Meanwhile, the statistical comparisons and p values for Figure 6H/I are already stated in the figure legend.

12. **Figure Consistency:** Standardize y-axis scaling across figures (e.g., confluency data in Figures 5B and 6J).

Response: We did not scale the y-axis (confluency) data across the two figures because the experiments giving rise to Figs. 5B and 6J were not exactly the same. In particular, minor splicing in Fig. 5B was inhibited with siRNAs to decrease the expression of *RNPC3*, whereas in Fig. 6J, minor splicing was inhibited with a potent ASO to U12 snRNA. Also, these two reagents required different transfection media (Dharmafect 1 and Lipofectamine RNAiMAX, respectively), which may vary in transfection efficiency.

13. Total cell count in Figure 6J: Verify whether changes in cell-cycle phases correspond to an overall reduction in cell number.

Response: The data in Fig. 6J were generated with an Incucyte Live Cell analysis system, which we used to monitor the confluency of unsynchronised A549 cells treated with U12 ASO for 72 h post-transfection. This platform calculates the %age confluency of live cells in each well over time. Meanwhile the data presented in panels 6H-I, were obtained in independent experiments by flow cytometry. While this makes it difficult to directly extrapolate data from the FACS experiment to the confluency data derived from the IncuCyte machine, **we have changed the final sentence on Page 20 of Results, to state that the decrease in cell confluency in response to treatment with 5-10nM U12 ASO is consistent with fewer cells in S-phase and more cells in G2.**

REVIEWER 3

Major points:

1. The authors provided rigorous data to convincingly show that heterozygous loss of *rnpc3* reduced hepatocyte hyperplasia in a zebrafish model of hepatocellular carcinoma. **However, they did not show any evidence that *rnpc3* reduction led to missplicing of minor introns in this zebrafish cancer model.** To strengthen the main claim of this study, the authors need to perform RT-PCR analyses on a few target minor introns to confirm that ~40% reduction of *rnpc3* is able to induce minor intron retention/alternative splicing.

Response: We are very pleased that this reviewer states that we “provided rigorous data to convincingly show that heterozygous loss of *rnpc3* reduced hepatocyte hyperplasia in a zebrafish model of hepatocellular carcinoma”. This is an accurate claim, and we are very careful throughout the paper not to extrapolate from this that *rnpc3* heterozygosity causes inhibition of minor splicing. However, it is worth mentioning here that in one of our previous studies (Markmiller S et al., PNAS 2014), we showed that *rnpc3* is indispensable for correct minor intron splicing *in vivo* during development. In this context, we found that the gradual depletion of maternally-deposited *rnpc3* mRNA from *rnpc3*^{-/-} zebrafish larvae from 3-7 days post fertilization restricted the rapid enlargement of the digestive organs, concomitant with the accumulation of aberrantly processed minor intron genes (MIGs) that could be readily detected by gel RT-PCR (Markmiller S, et al. PNAS 2014). This provides evidence of a role for *rnpc3* in zebrafish minor splicing.

Other factors contributing to our decision to conduct transcriptome-wide RNA-seq experiments in A549 cells, rather than a candidate approach in zebrafish, are discussed in detail on pages 1-2 of this document.

2. The authors need to **provide evidence that mice with a *Rnpc3*^{+/-} genotype indeed have reduced levels of *Rnpc3* in the mouse models of lung and gastric cancers.** It is well known that negative autoregulatory feedback is prevalent in splicing regulation. Some splicing factors can use a negative autoregulatory feedback to maintain constant levels when their own transcription is reduced. Therefore, it is important for the authors to confirm that *Rnpc3* levels are reduced in the heterozygous *Rnpc3*^{+/-}-mice.

Response: In response to this comment, we now show that *Rnpc3* mRNA levels are reduced by 50% in lung and gastric tissues harvested from heterozygous (*Rnpc3*^{+/-}) mice, compared to WT (Figure 3A). This argues against any involvement of autoregulatory mechanisms acting to restore *Rnpc3* expression levels back towards WT.

Minor points

1. On page 15, the authors wrote "133 (34.2%) exhibited significantly elevated IR" in the last paragraph. However, on page 16, the authors wrote "60 (10%) exhibited increased IR" in the last paragraph. These two percentage numbers are so different that the authors need to clearly explain this discrepancy. Why didn't the authors use their own RNA-seq data for the transcriptome-wide analysis with IRFinder?

Response: Firstly, we would like to clarify that we did use our own RNA-seq dataset for both sets of analyses; i.e.(1) the minor intron-specific analysis developed by our co-authors (A. Olthof and R. Kanadia) and (2) IRFinder, which uses a different algorithm to detect retention of major **and** minor introns (Middleton et al., 2017). We agree that the calculated percentages of retained minor introns differ between the two experiments (>34% versus 10%); however, we do not find this surprising given that two completely different algorithms were used to detect intron retention. Moreover, the conclusions from both sets of analyses are in agreement as they both clearly show that *RNPC3* knockdown has a selective effect on minor introns over major introns. **We have made this clearer in the paper in the paragraph bridging pages 17-18.**

2. **Figs. 4E and 4F, mice with *Rnpc3* lox/lox;UBC-CreERT2 AML cells survived longer than mice with *Rnpc3* lox/-;UBC-CreERT2 AML cells (29 d vs. 22 d).** This result seems to contradict with the authors' conclusion. Can the authors provide an explanation for this?

Response: This observation from Reviewer 3 prompted us to re-evaluate the survival data from our AML experiments. A detailed response to this question, along with a comment about the controls for our AML cell experiments are provided on page 4 of this document under "Controls".

3. **Is *RNPC3* upregulated in human lung cancer?**

Response: Analysis of the TCGA database and other recently curated datasets in cBioPortal suggests that *RNPC3* is not consistently up- or down-regulated in lung cancer, nor does its expression have a significant impact on patient survival.

4. Page 19 third paragraph

Response: "Fig. 7A, B" now reads **Fig. EV7A, B.**

5. Page 22, end of first paragraph, "**Fig. 7L**" should be **Fig. 6K.**

Response: This has been rectified.

6. In the Abstract, "correct expression of 755 genes" has been changed to "correct expression of approximately **700 genes**", because some genes have more than one minor intron. **Response: Agreed - this has been rectified.**

REVIEWER 4

Specific Comments:

1. The experiments shown in Fig. 1D-E should be performed using a *kras* wild-type (WT) setting to confirm the conclusions drawn.

Response: See specific response above in section “Controls” on page 3 of this document.

2. The statement that "cell cycle arrest and apoptosis are strongly suggestive of activation of the tumor suppressor functions of TP53" (page 7) is overly simplistic. **Numerous regulators control cell cycle and apoptosis. A more comprehensive or genome-wide analysis is necessary to identify the key regulators involved in this context.**

Response: We have changed the text on page 8, 2nd paragraph of our paper to read: “**Due to its known role in cell cycle arrest and apoptosis, we investigated whether the tumour suppressor protein Tp53 was activated in response to *rnpc3* heterozygosity**”. We did not perform a global analysis of gene expression in the zebrafish HCC model for reasons discussed on pages 1-2 of this document; instead, we carried-out a transcriptome wide analysis of gene expression in A549 cells in response to *RNPC3* knockdown.

3. **The observation that *kras* WT liver samples do not express *tp53* (Fig. 2A) raises questions about the zebrafish model's relevance. Does this imply that WT liver lacks *tp53* expression in zebrafish?** If so, this model may not be ideal for studying TP53-associated DNA damage responses, given the significant differences from human and mouse models.

Response: In normal cycling cells/tissues, Tp53 protein is generally absent/rarely detected due to post-transcriptional regulation by Mdm2. However, when cancer cells undergo replicative stress in response to oncogenic signalling, the Tp53 protein is stabilized and accumulates. We show this in Fig. 2E, where *tp53* mRNA expression is evident for all genotypes, but the Tp53 protein only accumulates in the context of *rnpc3* heterozygosity (Fig. 2A). This results in the activation of a Tp53-dependent transcription program that restricts tumour growth (Fig. 2D-K). Thus, the data we obtained from the zebrafish *kras* model is entirely consistent with how TP53 is regulated in mammalian cancer cells, including human.

4. **Fig. 2B does not appear to be cited in the manuscript text.**

Response: Fig. 2B shows representative liver images related to the liver volumes that are graphically represented in Fig. 2C. It was an oversight in the text on page 8 to only cite the quantification (Fig. 2C). **We have rectified this in the text.**

5. **To support the conclusions, global gene expression analyses are required to examine the roles of cell cycle, apoptosis, and DNA damage response pathways.**

Response: We generated a dataset of significantly differentially expressed genes (DEGs) from the RNA-seq experiment we conducted on A549 cells that had been treated with RNPC3 siRNA or a non-targeted (NT) siRNA control (Fig. 5). We then applied Ingenuity Pathway Analysis to reveal terms including Senescence, Cell cycle, DNA damage checkpoint regulation, Apoptosis and TP53 signalling, as being differentially expressed in response to knockdown of RNPC3 snRNA (Fig. 5H).

6. Protein expression levels of 65K (encoded by *Rnpc3*) and pERK in the AHH or adenoma lesion in Fig. 3B seem higher than those of *Rnpc3*^{+/+} counterpart in Fig. 3A. The differences shown in Fig. 3 are, if any, very subtle, which may significantly harm the authors' narrative.

Response: In Fig. 3A, B, we demonstrate the decreased area of the lesions in *Rnpc3*^{+/-} mice compared to *Rnpc3*^{+/+} mice. Because the ISH technique is not quantitative, we did not place much weight on the intensity on pERK and 65K staining. Rather, we interpret the data to mean that in the context of *Rnpc3* heterozygosity, small colonies may emerge that are pERK and 65K positive.

7. **The authors stated that "Our findings above, where we employed three different solid tumour models, suggest that hyperproliferative cancers require efficient minor splicing to proliferate and survive"** (in page 12). However, (minor) splicing was not analyzed at all.

Response: Agreed, this sentence was not accurate, and we have removed it from the text. The 2nd paragraph on page 13 now begins: **"Having shown that *Rnpc3* heterozygosity limits the growth of liver, lung and gastric hyperplasia/adenomas, we investigated whether hyperproliferative blood cancers were sensitive to *Rnpc3* expression as well."**

8. The critical comparison in Fig. 4E-F is the survival difference between *Rnpc3* lox/lox and *Rnpc3*^{+/+} mice treated with TMX. This data is missing. Additionally, the manuscript lacks evidence of AML development in recipients (At least Mac-1 and c-kit staining and HE staining of blood/bone marrow smear should be provided for the diagnosis).

Response: Please see our response on pages 3-5 of this document in section 4: "Controls".

9. **The manuscript states that "aberrant splicing of MIGs leads to DNA damage and the stabilization of TP53"** (page 19). However, TP53 protein stability was not analyzed, which undermines this conclusion.

Response: We refer this Reviewer to Fig. 6E, where we provided Western blot analysis demonstrating that TP53 protein accumulated in the nuclei of A549 cells in response to treatment with U12 ASO. We have now used the term **accumulation**, rather than stabilisation, in the sentence half-way down page 20.

10. The current title is overly generalized, as the study focuses solely on RNPC3. Furthermore, the findings contradict the known loss-of-function mutations in ZRSR2 in myeloid leukemia, which suggest opposite roles for the minor spliceosome in hematologic malignancies. **The title should be revised, and the conclusions should be appropriately tempered to reflect the study's scope**

Response:

1. Re. this reviewer's assertion that the loss-of-function mutations in ZRSR2 in myeloid leukemia suggest **opposite roles for the minor spliceosome in hematologic malignancies**, we think this is an over-simplification. Understanding the role of mutated ZRSR2 in myeloid cancers is a fascinating and intense area of investigation that remains unresolved. However, an aspect of ZRSR2 biology that appears to be of great significance is that human ZRSR2 resides on the X chromosome, whereas in other genomes (e.g. mice and zebrafish) it is found on autosomes. In a recent influential paper from Togami et al. (Cancer Discov. 2022 10:522-541), the authors connect mutations in *ZRSR2* to an extremely aggressive and male predominant leukaemia, BPDCN (Blastic plasmacytoid dendritic cell neoplasm). In their study, acquired *ZRSR2* mutations in haematopoietic progenitors lead to alterations in RNA splicing and impaired dendritic cell activation and apoptosis in response to inflammatory stimuli. Paraphrasing a huge amount of data in this paper, the authors propose that "*ZRSR2* is a male-biased X chromosome tumour suppressor gene". We surmise that *ZRSR2* operates in a **different** context to our studies of *RNPC3* and *PDCD7* deficiency, rather than playing an "**opposite**" role.

Meanwhile, studies in mouse and zebrafish models of *Zrsr2* deficiency (e.g. Weinstein et al. International Journal of Molecular Sciences (2022) 23:10668), produce outcomes that are very similar to those we obtained with reduced expression of *Rnpc3* and *Pdcd7*.

2. Looking at the title of our paper: "**Minor splicing is a therapeutically viable target for the treatment of a broad spectrum of cancers**", we think it is appropriate because (1) We targeted three minor spliceosome components in this study (*RNPC3*, *PDCD7* and *U12 snRNA*), not just *RNPC3* and (2) showed that a 50% dose of *Rnpc3* reduced tumour burden *in vivo* suggesting a viable therapeutic window and (3) employed a broad spectrum of different *in vitro* and *in vivo* cancer models. In summary, we think we are justified in requesting that the title of our paper remains the same.

Dear Karen,

Thank you for the submission of your revised manuscript. I sent it to referee 1 and referee 4 (now referee 2) and we received the comments and cross-comments below. I am uncertain about the meaning of the last sentence in referee 1's report and asked referee 4 for cross-comments. I also discussed the reports with my colleagues here, and we decided that we can offer to publish your ms but that referee 4's comment should be addressed. Please also adequately discuss the use of the different experimental systems with its advantages and disadvantages in the ms text.

A few editorial requests will also need to be addressed before we can proceed with the official acceptance of your manuscript:

- Please rename the conflict of interest statement to "Disclosure and Competing Interests Statement"
- There is one author name discrepancy: Benjamin B Williams in the ms vs. Benjamin B Borthwick in our online submission system, please correct.
- Please remove the bullet points in the REFERENCE list.
- FUNDING INFO is missing in our online system: Victorian State Government Operational Infrastructure Support grant, please add.
- A callout for Figure 6E is missing please add.
- There are 2 datasets uploaded, the correct title of the first one should be Dataset EV1 (instead of Data Set Appendix 1).
- The Reagent & Tools TABLE needs to be removed from the ms file and uploaded as a separate file.
- Materials and Methods should be just Methods.
- "Main manuscript: 6 Figures, 7 EV Figures, 1 Table, 2 EV Datasets." should be removed from the ms file.
- Please note that the specific valid URL for GSE190943 dataset is not provided in the data availability statement, please add.

Figure Legends - Comments

- Please note that the exact p values are not provided in the legends of figures 1C, E, G; 2C, F, G, I, M; 3F, 4C, F; 5A, B; 6C, G, I; EV1 B, EV3 B, EV4 A, B; EV5 C. Please provide exact p-values as reasonable.
- Please indicate the statistical test used for data analysis in the legends of figures 5A, D, H; EV5 B, EV5 C, F.
- Please note that information related to n is missing in the legends of figures 4C, 5D, EV5 C, F
- Please note that the error bars are not defined in the legends of figures 3F, 4C, EV2 H

- The ms title is an overstatement and must be changed. May be a title along the lines of "minor splicing inhibition restricts tumor growth" would be good ?
- Please write the abstract in present tense, according to journal policy.

EMBO press papers are accompanied online by A) a short (1-2 sentences) summary of the findings and their significance, B) 2-3 bullet points highlighting key results and C) a synopsis image that is exactly 550 pixels wide and 200-600 pixels high (the height is variable). The synopsis image should provide a sketch of the major findings, like a graphical abstract. Please note that text needs to be readable at the final size. Please send us this information along with the final manuscript.

Regarding the author Dr. Stephen Mieruszynski who is not reachable, can you please tell me what his contributions are? We could may be move him to the acknowledgements?

Referee #1:

The authors have carried out a revision that has addressed some of the issues raised by all reviewers, yet some critical problems are still present. Of high importance, they have provided evidence that knockdown of another component of the minor spliceosome (PDCD7) leads to similar results as those with knockdown of RNPC3 and U12 snRNA in A549 cells.

Overall, I think that there are enough interesting data in this study that would warrant publication. However, the authors should refocus their study and re-write the paper, concentrating on fewer experimental systems.

Specific comments

We appreciate the detailed and constructive feedback from the 4 reviewers. All express the view that our study is interesting and based on rigorous and convincing experimentation. They do, however, have many suggestions for its improvement, most of which boil down to a thirst for more detail. We agree that the points raised are interesting and relevant; however, most of the comments do not challenge our interpretation of the data or the conclusions we reach.

This is not accurate, the reviewers at least in my case, are not simply asking for more details, but rather suggesting a revision that leads to a more coherent paper that does not move from experimental system to experimental system, leading to a lack of depth and focus.

1- Novelty

We have corrected our omission by citing the significance of the Augsprach paper prominently in our revised manuscript (page 5, second paragraph).

This reviewer finds hard to believe that this was an honest omission, unless the authors do not look at the literature at all. It is difficult to comprehend the reasons behind this omission, but now that two reviewers have raised this critical issue, they have decided to cite this influential paper. Besides this, the authors are not addressing here the issue of novelty, and one does not create novelty by ignoring previous work in the field. What is indeed true, is that this study complements the Augspach et al paper.

4. CONTROLS

We identified three instances where the reviewers highlight a need for additional controls.

Reviewer 1, point 2: requests a knock-down (and/or heterozygosity) of another component of the minor spliceosome, to rule out other gene-specific effects (of RNPC3) that could operate independently of defects in minor splicing.

Response: In our revised paper, we demonstrate that reduced expression of two unique components of the minor spliceosome, RNPC3 and U12 snRNA, impact similarly on A549 cells to disrupt minor splicing and reduce their growth (Fig.5, Fig. EV4, Fig. EV5 and Fig.6A& J). In data not included in our original manuscript, we now show that knockdown of PDCD7 encoding the unique 59K protein component of the minor spliceosome, produces essentially the same results in A549 cells as knockdown of RNPC3 and U12 snRNA. These data diminish the possibility that RNPC3 knockdown produces gene-specific effects that operate independently of defects in minor splicing (see Fig. EV4A, C).

This is a crucial control that strengthens the conclusions of this paper.

3. The fragmented nature of this manuscript could be considered positive, due to the exploration of different cancer settings; however, it is not clear why the authors investigate global effects of RNPC3 in human A549 cells and not in any of the mouse models, used in previous figures.

Response: The reasons for this are discussed in detail on page 2 of this document.

I do sympathise with the problems to re-derive the mouse lines; however, this reviewer does not see the need or thinks that the large number of different experimental systems are indeed a positive aspect of this study.

Referee #2:

*The authors have addressed most of my previous concerns; however, I remain unclear as to why they have not included a direct comparison between *Rnpc3 lox/lox* and *Rnpc3+/+* mice in the presence of tamoxifen (TMX) in their leukemia models. This comparison is a widely accepted standard in the field, particularly given the well-documented effects of TMX on the hematopoietic system-even in Cre-negative mice.*

Several studies have highlighted this issue, including Hayashi & McMahon (2002), which demonstrated that TMX can independently alter tissue function; Hofmann et al. (2015), which reported TMX-induced toxicity in hematopoietic stem cells regardless of Cre status; and Feil et al. (2009), which emphasized the need for caution when using TMX in hematopoietic models due to its time- and dose-dependent effects. Numerous other reports support this concern.

*Given that the authors indicate these data are already available, I strongly encourage them to include the direct comparison and the corresponding source data to allow for proper interpretation of the phenotype observed in the *Rnpc3* conditional knockout model.*

Cross-comments from referee 2:

While I understand that Referee #1 may have found aspects of the authors' response partly lacking in sincerity-a sentiment I share to some extent-I believe the authors have provided sufficient experimental data to support their conclusions.

Given that their claims are scientifically well substantiated, I am of the opinion that requesting additional specific experiments would only serve to unnecessarily delay the publication.

Response to Reviewers Round 2

- Address Referee 4's comment

*The authors have addressed most of my previous concerns; however, I remain unclear as to why they have not included a direct comparison between *Rnpc3 lox/lox* and *Rnpc3+/+* mice in the presence of tamoxifen (TMX) in their leukemia models. This comparison is a widely accepted standard in the field, particularly given the well-documented effects of TMX on the hematopoietic system—even in *Cre*-negative mice.*

Regarding the controls for the AML experiments, we apologize for misunderstanding the reviewer's question. We now realize the inquiry was about the potential off-target/toxicity effects of Tamoxifen (TMX) on the hematopoietic system of mice receiving AML cells even in the absence of a *Cre* transgene. Although the question was correctly posed, we mistakenly assumed the cells the reviewer was referring to (both *Rnpc3* genotypes) also carried the *CreERT2* transgene. In short, we misunderstood that the reviewer was specifically asking about the impact of TMX alone on the longevity of the mice.

To address this issue, we offer the controls shown in Fig. EV3C, where we compare the survival of mice receiving *Rnpc3 +/+* AML cells in the presence and absence of TMX. We observed no difference in median survival, indicating that TMX had no systemic impact on the recipient mice. Similarly, when we compare the longevity of mice transplanted with *Rnpc3+/+;UBC-CreERT2* AML cells, we found no significant difference in the presence and absence of TMX.

While these are not the exact comparisons the reviewer requested, we hope they provide confidence that TMX was not a confounding factor in our experiments. We have inserted an extra sentence in this section of the manuscript to explain the importance of these controls (top of page 15).

To minimise the possibility that treatment of mice with TMX had a confounding influence on disease latency, we performed additional controls (Fig. EV3C).

- Please also adequately discuss the use of the different experimental systems with its advantages and disadvantages in the ms text.

We have inserted the selection below at the beginning of the Discussion.

Cancer of the digestive organs remains a leading cause of mortality worldwide, necessitating the discovery of novel therapeutic targets. In our study, we utilized a forward genetic screen in zebrafish to identify *rnpc3*, a gene crucial for rapid digestive organ growth during development. Remarkably, heterozygous *rnpc3* reduced liver overgrowth in a zebrafish model of hepatocellular carcinoma (HCC), underscoring the potential of zebrafish genetic

screens in identifying new cancer therapy targets. To extend our findings to mammals, we introduced *Rnpc3* heterozygosity into a variety of mouse cancer models, and again observed significantly reduced tumour growth. These results establish *Rnpc3* as a relevant therapeutic target across species and cancer types. To explore the molecular mechanisms underlying these observations, we turned to the human lung cancer cell line A549, known for its functional TP53 pathway. This model allowed us to study *TP53*-mediated responses to DNA damage and cell cycle regulation, providing deeper insights into the therapeutic potential of targeting *RNPC3*.

✓ Please rename the conflict of interest statement to "Disclosure and Competing Interests Statement"

✓ There is one author name discrepancy: Benjamin B Williams in the ms vs. Benjamin B Borthwick in our online submission system, please correct.

We have tried to correct this unfortunately because of our error made when first in putting author information the email address required for Benjamin has been linked to the incorrect name Benjamin B Borthwick and the system does not allow us to use this email address linked to his correct name Benjamin B Williams (see screenshot below). We can confirm his correct details are:

Benjamin B Williams, Email: b.borthwick.williams@gmail.com

5 Benjamin B Williams b.borthwick.williams@gmail.com Walter Eliza Hall Institute

There is a problem with the information provided for this user.

ORCID: N/A

Person Title: Please select

* Name: Benjamin B Williams
First Middle Last

Suffix:

* Email (institutional): b.borthwick.williams@gmail.com

This email address is already assigned to a different user account. Each author must have a unique email address. Click here to view/select an existing user.

✓ Please remove the bullet points in the REFERENCE list.

✓ FUNDING INFO is missing in our online system: Victorian State Government Operational Infrastructure Support grant, please add.

✓ A callout for Figure 6E is missing please add.

This has been corrected in the text (page 19, rows 7 and 10).

- ✓ There are 2 datasets uploaded, the correct title of the first one should be Dataset EV1 (instead of Data Set Appendix 1).
- ✓ The Reagent & Tools TABLE needs to be removed from the ms file and uploaded as a separate file.
- ✓ Materials and Methods should be just Methods.
- ✓ Main manuscript: 6 Figures, 7 EV Figures, 1 Table, 2 EV Datasets." should be removed from the ms file.
- ✓ Please note that the specific valid URL for GSE190943 dataset is not provided in the data availability statement, please add

Apologies, there was a mistake in the URL. This has now been corrected in the text.

Figure Legends - Comments

- ✓ Please note that the exact p values are not provided in the legends of figures 1C, E, G; 2C, F, G, I, M; 3F, 4C, F; 5A, B; 6C, G, I; EV1 B, EV3 B, EV4 A, B; EV5 C. Please provide exact p-values as reasonable.

All p values are reported as exact values unless $p < 0.0001$, in which case they are reported as $p < 0.0001$.

- ✓ Please indicate the statistical test used for data analysis in the legends of figures 5A, D, H; EV5 B, EV5 C, F

These details have been added to all relevant figure legends

- ✓ Please note that information related to n is missing in the legends of figures 4C, 5D, EV5 C, F

These details have been added to all relevant figure legends

- ✓ Please note that the error bars are not defined in the legends of figures 3F, 4C, EV2 H

- ✓ The ms title is an overstatement and must be changed. May be a title along the lines of "minor splicing inhibition restricts tumor growth" would be good ?

We propose the following title:

Inhibition of the minor spliceosome restricts the growth of a broad spectrum of cancers

- ✓ Please write the abstract in present tense, according to journal policy.

Minor splicing is an under-appreciated splicing system required for the correct expression of approximately 700 genes in the human genome. This small subset of genes (0.35%) harbour introns containing non-canonical splicing sequences that are recognised uniquely by the minor spliceosome and cannot be processed by the major spliceosome. Using *in vivo* zebrafish and mouse cancer models, we show that heterozygous expression of *Rnpc3*, encoding a unique protein component of the minor spliceosome, restricts the growth and survival of liver, lung and gastric tumours without impacting healthy cells. *RNPC3* knockdown in human lung cancer-derived A549 cells also impairs cell proliferation and RNA-seq analysis reveals a robust and selective disruption to minor intron splicing and transcription-wide effects on gene expression. We further demonstrate that these perturbations are accompanied by DNA replication stress, DNA damage, accumulation of TP53 protein and activation of a Tp53-dependent transcriptional program that induces cell cycle arrest and apoptosis. Together our data reveal a vulnerability of cancer cells to minor splicing inhibition that restricts tumour growth.

EMBO press papers are accompanied online by

- ✓ **(A) Synopsis: a short (1-2 sentences) summary of the findings and their significance**

Disrupting the expression of components of the minor spliceosome such as *RNPC3* reduces the rapid growth of oncogene-driven cancer cells in a broad spectrum of tumours. Targeting the minor spliceosome may provide an effective approach to reducing tumour burden clinically.

- ✓ **(B) 2-3 bullet points highlighting key results:**

- Cancer cells expressing strong oncogenes such as mutant *KRAS* are dependent on high rates of DNA replication, transcription and pre-mRNA processing that leads to DNA replication stress.
- Disrupting the expression of protein and snRNA components of the minor spliceosome exacerbates DNA replication stress, causing an increase in DNA damage and the accumulation of TP53 protein.
- TP53 upregulates a transcriptional program that induces cell cycle arrest, senescence and apoptosis, which act in concert to reduce tumour burden.

✓ (C) a synopsis image that is exactly 550 pixels wide and 200-600 pixels high (the height is variable).

The synopsis image should provide a sketch of the major findings, like a graphical abstract. Please note that text needs to be readable at the final size. Please send us this information along with the final manuscript.

New synopsis image (below) file uploaded in portal with manuscript files

Minor splicing disruption

✓ Regarding the author Dr. Stephen Mieruszynski who is not reachable, can you please tell me what his contributions are? We could may be move him to the acknowledgements?

This question is addressed in the new (Final) version of the Cover Letter.

Dear Karen,

Thank you for the submission of your final ms files. It looks good overall, but 3 issues still need to be resolved.

1. In your point-by-point response you only mention the positive aspects of using so many different cellular and animal models. This is not exactly a critical discussion. Both advantages and disadvantages need to be mentioned and discussed please. You can send us a new ms file by email and we will replace the current file.

2. The missing author approval needs to be clarified. We agree that Stephen Mieruszynski should be an author on the ms, given the contributions.

For regular research papers, the corresponding author needs to ensure s/he has full approval from all authors. We (the journal) do not require an affirmative answer from every author, but the corresponding author has to sign on behalf of all the authors and all authors sign off on the final published paper and are responsible for the content.

Our advice is that you do all you can to reach the person and make sure that you can document that. If nothing this author contributed changed in the final manuscript AND if the authorship order/positions did not change, you already have the approval.

3. I cannot find the a short (1-2 sentences) summary of your findings and their significance, and the 2-3 bullet points highlighting key results. I am sorry if I missed these, can you please send this again?

Thank you and best regards,
Esther

The Walter and Eliza Hall Institute of Medical Research
ABN 12 004 251 423
1G Royal Parade Parkville Victoria 3052 Australia
T +61 3 9345 2555 F +61 3 9347 0852
www.wehi.edu.au

June 12 2025

Re: EMBOR-2025-61344-T

Dear Esther,

We would like to resubmit our re-revised manuscript: **Inhibition of the minor spliceosome restricts the growth of a broad spectrum of cancers.**

Regarding the 3 outstanding issues, please see our responses below.

We have added an additional sentence to the end of the first paragraph of the Discussion to address the potential disadvantages to our approach:

1. “However, while we were successful in demonstrating the wide applicability of our results across a variety of *in vivo* and *in vitro* cancer models, there are also disadvantages to our multifaceted approach. For example, if we had focused more strongly on a sub-set of our *in vivo* cancer models, we may have been successful in demonstrating a formal link between *Rnpc3* heterozygosity and aberrant minor splicing, which is absent from our current study.”

As requested, we attach a new ms file to this email to replace the current file on your records. The new text is highlighted in red font.

2. **(A) Synopsis: a short (1-2 sentences) summary of the findings and their significance**

Disrupting the expression of components of the minor spliceosome such as *RNPC3* reduces the rapid growth of oncogene-driven cancer cells in a broad spectrum of tumours. Targeting the minor spliceosome may provide an effective approach to reducing tumour burden clinically.

(B) 2-3 bullet points highlighting key results:

- Cancer cells expressing strong oncogenes such as mutant *KRAS* are dependent on high rates of DNA replication, transcription and pre-mRNA processing that leads to DNA replication stress.
- Disrupting the expression of protein and snRNA components of the minor spliceosome exacerbates DNA replication stress, causing an increase in DNA damage and the accumulation of TP53 protein.
- TP53 upregulates a transcriptional program that induces cell cycle arrest, senescence and apoptosis, which act in concert to reduce tumour burden.

(C) a synopsis image that is exactly 550 pixels wide and 200-600 pixels high (the height is variable).

Minor splicing disruption

3. We can confirm that we have maintained a comprehensive record of how many times and methods we have used to try and reach Dr Mieruszynski for the purpose of obtaining his agreement to be a co-author on this paper. Moreover, we attest that nothing this author contributed changed in the final manuscript, and the authorship order/positions did not change. As co-corresponding authors, we are happy to sign-off on behalf of all the authors and we take very seriously that it is the responsibility of all authors to sign-off on the final published paper and to be responsible for the content.

We hope our re-revised paper now meets your expectations and that you are ready to provide formal acceptance of our paper. Please do not hesitate to contact us if further clarification or information is needed.

We look forward to hearing from you at your earliest convenience.

Kind regards,

Karen Doggett PhD
Co-corresponding author

Joan Heath PhD
Co-corresponding author

Joan Heath, PhD | Laboratory Head, Genetics and Gene Regulation Division
The Walter and Eliza Hall Institute of Medical Research
Honorary Professorial Fellow, Department of Medical Biology, University of Melbourne

1G Royal Parade | Parkville Melbourne | Vic 3052 | Australia

T +61 3 9345 2872 | M +61 (0)488 599 996

Member, Ludwig Cancer Research

E joan.heath@wehi.edu.au

W <https://www.wehi.edu.au/researcher/joan-heath/>

Dr. Karen Doggett
Walter and Eliza Hall Institute of Medical Research
Inflammation
1G Royal Parade
Parkville, Vic 3052
Australia

Dear Karen,

I am very pleased to accept your manuscript for publication in the next available issue of EMBO reports. Thank you for your contribution to our journal.
